# ONE-STEP RESIDUAL SHIFTING DIFFUSION FOR IMAGE SUPER-RESOLUTION VIA DISTILLATION

## ABSTRACT

Diffusion models for super-resolution (SR) produce high-quality visual results but require expensive computational costs. Despite the development of several methods to accelerate diffusion-based SR models, some (e.g., SinSR) fail to produce realistic perceptual details, while others (e.g., OSEDiff) may hallucinate non-existent structures. To overcome these issues, we present **RSD**, a new distillation method for ResShift. Our method is based on training the student network to produce images such that a new fake ResShift model trained on them will coincide with the teacher model. RSD achieves single-step restoration and outperforms the teacher by a noticeable margin in various perceptual metrics (LPIPS, CLIPIQA, MUSIQ, DISTS, NIQE, MANIQA). We show that our distillation method can surpass the other distillation-based method for ResShift - SinSR - making it on par with state-of-the-art diffusion-based SR distillation methods with low computational costs in terms of perceptual quality. Compared to SR methods based on pre-trained text-to-image models, RSD produces competitive perceptual quality and requires fewer parameters, GPU memory, and training cost. We provide experimental results on various real-world and synthetic datasets, including RealSR, RealSet65, DRealSR, ImageNet, DIV2K, RealLR200 and RealLQ250.

## 1 INTRODUCTION

Single image super-resolution (SR) (Dong et al., 2016; Glasner et al., 2009; Irani & Peleg, 1991) belongs to the category of inverse imaging problems aiming to reconstruct the high-resolution (HR) image given its low-resolution (LR) observation suffering from degradations. These degradations are usually *complex and unknown* for real-world scenarios when dealing with digital single-lens reflex cameras (Cai et al., 2019; Wei et al., 2020; Ignatov et al., 2017), referred to as the blind real-world image SR problem (Real-ISR). The SR problem is highly ill-posed, and many methods have been proposed in the literature to address it.

Recently, diffusion models (DMs) have been developed for the blind SR problem and became a strong alternative for methods based on generative adversarial networks (GAN) (Wang et al., 2021; Zhang et al., 2021; Ji et al., 2020) due to their good capabilities to learn complex data distributions (Dhariwal & Nichol, 2021). The competitive perceptual quality of DMs for different Real-ISR problems is supported by bigger human evaluation preferences compared to GAN-based methods, as shown in (Saharia et al., 2023; Wang et al., 2024a). Early diffusion methods for SR constructed a denoising process, which starts from Gaussian prior and ends in the HR image, while the LR image is used as a condition for the denoiser (Saharia et al., 2023; Rombach et al., 2022; Luo et al., 2023b).

However, this strategy also leads to large computational resources and slow inference, requiring dozens or hundreds for the number of function evaluations (NFE) of the denoiser and limiting DMs based on these strategies from practically important real-time SR on consumer devices. Consequent works for SR methods with DMs developed different approaches to accelerate those models while maintaining their high quality. Among them, ResShift (Yue et al., 2023) achieves perceptually high results in solving the Real-ISR problem using only 15 NFE. This model surpasses the results of state-of-the-art (SOTA) models from the other classes, including GANs (Wang et al., 2021; Zhang et al., 2021), transformers (SwinIR, (Liang et al., 2021)) and previous DMs (Rombach et al., 2022).

Unfortunately, the inference time for ResShift still remains 10 times larger than that of GAN-based models, as shown in ResShift (Table 2). The challenge arises when considering the problem of further acceleration of DMs while maintaining their perceptual quality at the same level. As shown in SinSR (Wang et al., 2024b), ResShift exhibits degraded results if the NFE is further reduced. To solve this issue, SinSR proposed a **knowledge distillation** for ResShift in 1 NFE, which is based on the

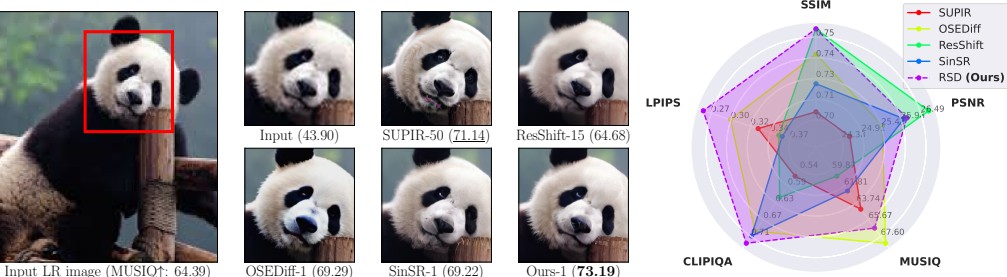

Figure 1: **Left.** A comparison between the recent diffusion-based methods for Real-ISR - ResShift, SinSR, OSEDiff, SUPIR - and the proposed RSD method. RSD has the following advantages: **(1)** It achieves superior perceptual quality compared to SinSR; **(2)** It requires less computational resources compared to OSEDiff; see Table 4. ("-N" behind the method name is the NFE, and the value in the bracket is MUSIQ↑ for full images). Please zoom in ×5 times for a better view. **Right.** Comparison among diffusion SR methods on RealSR. RSD (**Ours**) achieves top scores on most metrics while remaining computationally efficient compared to T2I methods such as OSEDiff and SUPIR.

deterministic sampling formulation of the reverse process for ResShift inspired by DDIM sampling (Song et al., 2021a). But SinSR tends to produce blurry results, as can be seen in the first row of Figure 3 and was also noted in recent works (Wu et al., 2024a; Sun et al., 2024; Dong et al., 2025).

Another promising direction in acceleration of DMs for SR is to add conditioning on the LR image to pre-trained text-to-image (T2I) models (Rombach et al., 2022; Saharia et al., 2022; Podell et al., 2024) with LoRA (Hu et al., 2022) and distill them with **variational score distillation** (VSD) (Wang et al., 2023c; Yin et al., 2024b; Dao et al., 2025) as proposed by OSEDiff (Wu et al., 2024a). While this approach greatly reduced NFE from tens or even hundreds to one across the class of T2I-based SR models (Wang et al., 2024a; Lin et al., 2025; Yang et al., 2025; Wu et al., 2024b; Yu et al., 2024) and achieved better perceptual results than ResShift and SinSR, we observe the following issues with T2I-based models for the SR problem: (1) as we show Table 4, using computationally expensive T2I architectures like Stable Diffusion (Rombach et al., 2022) or SDXL (Podell et al., 2024) still leads to high computational inference and trainings costs with ×10 more parameters than SinSR; (2) T2I-based models for SR also produce lower full-reference fidelity metrics such as PSNR and SSIM (Wang et al., 2004) when compared with ResShift and SinSR, as shown for various synthetic and Real-ISR benchmarks in Table 2 and Table 3 , aligning with (Wu et al., 2024a; Sun et al., 2024).

Due to these issues of distillation methods for diffusion SR, we address the following **2 questions**.

1. Are knowledge and variational score distillations the best methods for efficient 1-step SR DMs?
2. Can we unite the *best of two worlds* for these distillation methods and achieve a 1-step diffusion-based SR model that has a good perceptual quality comparable to recent T2I-based diffusion SR models, like OSEDiff, and use relatively small computational costs, like SinSR, at the same time?

**Contributions**. Our main contributions are as follows:

**(I) Theory**. Inspired by the successful distillation of ResShift achieved by SinSR and recent progress in the distillation of image-to-image DMs (He et al., 2024; Gushchin et al., 2025), we propose a novel objective for the 1-step distillation of the diffusion-based SR model **in discrete time** and derive its tractable version. We show the difference between the proposed objective and the objective of VSD. Motivated by ResShift's good perception-distortion trade-off across DMs and its justified diffusion process, we build our method on top of it and name it as **RSD**: **R**esidual **S**hifting **D**istillation.

**(II) Practice**. We show that RSD trained with the proposed objective combined with additional supervised losses notably surpasses the teacher's results on the Real-ISR problem for various perceptual metrics, including LPIPS (Zhang et al., 2018a), CLIPIQA (Wang et al., 2023a), and MUSIQ (Ke et al., 2021). We show that in practice, discrete-time RSD objective leads to much better performance compared to the related continuous-time IBMD objective (Gushchin et al., 2025) for Real-ISR (Appendix A.3). Our method aims to improve the compromise between fidelity, perceptual quality, and computational efficiency for Real-ISR with DMs in several aspects, see Figure 1 and Table 4:

1. **Perceptual quality**. Compared to the other 1-step distillation method for ResShift (SinSR), our method achieves better perceptual results on synthetic and real data for the blind SR problem.
2. **Performance-efficiency trade-off**. Compared to the other 1-step diffusion SR model, which is based on pre-trained T2I models, OSEDiff, our method provides competitive perceptual results.

At the same time, to bring DMs closer to real-time SR applications, RSD suggests an alternative 1-step model with much lower computational costs compared to T2I-based SR models.

## 2   RELATED WORK

**GAN-based SR models**. With the rise of GANs (Goodfellow et al., 2014), one line of research adapted the GAN framework to the SR problem (Ledig et al., 2017; Sajjadi et al., 2017; Wang et al., 2021; Zhang et al., 2021) and achieved much better perceptual quality of the generated images than previously developed regression-based methods (Dong et al., 2016; Kim et al., 2016; Lim et al., 2017; Zhang et al., 2018b), which minimize the mean squared error (MSE) between predicted and ground truth HR images. Among these works, Real-ESRGAN (Wang et al., 2021) and BSRGAN (Zhang et al., 2021) suggested complex degradation pipelines to synthesize LR-HR image pairs for modeling real-world data. Previous methods assumed a pre-defined degradation (e.g., bicubic) leading to limited generalization. The degradations of Real-ESRGAN and BSRGAN improved the results of GAN-based Real-ISR models and have also been widely used by diffusion (Yue et al., 2023; Wu et al., 2024a; Wang et al., 2024a) and transformer SR models (Liang et al., 2021).

**Diffusion-based SR models.** Existing methods, which adapt DMs (Ho et al., 2020; Song et al., 2021b; Song & Ermon, 2019; Sohl-Dickstein et al., 2015) for the blind SR problem, can be split into several categories depending on how they utilize the LR image. The first category of methods uses the LR image as an additional condition for the denoiser and trains the denoiser from scratch (Saharia et al., 2023; Rombach et al., 2022; Luo et al., 2023b). The second category of methods uses unconditional to the LR image pre-trained diffusion priors and modifies their reverse process (Wang et al., 2024a; Choi et al., 2021; Chung et al., 2022). The third category of SR methods with DMs argues that large NFE is needed for those models due to the Gaussian prior, which is not optimal for an SR problem where the LR image already contains structural information about the HR image. Following this motivation, methods from the third category suggest starting the denoising process from the combination of the LR image and a random noise (Yue et al., 2023; Liu et al., 2023b; Yue et al., 2024). The model of this class, ResShift (Yue et al., 2023) has two advantages: **(1)** it achieves SOTA results for blind Real-ISR using only 15 NFE; at the same time, the methods (Liu et al., 2023b; Yue et al., 2024) considered only simple degradations and used hundreds of NFE; **(2)** compared to LDM (Rombach et al., 2022), ResShift also performs a diffusion in the latent space of an autoencoder (Esser et al., 2021), but is 2-4 times faster and provides a better perception-distortion trade.

**Acceleration of diffusion-based SR models**. Although DMs surpass GANs in generative performance (Dhariwal & Nichol, 2021), their slow inference remains the key challenge. To mitigate this issue, various acceleration techniques have been proposed, with distillation emerging as one of the most effective. These methods have also been extended to SR models with DMs. For example, to improve the efficiency of ResShift, SinSR (Wang et al., 2024b) applied knowledge distillation to its diffusion process. With the consistency-preserving loss that uses ground-truth data during training, SinSR achieved results comparable to the teacher ResShift for blind Real-ISR with only 1 NFE. In our work, we draw inspiration from distillation methods that involve training an auxiliary "fake" model (Luo et al., 2023a; Yin et al., 2024a; Zhou et al., 2024; Huang et al., 2024; Gushchin et al., 2025). We give a detailed discussion of relation of our method to these approaches in Appendix A. CTMSR (You et al., 2025) proposed a one-step distillation-free method leveraging recent advances in consistency training (Song et al., 2023; Song & Dhariwal, 2024).

**T2I-based SR models**. However, as pointed out in recent works (Wu et al., 2024a;b; Dong et al., 2025), ResShift and SinSR show worse perceptual metrics and may fail to synthesize realistic structures when compared with other models, which exploit pre-trained T2I DMs for blind Real-ISR problem. The reason for this is the limited generalization of those models, which is constrained due to the absence of large-scale data for the training. In contrast, T2I models were trained on billions of image-text pairs and became the natural choice for applying to Real-ISR. To adapt T2I models for SR problem, such methods have two components: **(1)** conditioning on the LR image is realized with T2I controllers such as LoRA layers (Hu et al., 2022) (OSEDiff (Wu et al., 2024a), PiSA-SR (Sun et al., 2025)), ControlNet (Zhang et al., 2023) (SeeSR (Wu et al., 2024b), DiffBIR (Lin et al., 2025), SUPIR (Yu et al., 2024)) or other modules (StableSR (Wang et al., 2024a), PASD (Yang et al., 2025)); **(2)** prompts for LR images are used as predefined (StableSR, DiffBIR, TSD-SR (Dong et al., 2025)) or extracted with additional models (SeeSR, OSEDiff, SUPIR, PASD).

Such adaptations to SR problem also lead to challenges. The first challenge is their computationally demanding requirements, as many methods utilizing pre-trained T2I models for Real-ISR require tens

or even hundreds of NFE (Wang et al., 2024a; Lin et al., 2025; Yang et al., 2025; Wu et al., 2024b; Yu et al., 2024). The recently developed one-step diffusion distillation methods utilize variational score distillation (VSD) (Wang et al., 2023c; Yin et al., 2024b; Dao et al., 2025) (OSEDiff (Wu et al., 2024a)), adversarial diffusion distillation (ADD) (Sauer et al., 2025) (AddSR (Xie et al., 2024)), or target score distillation (TSD-SR (Dong et al., 2025)). These methods significantly accelerate the inference of T2I-based SR models but do not solve the problem of big T2I architectures with billion parameters. To reduce the problem of heavy architectures AdcSR (Chen et al., 2025) proposed the knowledge distillation method applied for OSEDiff, which is based on adversarial training of the compressed student network by removing and pruning teacher modules. InvSR (Yue et al., 2025) proposed a diffusion inversion technique for Real-ISR problems, which supports an arbitrary number of NFE for the trained T2I-based SR model. The second challenge is their unstable predictions for a fixed input due to the high dependence on noise initialization for the start of the denoising process, as pointed out in CCSR (Sun et al., 2024). Such instability may lead to poor fidelity and random unfaithful details. PiSA-SR (Sun et al., 2025) proposed the T2I-based SR model, which can adjust the perception-distortion trade-off (Blau & Michaeli, 2018) during inference without re-training.

## 3 METHOD

We start by recalling the ResShift formulation in §3.1. Then, we propose our method for distillation of the ResShift teacher in a one-step generator and derive its computationally tractable form in §3.2. We expand the method to the multistep generator in §3.3 and add additional supervised losses in §3.4. We then formulate the final objective for our RSD method in §3.5. We discuss the novelty of RSD in relation to existing distillation methods in §3.6.

**Remark.** While we derive our distillation method for ResShift, we note that ResShift is essentially a conditional DDPM (Ho et al., 2020), where the forward process ends in a Gaussian centered at the LR image. RSD can be generalized to any DMs built on the DDPM framework, see Appendix A.4

### 3.1 BACKGROUND

As a part of DMs, ResShift can be described by specifying the forward (noising) process, the parametrization of the reverse (denoising) process, and the objective for training the reverse process.

**Forward process.** Consider a pair of $(\text{LR}, \text{HR})$ images $(y_0, x_0) \sim p_{\text{data}}(y_0, x_0)$. For a residual $e_0 = y_0 - x_0$, ResShift uses the forward process with Gaussian kernel:

$$q(x_t|x_{t-1}, y_0) = \mathcal{N}(x_t|x_{t-1} + \alpha_t e_0, \kappa^2 \alpha_t \mathbf{I}), \tag{1}$$

where $\alpha_t = \eta_t - \eta_{t-1}$, $\alpha_1 = \eta_1$ and $\{\eta\}_{t=1}^{T}$ is a schedule, while $k$ is a hyper-parameter controlling the noise variance. The corresponding posterior distribution is given as:

$$q(x_{t-1}|x_t, x_0, y_0) = \mathcal{N}\left(x_{t-1}\left|\frac{\eta_{t-1}}{\eta_t}x_t + \frac{\alpha_t}{\eta_t}x_0, \frac{\kappa^2 \eta_{t-1}}{\eta_t}\alpha_t \mathbf{I}\right.\right) \tag{2}$$

**Reverse process.** ResShift suggests the reverse process in the following parametrized form:

$$p_\theta(x_0|y_0) = \int p(x_T|y_0) \prod_{t=1}^{T} p_\theta(x_{t-1}|x_t, y_0) dx_{1:T} \tag{3}$$

Here $p(x_T|y_0) = \mathcal{N}(x_T|y_0, \kappa^2 I)$, and $p_\theta(x_{t-1}|x_t, y_0)$ is the reverse transition kernel from $x_{t-1}$ to $x_t$ approximated with Gaussian distribution with parameters $\mu_\theta$ and $\Sigma_\theta$.

**Objective.** ResShift sets the variance parameter $\Sigma_\theta(x_t, y_0, t)$ to be independent of $x_t$ and $y_0$ and reparametrizes the parameter $\mu_\theta(x_t, y_0, t)$ as:

$$\mu_\theta(x_t, y_0, t) = \frac{\eta_{t-1}}{\eta_t}x_t + \frac{\alpha_t}{\eta_t}f_\theta(x_t, y_0, t), \tag{4}$$

where $f_\theta$ is a deep neural network with parameter $\theta$, aiming to predict $x_0$. The training objective is:

$$\min_\theta \sum_{t=1}^{T} \mathbb{E}_{p(x_0, y_0, x_t)} w_t \|f_\theta(x_t, y_0, t) - x_0\|^2, \tag{5}$$

where $w_t$ are some positive weights and $p(x_0, y_0, x_t)$ is provided by the forward process of ResShift. More detailed information on ResShift can be found in Appendix I.

### 3.2 RESIDUAL SHIFTING DISTILLATION (RSD)

Our goal is to distill a given ResShift *teacher* $f^*(x_t, y_0, t)$ into a stochastic one-step *student* generator $G_\theta$, which maps the LR image $y_0$ to the HR image $x_0$. To achieve it, we parametrize the generator

**Figure 2: The training framework of RSD.** We begin by encoding the (LR, HR) pair into the latent space $(z_y, z_0)$. First, to compute $\mathcal{L}_{\text{LPIPS}}$, we use $z_y$ to sample $z_T$ and generate the output $\widehat{z}_0$ from timestep $T$ (following procedure of one-step inference), then decode it back to pixel space to obtain $\widehat{x}_0$. Then, we obtain $z_{t_n}$ from the forward diffusion process in latent space (1) and generate $\widehat{z}_0^{t_n}$. We then perform posterior sampling (2) to obtain $z_t$, process it using both the fake and teacher ResShift models, and compute the distillation losses $\mathcal{L}_\theta$ and $\mathcal{L}_{\text{fake}}$ from Proposition 3.1. To compute $\mathcal{L}_{\text{GAN}}$, we extract features from the encoder part of the fake model $f_\phi$ and use an additional discriminator head.

$\widehat{x}_0 = G_\theta(x_T, y_0, \epsilon)$ to have three inputs: the LR image $y_0$, its noisy version $x_T \sim q(x_T|y_0)$ and additional noise input $\epsilon \sim \mathcal{N}(\epsilon|0, I)$. We denote by $p_\theta(\widehat{x}_0|x_T, y_0)$ the distribution of $G_\theta$ produced for given $y_0, x_T$ and random $\epsilon$. Then, **we force the generator to produce such data $p_\theta(\widehat{x}_0|y_0)$, that ResShift trained on it will coincide with the teacher model** $f^*(x_t, y_0, t)$. We assume that if the ResShift $f_{G_\theta}$, which was learned on student outputs $G_\theta$, and the teacher model $f^*$, which was learned on real data, coincide, then generator outputs and datasets data come from the same distributions:

$$f_{G_\theta} \approx f^* \Rightarrow p_\theta(y_0, x_0) \approx p_{\text{data}}(y_0, x_0) \tag{6}$$

Based on this assumption, we propose the following problem to align student $G_\theta$ by producing the data from the same distribution of LR-HR pairs as were in the train datasets for the teacher $f^*$:

$$\min_\theta \mathcal{L}_\theta, \quad \text{where} \quad \mathcal{L}_\theta \overset{\text{def}}{=} \sum_{t=1}^T w_t \mathbb{E}_{p_\theta(\widehat{x}_0, y_0, x_t)} \|f_{G_\theta}(x_t, y_0, t) - f^*(x_t, y_0, t)\|_2^2 \tag{7}$$

and $p_\theta(\widehat{x}_0, y_0, x_t)$ is given by mapping LR image by a generator $\widehat{x}_0 = G_\theta(x_T, y_0, \epsilon)$ and using posterior distribution $q(x_t|\widehat{x}_0, y_0)$ (2). In turn, $f_{G_\theta}(x_t, y_0, t)$ is the ResShift trained on the generator data $p_\theta(\widehat{x}_0|y_0)$. $\nabla_\theta \mathcal{L}_\theta$ includes the term $\nabla_\theta f_{G_\theta}(x_t, y_0, t)$, which is not tractable since backpropagation through the whole learning of the ResShift $f_{G_\theta}(x_t, y_0, t)$ is computationally infeasible, as discussed in Appendix K. To solve the problem, we propose another equivalent expression of $\mathcal{L}_\theta$:

**Proposition 3.1.** *Given a teacher model $f^*$, loss in* (7) *can be evaluated in a tractable form:*

$$\mathcal{L}_\theta = -\min_\phi \Big\{ \sum_{t=1}^T w_t \mathbb{E}_{p_\theta(\widehat{x}_0, y_0, x_t)} \Big( -\|f^*(x_t, y_0, t)\|_2^2 +$$

$$\underbrace{\|f_\phi(x_t, y_0, t)\|_2^2 - 2\langle f_\phi(x_t, y_0, t) - f^*(x_t, y_0, t), \widehat{x}_0\rangle}_{\text{This objective } \mathcal{L}_{fake} \text{ is equivalent to training a fake model } f_\phi \text{ with objective (5).}} \Big) \Big\}. \tag{8}$$

*Here, $f_\phi$ is an additional ResShift trained to optimize $\mathcal{L}_{fake}$ in* (8) *for estimation of $\mathcal{L}_\theta$. Furthermore, minimizing* (8) *over $\phi$ is equivalent to training a "fake" ResShift using data generated by $G_\theta$.*

Thus, we solve the intractable gradient problem in (7) by incorporating the fake ResShift model training into $\mathcal{L}_\theta$ (7). The proof of Proposition 3.1 is in Appendix K. In Appendix A.1, we also compare our method with the VSD used in OSEDiff. We provide an additional point of view on the motivation for the proposed RSD loss (7) in Appendix A.1, showing that the RSD loss (7) is equivalent to:

$$\mathcal{L}_\theta = \mathbb{E}_{p(y_0)} \mathcal{D}_{\text{KL}}\left(p(x_{0:T}|y_0) \,\|\, p^*(x_{0:T}|y_0)\right) \tag{9}$$

### 3.3 MULTISTEP RSD TRAINING

To further improve the quality of images produced by our method, we consider the multistep training of the generator following previous diffusion distillation works (Yin et al., 2024a; Zhou et al., 2024;

Song et al., 2023). We fix a subset of $N$ timesteps $1 < t_1 < \cdots < t_N = T$ and append additional time conditioning for the generator $G_\theta(x_t, t, y_0, \epsilon)$. We denote by $\widehat{x}_0^{t_n}$ output of $G_\theta(x_t, t, y_0, \epsilon)$ at timestep $t_n$. In this setup, the generator $G_\theta$ should approximate distributions $p_\theta(\widehat{x}_0|x_{t_n}, y_0) \approx q(x_0|x_{t_n}, y_0)$ for all fixed timesteps $t_n$ instead of only approximating the distribution $p_\theta(\widehat{x}_0|x_T, y_0) \approx q(x_0|x_T, y_0)$ in one-step training. For multistep training, we generate input data $q(x_{t_n}|y_0)$ using ground truth data distribution $p(x_0|y_0)$ of LR and HR images and posterior distribution (2). Then, we use the objective from Proposition 3.1 to train the generator for all $t_n$ simultaneously. At inference, we use a single sampling step to maximize speed. This strategy shows better results than one-step training since training across multiple time steps appears to help the network learn more robust mappings (see Table 5). For consistency, we denote the output of the single-step network at the timestep $T$ by $\widehat{x}_0$.

### 3.4 SUPERVISED LOSSES

In our distillation approach, we rely on the teacher's prediction to guide the solution. However, this approach may yield suboptimal results due to inherent approximation errors in the estimation of $x_0$. To mitigate this issue, we integrate additional losses into the distillation process.

**LPIPS loss.** Inspired by OSEDiff, we used LPIPS loss in our approach. By employing LPIPS loss ($\mathcal{L}_{\text{LPIPS}}$ (Zhang et al., 2018a)), we enable the student to directly compare its output with the HR ground truth in terms of perceptual features. This comparison helps the network to recover essential textures and structural details that might be missed when relying on the teacher's guidance. Although OSEDiff also used MSE loss for better fidelity alignment, we found that it did not help in our setup.

**GAN loss.** In line with DMD2 (Yin et al., 2024a), we integrate a GAN loss into our framework. Incorporating the GAN loss enhances the student model's capacity to align its predictions with the distribution of HR images, thereby yielding overall superior image quality. Our minimalist design - adding a classification branch to the bottleneck of the fake ResShift (see Figure 2) - mirrors DMD2. While previous works (Yin et al., 2024a; Xu et al., 2024) implemented GAN loss for comparing marginal distributions of noised data and generator output, we notice that using GAN loss to compare data distribution $p_{\text{data}}(x_0|y_0)$ with generator distribution $p_\theta(\widehat{x}_0^{t_n}|y_0)$ at each $t_n$ is more effective:

$$\mathcal{L}_{\text{GAN}} = \mathbb{E}_{p_{\text{data}}(x_0|y_0)}\big[\log D\big(x_0|y_0\big)\big] - \mathbb{E}_{p_\theta(\widehat{x}_0^{t_n}|y_0)}\big[\log D\big(\widehat{x}_0^{t_n}|y_0\big)\big]. \tag{10}$$

### 3.5 PUTTING EVERYTHING TOGETHER

**Translation into a latent space.** Thus far, we assumed that losses operate in the image space (denoted as $x$), although the ResShift was originally trained in the latent space (denoted as $z$). We also move our losses to the latent space, eliminating latent encoding and decoding when computing losses. Specifically, we calculate the distillation loss ($\mathcal{L}_\theta$) and GAN loss ($\mathcal{L}_{\text{GAN}}$) in the latent space, while the LPIPS loss ($\mathcal{L}_{\text{LPIPS}}$) remains in the image space, as the LPIPS network was originally trained there.

**Final algorithm.** The final loss function for each $t_n$ is:

$$\mathcal{L}_\theta + \lambda_1 \mathcal{L}_{\text{LPIPS}} + \lambda_2 \mathcal{L}_{\text{GAN}} \tag{11}$$

A full description of the RSD algorithm is given in Appendix B, with an illustration in Figure 2.

### 3.6 DIFFERENCE BETWEEN RSD, VSD AND ADD

We note that the proposed RSD loss (7) differs from VSD and ADD distillation losses.

Specifically, we show that our method is different from VSD conceptually and computationally, and discuss the relation of the RSD loss (7) with the VSD loss (see Eq. 5 in VSD (Wang et al., 2023c)) in Appendix A.1. The key difference is that the VSD loss (16) aligns the marginal distributions at each timestep $t$ between the teacher's and fake's distributions, while the RSD loss (9) matches the joint distribution across all $t$. In Section 4.2, we discuss the results of RSD and VSD for the SR.

The SDS loss (Poole et al., 2023), which was adopted in ADD (Sauer et al., 2025) and its SR extension, AddSR (Xie et al., 2024), is closely related to the VSD loss (16), but has a different implementation. Specifically, the gradient $\nabla_\theta \log p(x_t|y_0)$ is estimated using the score of the model distribution, which is computed via a single-sample Monte Carlo approximation (Eq. 4 in (Poole et al., 2023)). This implies that SDS-based methods do not require an auxiliary (fake) model, but suffer from high gradient variance and training instability (Section 3.3 in (Wang et al., 2023c)). In Appendix H, we show that AddSR achieves worse results than RSD, while having $\times 10$ more parameters.

## 4 EXPERIMENTS

In this section, we aim to achieve two main goals: **(1)** to demonstrate that our proposed *distillation* method outperforms existing *distillation* methods under the same experimental setup. We chose

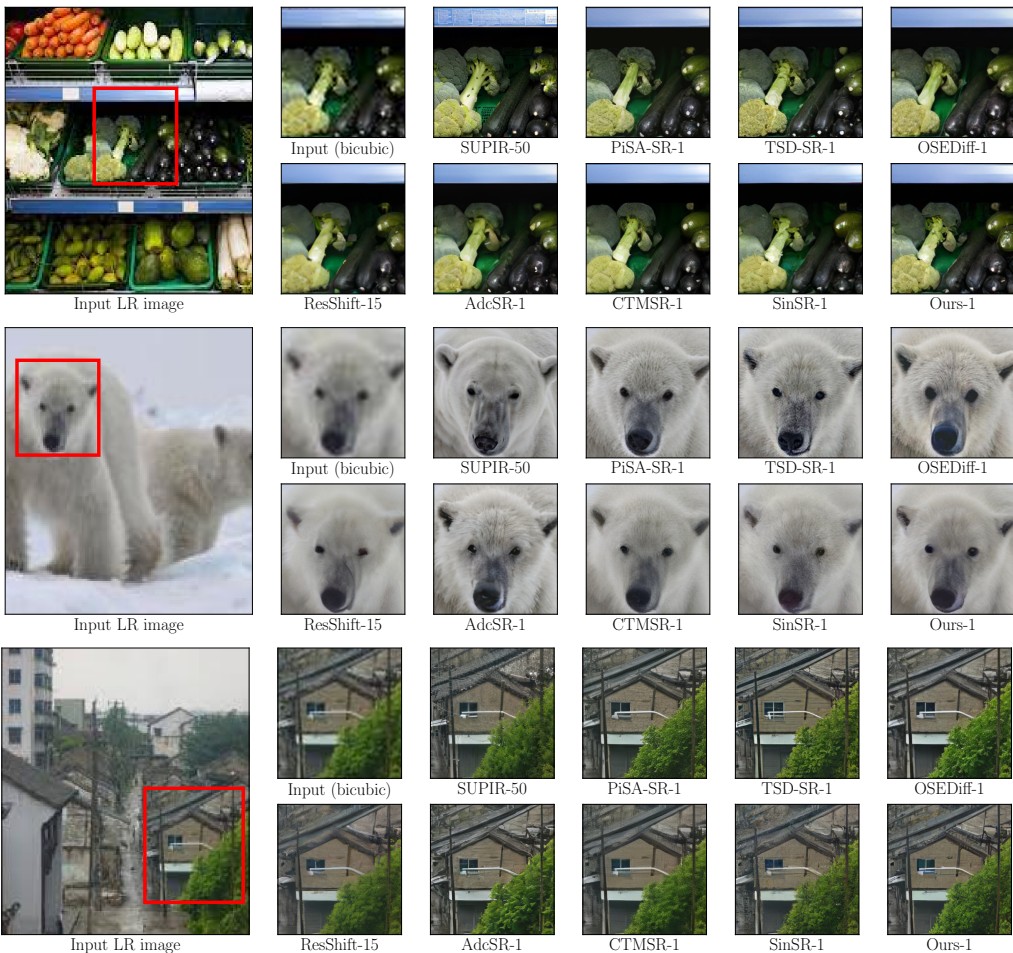

Figure 3: Comparison on Real-ISR (RealSet65, (Yue et al., 2023)) for diffusion SR models. Bottom images: ResShift, AdcSR, CTMSR, SinSR and the proposed RSD. Top images: bicubic LR, SUPIR, PiSA-SR, TSD-SR and OSEDiff. Please zoom in ×5 times for a better view.

the setup of ResShift to be consistent with our teacher model and SinSR. We show our enhancements compared to the current SOTA ResShift distillation method (SinSR), and we also implement VSD-based method applied to the ResShift setup, called **ResShift-VSD** (see Appendix A.1); **(2)** to show that RSD achieves competitive perceptual performance with recent T2I-based SR methods such as OSEDiff and SUPIR while having much smaller computational training and inference resources with better fidelity. These goals are supported by evaluations using the SinSR and OSEDiff setups. We present two types of models: RSD (**Ours**, distill only), where we used only distillation loss during training, and RSD (**Ours**), where we used additional losses (§3.4). Appendix C provides all relevant experimental details. For a fair comparison with the SOTA models, we also provide results of very recent diffusion SR models, including CTMSR, AdcSR, PiSA-SR, and TSD-SR.

Table 1: Results on real-world datasets. The best and second best results are highlighted in **bold** and underline.

| Methods | T2I prior | NFE | RealSR | | | | | RealSet65 | |
|---|---|---|---|---|---|---|---|---|---|
| | | | PSNR↑ | SSIM↑ | LPIPS↓ | CLIPIQA↑ | MUSIQ↑ | CLIPIQA↑ | MUSIQ↑ |
| SUPIR | | 50 | 24.38 | 0.698 | 0.331 | 0.5449 | 63.676 | 0.6133 | 66.460 |
| OSEDiff | | 1 | 25.25 | 0.737 | 0.299 | 0.6772 | 67.602 | 0.6836 | 68.853 |
| AdcSR | yes, | 1 | 25.63 | 0.735 | 0.300 | 0.7033 | 67.550 | 0.7044 | 69.185 |
| PiSA-SR | > 450M params | 1 | 25.59 | 0.750 | **0.271** | 0.6678 | 67.993 | 0.7062 | 70.208 |
| TSD-SR | | 1 | 24.88 | 0.723 | 0.281 | 0.7336 | **69.871** | 0.7263 | **70.958** |
| ResShift | | 15 | **26.49** | 0.754 | 0.360 | 0.5958 | 59.873 | 0.6537 | 61.330 |
| CTMSR | | 1 | 26.18 | **0.765** | 0.294 | 0.6449 | 64.796 | 0.6893 | 67.173 |
| SinSR (distill only) | no, | 1 | 26.14 | 0.732 | 0.357 | 0.6119 | 57.118 | 0.6822 | 61.267 |
| SinSR | < 200M params | 1 | 25.83 | 0.717 | 0.365 | 0.6887 | 61.582 | 0.7150 | 62.169 |
| ResShift-VSD (Appendix A.1) | | 1 | 23.96 | 0.616 | 0.466 | 0.7479 | 63.298 | **0.7606** | 66.701 |
| RSD (**Ours**, distill only) | | 1 | 24.92 | 0.696 | 0.355 | **0.7518** | 66.430 | 0.7534 | 68.383 |
| RSD (**Ours**) | | 1 | 25.91 | 0.754 | 0.273 | 0.7060 | 65.860 | 0.7267 | 69.172 |

Table 2: Results on ImageNet-Test. The best and second best results are highlighted in **bold** and underline.

| Methods | T2I prior | NFE | PSNR↑ | SSIM↑ | LPIPS↓ | CLIPIQA↑ | MUSIQ↑ | FID↓ |
|---|---|---|---|---|---|---|---|---|
| SUPIR | yes, > 450M params | 50 | 22.56 | 0.574 | 0.302 | **0.786** | 60.487 | 24.70 |
| OSEDiff | | 1 | 23.02 | 0.619 | 0.253 | 0.677 | 60.755 | 23.13 |
| AdcSR | | 1 | 22.99 | 0.615 | 0.252 | 0.711 | 63.218 | 34.61 |
| PiSA-SR | | 1 | 24.29 | 0.670 | 0.213 | 0.629 | 62.137 | **19.34** |
| TSD-SR | | 1 | 23.58 | 0.645 | 0.197 | 0.673 | **65.299** | 20.55 |
| ResShift | no, < 200M params | 15 | **25.01** | **0.677** | 0.231 | 0.592 | 53.660 | 30.34 |
| CTMSR | | 1 | 24.73 | 0.666 | 0.197 | 0.691 | 60.142 | 24.19 |
| SinSR (distill only) | | 1 | 24.69 | 0.664 | 0.222 | 0.607 | 53.316 | 32.13 |
| SinSR | | 1 | 24.56 | 0.657 | 0.221 | 0.611 | 53.357 | 25.85 |
| ResShift-VSD (Appendix A.1) | | 1 | 23.69 | 0.624 | 0.230 | 0.665 | 58.630 | 32.22 |
| RSD (**Ours**, distill only) | | 1 | 23.97 | 0.643 | 0.217 | 0.660 | 57.831 | 28.93 |
| RSD (**Ours**) | | 1 | 24.31 | 0.657 | **0.193** | 0.681 | 58.947 | 25.46 |

Table 3: Results on crops $512 \times 512$. The best and second best results are highlighted in **bold** and underline.

| Datasets | Methods | T2I prior | NFE | PSNR↑ | SSIM↑ | LPIPS↓ | DISTS↓ | NIQE↓ | MUSIQ↑ | MANIQA↑ | CLIPIQA↑ | FID↓ |
|---|---|---|---|---|---|---|---|---|---|---|---|---|
| DIV2K-Val | SUPIR | yes, > 450M params | 50 | 22.13 | 0.5280 | 0.3923 | 0.2314 | 5.6758 | 63.82 | 0.5933 | 0.7147 | 31.46 |
| | OSEDiff | | 1 | 23.72 | 0.6108 | 0.2941 | 0.1976 | 4.7097 | 67.97 | 0.6148 | 0.6683 | 26.32 |
| | AdcSR | | 1 | 23.74 | 0.6017 | 0.2853 | 0.1899 | 4.3579 | 68.00 | 0.6073 | 0.6764 | 25.52 |
| | PiSA-SR | | 1 | 23.87 | 0.6058 | 0.2823 | 0.1934 | 4.5565 | 69.68 | 0.6375 | 0.6928 | **25.09** |
| | TSD-SR | | 1 | 23.02 | 0.5808 | **0.2673** | **0.1821** | 4.3244 | **71.69** | 0.6192 | **0.7416** | 29.16 |
| | ResShift | no, < 200M params | 15 | 24.65 | 0.6181 | 0.3349 | 0.2213 | 6.8212 | 61.09 | 0.5454 | 0.6071 | 36.11 |
| | SinSR | | 1 | 24.41 | 0.6018 | 0.3240 | 0.2066 | 6.0159 | 62.82 | 0.5386 | 0.6471 | 35.57 |
| | CTMSR | | 1 | **24.88** | **0.6265** | 0.3026 | 0.2040 | 5.1146 | 65.62 | 0.5165 | 0.6601 | 34.15 |
| | RSD (**Ours**) | | 1 | 23.91 | 0.6042 | 0.2857 | 0.1940 | 5.1987 | 68.05 | 0.5937 | 0.6967 | 34.84 |
| DrealSR | SUPIR | yes, > 450M params | 50 | 24.93 | 0.6360 | 0.4263 | 0.2823 | 7.4336 | 59.39 | 0.5537 | 0.6799 | 164.86 |
| | OSEDiff | | 1 | 27.92 | **0.7835** | 0.2968 | 0.2165 | 6.4902 | 64.65 | 0.5899 | 0.6963 | 135.30 |
| | AdcSR | | 1 | 28.10 | 0.7726 | 0.3046 | 0.2200 | 6.4467 | 66.27 | 0.5916 | 0.7049 | 134.05 |
| | PiSA-SR | | 1 | 28.32 | 0.7804 | **0.2960** | 0.2169 | 6.1766 | 66.11 | **0.6161** | 0.6968 | **130.61** |
| | TSD-SR | | 1 | 27.77 | 0.7559 | 0.2967 | **0.2136** | 5.9131 | **66.62** | 0.5874 | **0.7343** | 134.98 |
| | ResShift | no, < 200M params | 15 | 28.46 | 0.7673 | 0.4006 | 0.2656 | 8.1249 | 50.60 | 0.4586 | 0.5342 | 172.26 |
| | SinSR | | 1 | 28.36 | 0.7515 | 0.3665 | 0.2485 | 6.9907 | 55.33 | 0.4884 | 0.6383 | 170.57 |
| | CTMSR | | 1 | **28.65** | 0.7834 | 0.3238 | 0.2358 | 6.1828 | 59.78 | 0.4861 | 0.6497 | 163.63 |
| | RSD (**Ours**) | | 1 | 27.40 | 0.7559 | 0.3042 | 0.2343 | 6.2577 | 62.03 | 0.5625 | 0.7019 | 167.47 |
| RealSR | SUPIR | yes, > 450M params | 50 | 23.61 | 0.6606 | 0.3589 | 0.2492 | 5.8877 | 63.21 | 0.5895 | 0.6709 | 128.35 |
| | OSEDiff | | 1 | 25.15 | 0.7341 | 0.2921 | 0.2128 | 5.6476 | 69.09 | 0.6326 | 0.6693 | 123.49 |
| | AdcSR | | 1 | 25.47 | 0.7301 | 0.2885 | 0.2128 | 5.3477 | 69.90 | 0.6353 | 0.6730 | 118.41 |
| | PiSA-SR | | 1 | 25.50 | 0.7418 | **0.2672** | **0.2044** | 5.5046 | 70.15 | **0.6551** | 0.6696 | 124.09 |
| | TSD-SR | | 1 | 24.81 | 0.7172 | 0.2743 | 0.2105 | **5.1266** | **71.18** | 0.6346 | **0.7160** | **114.45** |
| | ResShift | no, < 200M params | 15 | **26.31** | 0.7421 | 0.3421 | 0.2498 | 7.2365 | 58.43 | 0.5285 | 0.5442 | 141.71 |
| | SinSR | | 1 | 26.28 | 0.7347 | 0.3188 | 0.2353 | 6.2872 | 60.80 | 0.5385 | 0.6122 | 135.93 |
| | CTMSR | | 1 | 25.98 | **0.7546** | 0.2897 | 0.2208 | 5.5546 | 64.26 | 0.5270 | 0.6318 | 135.35 |
| | RSD (**Ours**) | | 1 | 25.61 | 0.7420 | 0.2675 | 0.2205 | 5.7500 | 66.02 | 0.5930 | 0.6793 | 138.23 |

## 4.1 EXPERIMENTAL SETUP

**Training and evaluation details.** For a fair comparison, we follow the *training setup* of SinSR and ResShift, using $256 \times 256$ HR images randomly cropped from ImageNet (Deng et al., 2009) and generating LR images via the Real-ESRGAN degradations with $\times 4$ SR factor. We also adopt the ResShift teacher used in SinSR. *For the evaluation*, we follow two different protocols from SinSR and OSEDiff ($\times 4$ SR factor). Following SinSR, we use the following datasets: (1) for real-world degradations, we use full-size images from RealSR (Cai et al., 2019) and RealSet65 (Yue et al., 2023); (2) for synthetic degradations, we use ImageNet-Test (Yue et al., 2023). Following OSEDiff, we use test sets of HR crops $512 \times 512$ from StableSR (Wang et al., 2024a), including synthetic DIV2K-Val (Agustsson & Timofte, 2017) and real-world pairs from RealSR and DRealSR (Wei et al., 2020).

**Compared methods.** We follow two distinct experimental setups with different baseline comparisons. Following (Wang et al., 2024b, Table 1 and Table 2) and (Wu et al., 2024a, Table 1), we incorporate baselines and evaluation setups for real-world and synthetic data from SinSR and OSEDiff. We compare RSD against **diffusion-based** SR models in the main text: models with relatively small architectures (ResShift, SinSR, CTMSR), and recent T2I-based SR models - one-step OSEDiff and multistep SUPIR. We highlight that closely related to RSD models, such as ResShift, SinSR, and CTMSR, compared **only with early T2I-based SR models**, namely LDM and StableSR. In addition to OSEDiff and SUPIR, we extend the comparison of those methods with very recent SOTA one-step T2I-based SR methods, namely AdcSR, PiSA-SR and TSD-SR. In Appendix D,

we provide quantitative results of other baselines for evaluation setups of SinSR , including GANs for SR (ESRGAN, RealSR-JPEG, Real-ESRGAN, BSRGAN) and other methods (SwinIR, LDM, DASR (Liang et al., 2022b), InvSR, CCSR). For evaluation setups of OSEDiff , in Appendix D we compare RSD with multistep T2I-based (StableSR, DiffBIR, SeeSR, PASD), GAN-based SR methods (Real-ESRGAN, BSRGAN, LDL (Liang et al., 2022a), FeMASR (Chen et al., 2022)), and other recent one-step diffusion SR methods (InvSR, CCSR).

**Metrics.** Each setup employs different evaluation metrics, which we adopt from SinSR (Table 1 and Table 2) and OSEDiff (Table 1). For all evaluation setups from SinSR , we compute image-quality no-reference metrics CLIPIQA (Wang et al., 2023b) and MUSIQ (Ke et al., 2021) following SinSR. In the OSEDiff configuration, evaluation is conducted using fidelity metrics (PSNR, SSIM), full-reference perceptual metrics (LPIPS, DISTS (Ding et al., 2020)), and no-reference image-quality metrics, (NIQE (Zhang et al., 2015), MANIQA-PIPAL (Yang et al., 2022), MUSIQ, CLIPIQA). We calculate PSNR and SSIM on the Y channel of the YCbCr space following SinSR and OSEDiff . We also compute FID (Heusel et al., 2017) as a distribution alignment measure in Table 2 and Table 3, since ImageNet-Test and DIV2K-Val datasets have 3000 images while other datasets $\leq$ 100 images.

## 4.2 Experimental results

**Quantitative comparisons**. The key quantitative results are summarized in Table 1 , Table 2 , Table 3 and visualized in Figure 1. In these tables, the compared methods are grouped into two classes: models, which do not use the prior of pre-trained T2I models and have a relatively small architectures (ResShift, SinSR, CTMSR, RSD) and T2I-based models with heavy architectures (SUPIR, OSEDiff, AdcSR, PiSA-SR, TSD-SR). We make the following observations based on them. **(1)** RSD outperforms the teacher ResShift model and our closest competitor, SinSR, by a large margin for all **perceptual** metrics (LPIPS, CLIPIQA, MUSIQ, DISTS, NIQE, MANIQA) and **all test datasets** while training on the same data. RSD also has competitive fidelity metrics (PSNR and SSIM). Furthermore, RSD demonstrates comparable or even better results than the ResShift model distilled with the VSD loss, ResShift-VSD (Appendix A.1). CTMSR is the recent one-step diffusion SR method, which also used the same ImageNet during training and, therefore, can be fairly comparable to RSD. RSD achieves better perceptual metrics in **all real-world** datasets in most perceptual metrics (LPIPS, CLIPIQA, MUSIQ) with a notable improvement in MANIQA in Table 3. **(2)** Compared to T2I-based OSEDiff and SUPIR models on Real-ISR benchmarks, RSD achieves the best value of the latest image-quality CLIPIQA and top-1 or top-2 results in terms of MUSIQ. RSD has worse CLIPIQA than SUPIR for synthetic datasets but better than OSEDiff. We hypothesize the gap with SUPIR is due to its multistep nature, rich SDXL prior, and large proprietary dataset with high-quality images, which leads to better details and better preferences by no-reference metrics. However, this also leads to poor consistency with the LR image, as noticeable by PSNR, SSIM, and LPIPS metrics. We highlight that RSD, even with slightly worse MUSIQ, achieves much better fidelity metrics than OSEDiff and SUPIR for most setups while using a much smaller number of parameters and GPU memory, as shown in Table 4. Compared to very recent SOTA one-step diffusion T2I-based SR methods (AdcSR, PiSA-SR), RSD is able to achieve competitive results in perceptual (LPIPS, CLIPIQA) and fidelity (PSNR, SSIM) quality in Table 1 and 2. TSD-SR achieves better no-reference perceptual metrics (CLIPIQA, MUSIQ) compared to RSD, while RSD outperforms in reference-based fidelity (PSNR, SSIM) and perceptual metrics (LPIPS). **(3)** In Table 3 , we show that RSD achieves top-2 or top-1 perceptual metrics compared to OSEDiff and all DMs, which were trained on ImageNet. Compared to recent SOTA models PiSA-SR and TSD-SR, there is a gap between RSD in perceptual no-reference metrics. We highlight different training HR resolutions of RSD and OSEDiff - we used HR crops of the size $256 \times 256$ as in the teacher ResShift, while OSEDiff, AdcSR, PiSA-SR, and TSD-SR used HR crops of the size $512 \times 512$ for training on LSDIR (Li et al., 2023), which aligns with the crop size in Table 3 . Additional quantitative results are given in Appendix D. We also discuss the RSD results trained on $512 \times 512$ HR images in Appendix G.

**Qualitative comparisons**. We visually compare RSD with SinSR, OSEDiff, ResShift, SUPIR, CTMSR, AdcSR, PiSA-SR and TSD-SR on test images from RealSet65 in Figure 3. As illustrated in the top image, SUPIR tends to produce rich details that semantically do not correspond to the LR image. Please zoom in for excessive broccoli. ResShift, SinSR, and CTMSR produce conservative images, which may struggle from severely blurred details like the house's roof on the bottom image. OSEDiff may hallucinate excessive details, as can be seen for the bear's nose in Figure 3 and panda's

nose in Figure 1. RSD compromises between the good details of OSEDiff and SUPIR and the high fidelity of ResShift and SinSR. Additional visual results are given in Appendix L.

**Complexity comparisons**. We compare the complexity of competing diffusion-based SR models in Table 4, including NFE, inference time, total number of parameters, and maximum required GPU memory during inference. We also provide information about the training time and used GPUs of those methods according to the respective papers. All methods are tested on an NVIDIA A100 GPU with an HR image of size $256 \times 256$ following the training setup of ResShift, SinSR, and RSD. We observe that RSD and SinSR require at least $\times 5$ less GPU memory and have $\times 10$ fewer parameters than T2I-based models, which highlights the efficiency of these models in terms of computational budget. We also note the training efficiency of RSD compared to SinSR: RSD is a **simulation-free method**. Unlike SinSR, which runs the ResShift teacher model for all 15 steps (see Eq. 5 and 6 in (Wang et al., 2024b)), RSD only uses $f^*(x_t, y_0, t)$ at each step to update the generator (see Algorithm 1 in Appendix B). In Appendix C, we discuss that SinSR empirically converges roughly 3 times slower than RSD. Despite the good perceptual performance of very recent one-step diffusion T2I-based SR methods, such as AdcSR, PiSA-SR and TSD-SR, the computational budget of these models remains much bigger than for RSD. For example, we highlight the training efficiency of our RSD compared to TSD-SR: RSD has $\times 19$ faster training with $\times 2$ times less GPUs compared to TSD-SR. During inference, TSD-SR requires $\times 13$ more parameters and $\times 8$ more GPU memory than RSD. More discussion about performance-efficiency trade-off for RSD and other SOTA methods is given in Appendix E.

Table 4: Training and inference complexity. All methods are tested with an LR image of size $64 \times 64$ for SR factor $\times 4$, and the inference is done on an NVIDIA A100 GPU. The best values are highlighted in **bold**.

| Methods | SUPIR | OSEDiff | AdcSR | PiSA-SR | TSD-SR | ResShift | SinSR | CTMSR | RSD (**Ours**) |
|---|---|---|---|---|---|---|---|---|---|
| Inference Step (NFE) | 50 | **1** | **1** | **1** | **1** | 15 | **1** | **1** | **1** |
| Inference Time (s) | 17.704 | 0.075 | **0.024** | 0.089 | 0.074 | 0.643 | 0.060 | 0.059 | 0.059 |
| # Total Param (M) | 4801 | 1775 | 456 | 1290 | 2207 | 174 | 174 | **172** | 174 |
| Maximum GPU memory (MB) | 52535 | 3651 | 3940 | 4771 | 4611 | 1167 | 570 | 904 | **539** |
| Training time (hours / # GPU) | 240 / 64 A6000 | 24 / 4 A100 | >124 / 8 A100 | 5.5 / 4 A100 | 96 / 8 A100 | 110 / 1 A100 | 60 / 1 A100 | 58 / 4 A100 | **5 / 4 A100** |

### 4.3 ABLATION STUDY

**Multistep training.** We analyze the performance of our method under different timestep configurations in multistep training §3.3. As shown in Table 5, we compare various numbers of timesteps $N$ ranging from 1 to 15 with the maximum number matching that of ResShift; timesteps are evenly placed. Selecting $N = 4$ provides the optimal choice for the compromise between perceptual quality and distortion, which is known as perceptual-distortion trade-off (Blau & Michaeli, 2018).

Table 5: Impact of multistep training of our RSD on RealSR (Cai et al., 2019). The best and second best results are highlighted in **bold** and underline.

| $N$ | PSNR↑ | LPIPS↓ | CLIPIQA↑ | MUSIQ↑ |
|---|---|---|---|---|
| 1 | 24.82 | 0.4052 | 0.7444 | 64.290 |
| 2 | 24.77 | 0.3772 | **0.7523** | 65.760 |
| 4 | 24.92 | 0.3552 | 0.7518 | 66.430 |
| 8 | 25.63 | 0.3199 | 0.7286 | **66.445** |
| 15 | **25.91** | **0.2940** | 0.6857 | 65.689 |

Table 6: Effect of incorporating supervised losses on RealSR (Cai et al., 2019). The best and second best results are highlighted in **bold** and underline.

| Method | PSNR↑ | LPIPS↓ | CLIPIQA↑ | MUSIQ↑ |
|---|---|---|---|---|
| $\lambda_{1,2} = 0$ | 24.92 | 0.3552 | **0.7518** | 66.430 |
| $\lambda_1 \neq 0$ | **26.01** | **0.2708** | 0.7089 | 65.178 |
| $\lambda_2 \neq 0$ | 24.98 | 0.3064 | 0.6970 | **67.615** |
| **Ours** | 25.91 | 0.2726 | 0.7060 | 65.860 |

**Supervised losses.** Table 6 examines the impact of incorporating supervised losses, as discussed in §3.4. Our results show that adding these losses significantly enhances quality in PSNR, SSIM and in LPIPS while introducing compromised yet acceptable changes in no-reference metrics (CLIPIQA, MUSIQ). In all evaluations, we use full-size images with real-world degradations from RealSR.

We provide additional ablation studies on the number of updates for the fake model per student update, $K$, and visual results for Table 6 in Appendix F.

### 5 CONCLUSION AND FUTURE WORK

In this work, we propose RSD, a novel approach to distill the ResShift model into a student network with a single inference step. Our model is computationally efficient thanks to its ResShift framework, but remains constrained by its teacher capacity issue, as validated in Appendix G. A more advanced teacher, such as a T2I-based model, could improve performance and enable the application of our method at higher resolutions. We discuss the limitations of RSD, failure cases, and potential societal impact in Appendix J.

# 6 REPRODUCIBILITY STATEMENT

To support the reproducibility of the proposed RSD method, we provide the following:

1. **Source code**. We provide a reproducible training and validation anonymous code for the main results in Table 1 , Table 2 , Table 3 of the proposed RSD method in the supplementary material to the submission. The code is written in Python using the PyTorch framework (Paszke et al., 2019) based on (Wang et al., 2024b, GitHub SinSR source code):



`https://github.com/wyf0912/SinSR`



The source code is supported with **README** instructions for reproducibility.

2. **Pseudocode for algorithms**. We provide the pseudocode for the RSD and ResShift-VSD algorithms in Appendix B, Algorithms 1 and 2, respectively.

3. **Experimental details**. We provide all relevant experimental details, including training hyperparameters, training time, datasets, baselines, and metrics calculation in the Appendix C.

4. **Evaluation of baselines**. We provide details on the models used for the evaluation of the baselines in the Appendix D.

5. **Proofs and theoretical explanations**. We provide the proof of Proposition 3.1 and discusses the computational issues of the original problem (7) in Appendix K.

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

APPENDIX

We organize the structure of supplementary materials as follows:

1. Appendix A discusses relation of RSD to relevant methods that involve training an aulixiary "fake" model - variational score distillation (VSD (Yin et al., 2024b; Wu et al., 2024a; Wang et al., 2023c)), score identity distillation (SiD (Zhou et al., 2024)), Flow Generator Matching (FGM (Huang et al., 2024)), and inverse bridge matching distillation (IBMD (Gushchin et al., 2025)). Appendix A.1 includes the derivation of the variational score distillation for ResShift and its comparison with our RSD loss $\mathcal{L}_\theta$. Appendix A.2 discusses the relation of RSD to SiD and FGM. Appendix A.3 discusses the relation of RSD to IBMD and their quantitative comparison. In Appendix A.4, we also discuss the generalization of RSD method for other diffusion models.

2. Appendix B details the correspondence between propositions and their implementation with the pseudocode of RSD. We also present the pseudocode for ResShift-VSD, introduced in Appendix A

3. Appendix C consists of experimental details for the implementation of RSD and baselines.

4. Appendix D consists of full quantitative results including additional baselines and results on full-size DRealSR, RealLR200, and RealLQ250 which haven't been shown in the main text due to space limitations.

5. Appendix E provides comparison in performance and efficiency of RSD and state-of-the-art diffusion SR models: PiSA-SR, TSD-SR, InvSR, CCSR, CTMSR, and AdcSR.

6. Appendix F provides additional discussion of ablation studies on hyperparameter $K$, training stability of RSD, and supervised losses.

7. Appendix G provides the quantitative comparison between RSD, SinSR and ResShift when all these models are trained on HR cropped images with resolution $512 \times 512$ from the LSDIR dataset (Li et al., 2023), which follows the training setup of OSEDiff (Wu et al., 2024a).

8. Appendix H discusses the qualitative and quantitative comparison between RSD and AddSR (Xie et al., 2024).

9. Appendix I includes additional details of ResShift, which have not been shown in the main text due to space limitations.

10. Appendix J discusses the limitations of RSD, failure cases and potential societal impact.

11. Appendix K presents the proof of Proposition 3.1 and discusses the computational issues of the original problem (7).

12. Appendix L contains additional visual results for comparison between RSD and baselines.

13. Appendix M describes the details of the LLM usage in the paper.

## A   RELATION OF RSD TO VSD, SID, FGM AND IBMD

### A.1   DERIVATION OF VSD OBJECTIVE FOR RESSHIFT (RESSHIFT-VSD) AND COMPARATIVE ANALYSIS WITH OUR OBJECTIVE

In this section, we aim to: (1) derive the VSD loss in the ResShift framework to compare it with our distillation loss under the same experimental conditions (see Table 1 and Table 2 ); and (2) explain the main differences between our approach and the VSD loss. To achieve this, we consider a generator $G_\theta$ with parameters $\theta$ and seek an update rule for them. We use a fake ResShift model to solve the following problem:

$$\arg\min_f \sum_{t=1}^T w_t \mathbb{E}_{p_\theta(\widehat{x}_0, y_0, x_t)}\big[\|f(x_t, y_0, t) - \widehat{x}_0\|_2^2\big], \tag{12}$$

Since it is the optimization with MSE function, the solution is given by the conditional expectation:

$$f_{G_\theta}(x_t, y_0, t) = \mathbb{E}_{p_\theta(\widehat{x}_0 | y_0, x_t)}[\widehat{x}_0]. \tag{13}$$

**Notation.** Further we will use the following notation:

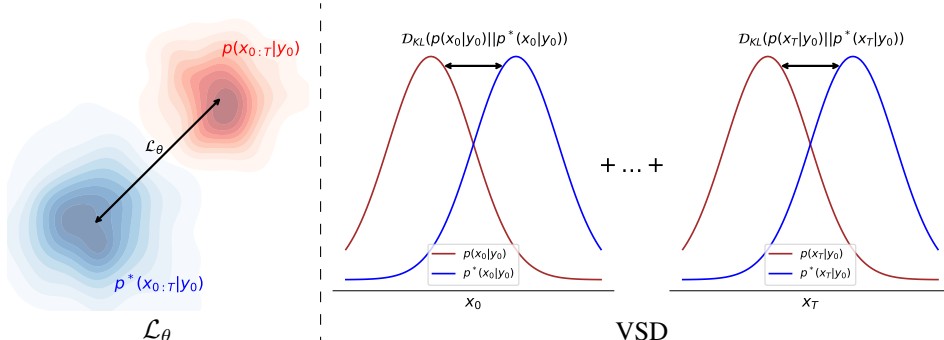

Figure 4: Illustration of the distinct distribution alignment strategies employed by the RSD $\mathcal{L}_\theta$ (**Ours**) and VSD loss functions. We denote by $p^*(x_{0:T}|y_0)$ reverse process of teacher ResShift model and by $p(x_{0:T}|y_0)$ reverse process of ResShift trained on generator $G_\theta$ data. The $\mathcal{L}_\theta$ loss enforces alignment of the joint distributions $p^*(x_{0:T}|y_0)$ and $p(x_{0:T}|y_0)$ across **all timesteps**, whereas the VSD loss aligns the marginal distributions at **each timestep** $t$ **simultaneously** between distributions of teacher ResShift and ResShift trained on generator $G_\theta$ data. For formal derivations, see Eqs. (20) and (16).

- $f^*$ – teacher ResShift.

- $x_{t_1:t_2} \overset{\text{def}}{=} (x_{t_1}, x_{t_1+1}, \ldots, x_{t_2})$ and $dx_{t_1:t_2} \overset{\text{def}}{=} \prod_{t=t_1}^{t_2} dx_t$ for any integer $t_1 < t_2$.

- The joint distribution across all timesteps is defined as follows:

$$p(x_{0:T}|y_0) \overset{\text{def}}{=} p(x_T|y_0) \prod_{t=1}^{T} p(x_{t-1}|x_t, y_0). \tag{14}$$

The transition probabilities are determined using Eq. (2) and Eq. (4):

$$p(x_{t-1}|x_t, y_0) = \mathcal{N}\left(x_{t-1} \Big| \frac{\eta_{t-1}}{\eta_t} x_t + \frac{\alpha_t}{\eta_t} f_{G_\theta}(x_t, y_0, t), \kappa^2 \frac{\eta_{t-1}}{\eta_t} \alpha_t \mathbf{I}\right). \tag{15}$$

In the same way we define $p^*(x_{0:T}|y_0) \overset{\text{def}}{=} p^*(x_T|y_0) \prod_{t=1}^{T} p^*(x_{t-1}|x_t, y_0)$, where the transition probabilities are determined using $f^*$.

- $p^*(x_t|y_0) \overset{\text{def}}{=} \int p^*(x_{0:T}|y_0) dx_{0:t-1} dx_{t+1:T}$ and $p(x_t|y_0) \overset{\text{def}}{=} \int p(x_{0:T}|y_0) dx_{0:t-1} dx_{t+1:T}$ are marginal distributions.

**Derivation of VSD loss for ResShift (ResShift-VSD).** Initially, main objective of VSD loss (Yin et al., 2024b; Wu et al., 2024a; Wang et al., 2023c) is:

$$\mathcal{L}_{\text{VSD}} = \mathbb{E}_{p(y_0)}\Big[ \sum_{t=1}^{T} w_t \mathcal{D}_{\text{KL}}\Big(p(x_t|y_0)||p^*(x_t|y_0)\Big)\Big]. \tag{16}$$

We can get another expression for this loss using reparametrization based on the Eq. (27):

$$\mathcal{L}_{\text{VSD}} = \sum_{t=1}^{T} w_t \mathbb{E}_{p(y_0)}\Big[ \mathcal{D}_{\text{KL}}\Big(p(x_t|y_0)||p^*(x_t|y_0)\Big)\Big] = \sum_{t=1}^{T} w_t \mathbb{E}_{p(y_0)p(x_t|y_0)} \log \frac{p(x_t|y_0)}{p^*(x_t|y_0)} =$$

$$\sum_{t=1}^{T} w_t \mathbb{E}_{p(y_0)} \underset{\substack{x_t=(1-\eta_t)\widehat{x}_0+\eta_t y_0+\kappa^2\eta_t\epsilon' \\ \widehat{x}_0=G_\theta(y_0,\epsilon) \\ \epsilon',\epsilon\sim\mathcal{N}(0;\mathbf{I})}}{\mathbb{E}} \log \frac{p(x_t|y_0)}{p^*(x_t|y_0)} \tag{17}$$

Initially, this loss is intractable because it requires computing probability densities. However, taking the gradient facilitates its computation:

$$\nabla_\theta \mathcal{L}_{\text{VSD}} =$$

$$-\sum_{t=1}^T w_t \mathbb{E}_{p(y_0)} \mathop{\mathbb{E}}_{\substack{x_t=(1-\eta_t)\widehat{x}_0+\eta_t y_0+\kappa^2\eta_t\epsilon' \\ \widehat{x}_0=G_\theta(y_0,\epsilon) \\ \epsilon',\epsilon\sim\mathcal{N}(0;\mathbf{I})}} \left[ (\nabla_{x_t} \log p^*(x_t|y_0) - \nabla_{x_t} \log p(x_t|y_0)) \frac{dx_t}{d\theta} \right] =$$

$$-\sum_{t=1}^T w_t \mathbb{E}_{p(y_0)} \mathop{\mathbb{E}}_{\substack{x_t=(1-\eta_t)\widehat{x}_0+\eta_t y_0+\kappa^2\eta_t\epsilon' \\ \widehat{x}_0=G_\theta(y_0,\epsilon) \\ \epsilon',\epsilon\sim\mathcal{N}(0;\mathbf{I})}} \left[ (\nabla_{x_t} \log p^*(x_t|y_0) - \nabla_{x_t} \log p(x_t|y_0)) \frac{dx_t}{d\widehat{x}_0}\frac{d\widehat{x}_0}{d\theta} \right] =$$

$$-\sum_{t=1}^T w'_t \mathbb{E}_{p(y_0)} \mathop{\mathbb{E}}_{\substack{x_t=(1-\eta_t)\widehat{x}_0+\eta_t y_0+\kappa^2\eta_t\epsilon' \\ \widehat{x}_0=G_\theta(y_0,\epsilon) \\ \epsilon',\epsilon\sim\mathcal{N}(0;\mathbf{I})}} \left[ (\nabla_{x_t} \log p^*(x_t|y_0) - \nabla_{x_t} \log p(x_t|y_0)) \frac{d\widehat{x}_0}{d\theta} \right], \quad (18)$$

where $w'_t \stackrel{\text{def}}{=} w_t \frac{dx_t}{d\widehat{x}_0} = w_t(1-\eta_t)$.

The expression $\nabla_{x_t} \log p(x_t|y_0)$ can be utilized as follows (Zheng et al., 2024):

$$\nabla_{x_t} \log p(x_t|y_0) = \frac{\nabla_{x_t} p(x_t|y_0)}{p(x_t|y_0)} = \frac{\nabla_{x_t} \int q(x_t|y_0,x_0)p(x_0|y_0)dx_0}{p(x_t|y_0)} =$$

$$\frac{\int p(x_0|y_0)\nabla_{x_t} q(x_t|y_0,x_0)dx_0}{p(x_t|y_0)} = \frac{\int p(x_0|y_0)q(x_t|y_0,x_0)\nabla_{x_t}\log q(x_t|y_0,x_0)dx_0}{p(x_t|y_0)} =$$

$$\int \frac{p(x_0|y_0)q(x_t|y_0,x_0)}{p(x_t|y_0)}\nabla_{x_t}\log q(x_t|y_0,x_0)dx_0 = \int p(x_0|x_t,y_0)\nabla_{x_t}\log q(x_t|y_0,x_0)dx_0$$

$$= \mathop{\mathbb{E}}_{p(x_0|x_t,y_0)}\left[ \nabla_{x_t}\log q(x_t|y_0,x_0) \right]$$

Since $q(x_t|y_0,x_0) = \mathcal{N}(x_t|x_0+\eta_t e_0, \kappa^2\eta_t\mathbf{I})$ (See Eq. (27)), we get:

$$\nabla_{x_t} \log p(x_t|y_0) = -\mathop{\mathbb{E}}_{p(x_0|x_t,t,y_0)}\left[ \frac{x_t - \eta_t y_0 - (1-\eta_t)x_0}{\kappa^2\eta_t} \right],$$

which leads to:

$$\nabla_\theta \mathcal{L}_{\text{VSD}} = -\sum_{t=1}^T w''_t \mathbb{E}_{p(y_0)} \mathop{\mathbb{E}}_{\substack{x_t=(1-\eta_t)\widehat{x}_0+\eta_t y_0+\kappa^2\eta_t\epsilon' \\ \widehat{x}_0=G_\theta(y_0,\epsilon) \\ \epsilon',\epsilon\sim\mathcal{N}(0;\mathbf{I})}} \left[ (f^*(x_t,y_0,t) - f_{G_\theta}(x_t,y_0,t)) \frac{d\widehat{x}_0}{d\theta} \right] \quad (19)$$

where $w''_t \stackrel{\text{def}}{=} w'_t \frac{1-\eta_t}{\kappa^2\eta_t}$. As a result this loss can be implemented to match the gradients with $\nabla_\theta \mathcal{L}_{\text{VSD}}$ (see Algorithm 2). We call this model ResShift-VSD.

**Reformulation of our $\mathcal{L}_\theta$ loss.** We can express our RSD loss function as:

$$\mathcal{L}_\theta = \mathbb{E}_{p(y_0)}\left[ \mathcal{D}_{\text{KL}}\left( p(x_{0:T}|y_0) \,\|\, p^*(x_{0:T}|y_0) \right) \right], \quad (20)$$

Recalling that the joint probability distribution can be factorized (see Eq. 14), the above loss can be decomposed as:

$$\mathcal{L}_\theta = \mathbb{E}_{p(y_0)}\left[ \mathcal{D}_{\text{KL}}\left( p(x_{0:T}|y_0) \,\|\, p^*(x_{0:T}|y_0) \right) \right] = \mathbb{E}_{p(y_0)}\Big[ \underbrace{\mathcal{D}_{\text{KL}}\left( p(x_T|y_0) \,\|\, p^*(x_T|y_0) \right)}_{=0 \text{ since } p(x_T|y_0)=p^*(x_T|y_0) \text{ from Eq. (27)}} \Big] +$$

$$\mathbb{E}_{p(y_0)}\Big[ \sum_{t=1}^T \mathbb{E}_{p(x_t|y_0)} \mathcal{D}_{\text{KL}}\left( p(x_{t-1}|x_t,y_0) \,\|\, p^*(x_{t-1}|x_t,y_0) \right) \Big].$$

By applying Eq. (15), the KL divergence inside the expectation reduces to the KL divergence between Gaussian distributions, which can be computed in closed form. Consequently, we obtain:

$$\mathcal{L}_\theta = \sum_{t=1}^{T} \mathbb{E}_{p(y_0)} \mathbb{E}_{p(x_t|y_0)} \mathcal{D}_{\mathrm{KL}}\Big( p(x_{t-1}|x_t, y_0) \,\|\, p^*(x_{t-1}|x_t, y_0) \Big)$$

$$= \sum_{t=1}^{T} \mathbb{E}_{p(y_0)} \mathbb{E}_{p(x_t|y_0)} \underbrace{\frac{1}{2\kappa^2} \frac{\alpha_t}{\eta_t \eta_{t-1}}}_{\stackrel{\mathrm{def}}{=} w_t} \|f_{G_\theta}(x_t, y_0, t) - f^*(x_t, y_0, t)\|_2^2$$

$$= \sum_{t=1}^{T} w_t \mathbb{E}_{p(y_0)} \mathbb{E}_{p(x_t|y_0)} \|f_{G_\theta}(x_t, y_0, t) - f^*(x_t, y_0, t)\|_2^2 .$$

Since the distribution $p(x_t|y_0)$ is generally intractable, we instead use the tractable distribution $q(x_t|\widehat{x}_0, y_0)$, which is known to satisfy $q(x_t|\widehat{x}_0, y_0) = p(x_t|y_0)$. Thus, we have:

$$\mathcal{L}_\theta = \sum_{t=1}^{T} w_t \mathbb{E}_{p(y_0)} \mathbb{E}_{q(x_t|\widehat{x}_0, y_0)} \|f_{G_\theta}(x_t, y_0, t) - f^*(x_t, y_0, t)\|_2^2 .$$

Noting that the integrand is independent of $\widehat{x}_0$, so we can use $p_\theta(\widehat{x}_0, y_0)$ instead of $p(y_0)$, since $\int p_\theta(\widehat{x}_0|y_0) d\widehat{x}_0 = 1$, and therefore we obtain the following:

$$\mathcal{L}_\theta = \sum_{t=1}^{T} w_t \mathbb{E}_{q(x_t|\widehat{x}_0, y_0)\, p_\theta(\widehat{x}_0, y_0)} \|f_{G_\theta}(x_t, y_0, t) - f^*(x_t, y_0, t)\|_2^2 .$$

Finally, recognizing that the joint distribution $p_\theta(\widehat{x}_0, y_0, x_t)$ is defined as

$$p_\theta(\widehat{x}_0, y_0, x_t) \stackrel{\mathrm{def}}{=} q(x_t|\widehat{x}_0, y_0)\, p_\theta(\widehat{x}_0, y_0),$$

we arrive at the final form of the RSD loss:

$$\mathcal{L}_\theta = \sum_{t=1}^{T} w_t \mathbb{E}_{p_\theta(\widehat{x}_0, y_0, x_t)} \|f_{G_\theta}(x_t, y_0, t) - f^*(x_t, y_0, t)\|_2^2 .$$

This derivation demonstrates that the loss function in Eq. (20) reconstructs the initial objective presented in Eq. (7).

**Conceptual comparison of VSD and our RSD $\mathcal{L}_\theta$ losses.** The key difference between the VSD and $\mathcal{L}_\theta$ losses lies in how they match distributions. For a more clear intuitive explanation one can see formulations of losses with $\mathcal{D}_{\mathrm{KL}}$ for VSD (Eq. (16)) and $\mathcal{L}_\theta$ (Eq. (20)). The VSD loss aligns the marginal distributions at each timestep $t$ between the teacher's and fake's distributions. In contrast, the $\mathcal{L}_\theta$ loss matches the joint distribution across all timesteps. This difference is illustrated in Figure 4, where the $\mathcal{L}_\theta$ loss enforces joint distribution alignment, while the VSD loss aligns marginal distributions separately and then sums them.

RSD loss is superior to VSD loss for SR problem because it aligns the joint distribution across all timesteps, ensuring a more holistic match between teacher and student models. Unlike VSD, which aligns marginal distributions at each timestep separately, RSD captures temporal dependencies more effectively. This joint alignment is particularly beneficial for SR tasks, where maintaining consistency and accuracy across all image details and features is crucial for high-quality resolution. The RSD loss, by considering the entire distribution over multiple timesteps, leads to more precise and stable SR performance.

**Computational analysis of VSD and $\mathcal{L}_\theta$ losses.** As was shown in Proposition 3.1, our loss can be evaluated via:

$$\mathcal{L}_\theta = -\min_\phi \Big\{ \sum_{t=1}^{T} w_t \mathbb{E}_{p_\theta(\widehat{x}_0, x_t, y_0)} \Big[ \|f_\phi(x_t, y_0, t)\|_2^2 - \|f^*(x_t, y_0, t)\|_2^2 +$$

$$2\langle f^*(x_t, y_0, t) - f_\phi(x_t, y_0, t), \widehat{x}_0 \rangle \Big] \Big\}.$$

Using Eq. (13) we can rewrite it and make reparameterization:

$$
\mathcal{L}_\theta = -\sum_{t=1}^{T} w_t \mathbb{E}_{p_\theta(\widehat{x}_0, x_t, y_0)} \Big[ \|f_{G_\theta}(x_t, y_0, t)\|_2^2 - \|f^*(x_t, y_0, t)\|_2^2 +
$$

$$
2\langle f^*(x_t, y_0, t) - f_{G_\theta}(x_t, y_0, t), \widehat{x}_0 \rangle \Big] =
$$

$$
-\sum_{t=1}^{T} w_t \mathop{\mathbb{E}}_{\substack{x_t = (1-\eta_t)\widehat{x}_0 + \eta_t y_0 + \kappa^2 \eta_t \epsilon' \\ \widehat{x}_0 = G_\theta(y_0, \epsilon) \\ \epsilon', \epsilon \sim \mathcal{N}(0; \mathbf{I})}} \Big[ \|f_{G_\theta}(x_t, y_0, t)\|_2^2 - \|f^*(x_t, y_0, t)\|_2^2 +
$$

$$
2\langle f^*(x_t, y_0, t) - f_{G_\theta}(x_t, y_0, t), \widehat{x}_0 \rangle \Big].
$$

To compare it with VSD loss, we can take the gradient of $\mathcal{L}_\theta$ loss and get:

$$
\frac{d\mathcal{L}_\theta}{d\theta} = -\sum_{t=1}^{T} w_t \mathop{\mathbb{E}}_{\substack{x_t = (1-\eta_t)\widehat{x}_0 + \eta_t y_0 + \kappa^2 \eta_t \epsilon' \\ \widehat{x}_0 = G_\theta(y_0, \epsilon) \\ \epsilon', \epsilon \sim \mathcal{N}(0; \mathbf{I})}} \Big[ \frac{d\|f_{G_\theta}(x_t, y_0, t)\|_2^2}{d\theta} - \frac{d\|f^*(x_t, y_0, t)\|_2^2}{d\theta} +
$$

$$
2\langle \frac{df^*(x_t, y_0, t)}{d\theta} - \frac{df_{G_\theta}(x_t, y_0, t)}{d\theta}, \widehat{x}_0 \rangle + 2\langle f^*(x_t, y_0, t) - f_{G_\theta}(x_t, y_0, t), \frac{d\widehat{x}_0}{d\theta} \rangle \Big] =
$$

$$
-\sum_{t=1}^{T} w_t \mathop{\mathbb{E}}_{\substack{x_t = (1-\eta_t)\widehat{x}_0 + \eta_t y_0 + \kappa^2 \eta_t \epsilon' \\ \widehat{x}_0 = G_\theta(y_0, \epsilon) \\ \epsilon', \epsilon \sim \mathcal{N}(0; \mathbf{I})}} \Big[ \frac{d\|f_{G_\theta}(x_t, y_0, t)\|_2^2}{d\theta} - \frac{d\|f^*(x_t, y_0, t)\|_2^2}{d\theta} +
$$

$$
2\langle \frac{df^*(x_t, y_0, t)}{d\theta} - \frac{df_{G_\theta}(x_t, y_0, t)}{d\theta}, \widehat{x}_0 \rangle \Big]
$$

$$
\underbrace{-\sum_{t=1}^{T} w_t \mathop{\mathbb{E}}_{\substack{x_t = (1-\eta_t)\widehat{x}_0 + \eta_t y_0 + \kappa^2 \eta_t \epsilon' \\ \widehat{x}_0 = G_\theta(y_0, \epsilon) \\ \epsilon', \epsilon \sim \mathcal{N}(0; \mathbf{I})}} \Big[ 2\langle f^*(x_t, y_0, t) - f_{G_\theta}(x_t, y_0, t), \frac{d\widehat{x}_0}{d\theta} \rangle \Big]}_{=2 \cdot \nabla_\theta \mathcal{L}_{\text{VSD}} \text{ up to weighting term } w_t \text{ (see Eq.(19))}}
$$

Consequently, the gradients of the our RSD $\mathcal{L}_\theta$ loss function encompass those of the VSD loss, scaled by a constant factor of 2 and modulated by the time-dependent weighting term $w_t$. These scaling factors do not affect the optimal solution of the loss. However, $\mathcal{L}_\theta$ additionally incorporates gradient contributions from both the teacher and the fake models. To reduce $\mathcal{L}_\theta$ to the standard VSD formulation, the application of a stop-gradient operator is required to suppress the influence of these auxiliary gradient terms. For a detailed implementation, refer to Algorithm 2.

## A.2 RELATION OF RSD TO SID AND FGM

The loss function $\mathcal{L}_\theta$ can be reformulated as follows:

$$\mathcal{L}_\theta = \min_\phi \Big\{ \sum_{t=1}^{T} w_t \mathbb{E}_{p_\theta(\widehat{x}_0, y_0, x_t)} \Big( \|f^*(x_t, y_0, t)\|_2^2 - \|f_\phi(x_t, y_0, t)\|_2^2 +$$

$$2\langle f_\phi(x_t, y_0, t) - f^*(x_t, y_0, t), \widehat{x}_0 \rangle \Big) \Big\}$$

$$= \min_\phi \Big\{ \sum_{t=1}^{T} w_t \mathbb{E}_{p_\theta(\widehat{x}_0, y_0, x_t)} \Big( \|f^*(x_t, y_0, t)\|_2^2 - \|f_\phi(x_t, y_0, t)\|_2^2 +$$

$$2\langle f_\phi(x_t, y_0, t) - f^*(x_t, y_0, t), \widehat{x}_0 \rangle {\color{red}\pm 2\langle f^*(x_t, y_0, t), f_\phi(x_t, y_0, t)\rangle \pm \|f_\phi(x_t, y_0, t)\|_2^2} \Big) \Big\}$$

$$= \min_\phi \Big\{ \sum_{t=1}^{T} w_t \mathbb{E}_{p_\theta(\widehat{x}_0, y_0, x_t)} \Big( \|f^*(x_t, y_0, t) - f_\phi(x_t, y_0, t)\|_2^2 +$$

$$2\langle f^*(x_t, y_0, t) - f_\phi(x_t, y_0, t), f_\phi(x_t, y_0, t)\rangle + 2\langle f_\phi(x_t, y_0, t) - f^*(x_t, y_0, t), \widehat{x}_0 \rangle \Big) \Big\}$$

$$= \min_\phi \Big\{ \sum_{t=1}^{T} w_t \mathbb{E}_{p_\theta(\widehat{x}_0, y_0, x_t)} \Big( \|f^*(x_t, y_0, t) - f_\phi(x_t, y_0, t)\|_2^2 +$$

$$2\langle f^*(x_t, y_0, t) - f_\phi(x_t, y_0, t), f_\phi(x_t, y_0, t) - \widehat{x}_0 \rangle \Big) \Big\}$$

One can see that our objective can be reformulated in a manner closely resembling the formulations employed in SiD (Zhou et al., 2024, Eq. 23 with $\alpha = 0.5$) and FGM (Huang et al., 2024, Eqs. 4.11–4.12) with up to time weighting $w_t$. However, in both SiD (where $\alpha = 1.0, 1.2$ were used in the experiments) and FGM, the authors either omitted the quadratic term or assigned it a negative coefficient in image-based experiments due to the instability it introduced. In contrast to these approaches, we retain the complete original loss formulation as prescribed by theory, without discarding or modifying any of its components.

Furthermore, it is important to emphasize that SiD and FGM were primarily developed for image generation tasks, whereas our proposed RSD framework is specifically tailored for image restoration, with a focus on reconstructing high-resolution images from their low-resolution counterparts. To this end, we adopt a dedicated ResShift architecture that integrates both VAE and U-Net components, along with a diffusion process specifically designed for the super-resolution task. Additionally, we incorporate supervised loss terms tailored to the super-resolution objective (see Section 3.5). These task-specific design choices stand in contrast to SiD and FGM, which lack such adaptations for image restoration scenarios.

## A.3 RELATION OF RSD TO IBMD

**Conceptual comparison**. The loss function $\mathcal{L}_\theta$ can be equivalently reformulated in the following manner:

$$\mathcal{L}_\theta = \min_\phi \Big\{ \sum_{t=1}^{T} w_t \mathbb{E}_{p_\theta(\widehat{x}_0, y_0, x_t)} \Big( \|f^*(x_t, y_0, t)\|_2^2 - \|f_\phi(x_t, y_0, t)\|_2^2 +$$

$$2\langle f_\phi(x_t, y_0, t) - f^*(x_t, y_0, t), \widehat{x}_0 \rangle \Big) \Big\}$$

$$= \min_\phi \Big\{ \sum_{t=1}^{T} w_t \mathbb{E}_{p_\theta(\widehat{x}_0, y_0, x_t)} \Big( \|f^*(x_t, y_0, t)\|_2^2 - \|f_\phi(x_t, y_0, t)\|_2^2 +$$

$$2\langle f_\phi(x_t, y_0, t), \widehat{x}_0 \rangle - 2\langle f^*(x_t, y_0, t), \widehat{x}_0 \rangle {\color{red}\pm \|\widehat{x}_0\|_2^2} \Big) \Big\}$$

$$= \min_\phi \Big\{ \sum_{t=1}^{T} w_t \mathbb{E}_{p_\theta(\widehat{x}_0, y_0, x_t)} \Big( \|f^*(x_t, y_0, t) - \widehat{x}_0\|_2^2 - \|f_\phi(x_t, y_0, t) - \widehat{x}_0\|_2^2 \Big) \Big\} \quad (21)$$

It is worth noting that our RSD loss, denoted by $\mathcal{L}_\theta$, can be interpreted as a discrete variant of the inverse bridge matching distillation (IBMD) loss (Gushchin et al., 2025, Eq. 10), originally proposed for conditional Bridge Matching models. From a theoretical perspective, one of our main contributions is the development of a discretized form of the IBMD-conditional loss, which can offer practical benefits for tasks such as super-resolution.

While the IBMD framework has been applied to a broad range of problems, including image restoration, its experimental setup relied on relatively simplistic degradation processes, such as bicubic and pool. In contrast, we have tailored our objective specifically for image restoration by integrating it into the ResShift paradigm, incorporating additional supervised losses explicitly designed for real-world super-resolution (see Section 3.5), and utilizing a more challenging and realistic degradation model based on Real-ESRGAN. We also note that ResShift and RSD use VAE and a diffusion process in the latent space, while IBMD operates in the pixel space, which makes RSD more efficient and scalable for handling images of varying resolutions due to the reduced computational complexity and memory requirements in the latent domain. Another relevant difference between implementations of IBMD and ResShift for the SR problem is the bigger architecture of IBMD. IBMD for super-resolution uses the ADM architecture (Dhariwal & Nichol, 2021) following the I2SB (Liu et al., 2023b) with 552M parameters, while RSD follows the ResShift architecture with 174M parameters.

**Quantitative comparison**. Thus, in addition to our theoretical contribution, we extend the implementation of the IBMD loss to more severe and practically relevant degradation settings. These task-specific modifications differentiate our approach from the original IBMD formulation, which does not account for such adaptations in the context of real-world image restoration scenarios. To support this claim quantitatively and qualitatively, we conducted the following numerical experiments to show that both the teacher model of I2SB and the distilled student model of IBMD do not provide sufficient perceptual quality in Real-ISR problems compared to ResShift and RSD, respectively.

**Step 1: training the I2SB teacher using Real-ESRGAN degradations**. For a fair comparison with ResShift, we trained an I2SB model on ImageNet using the Real-ESRGAN degradations following the training setup of ResShift and RSD detailed in Section 4.1. The model was trained using the same hyperparameters as the original I2SB model trained on bicubic degradations with 4000 iterations. We used the official I2SB implementation published in the respective GitHub repository, which is provided by the I2SB authors:



`https://github.com/NVlabs/I2SB`



**Step 2: distillation of I2SB with IBMD**. We asked the authors of IBMD to provide the training and evaluation code of the IBMD method, which originally distilled I2SB for bicubic degradations. We adapt the provided implementation with the replacement of Real-ESRGAN degradations instead of original bicubic degradations and use the same hyperparameters, which were used for the training of IBMD model for bicubic degradations (first line in Table 7 of IBMD). We distilled the trained I2SB teacher with Real-ESRGAN degradations using IBMD method into one-step student model with 1500 gradient updates for the student model.

For a fair comparison with ResShift and RSD, we evaluated the trained teacher I2SB with NFE = 15 and 1 and the trained IBMD student with NFE = 1, which follows the inference NFE of ResShift and RSD, respectively. We report on the results of their evaluation on the ImageNet-Test dataset, which follows the RSD evaluation setup reported in Table 2, and compare them with ResShift, RSD with supervised losses (RSD (**Ours**)), and RSD with only distillation loss (RSD (**Ours**, distill only)) in Table 7. We also extend the complexity comparison in our Table 4 of RSD with IBMD by providing the training time of both methods in Table 8.

**Comparison between teachers, I2SB and ResShift**. ResShift achieves better results on the ImageNet dataset with complex Real-ESRGAN degradations compared to I2SB in all evaluation metrics (PSNR, SSIM, LPIPS, MUSIQ, CLIPIQA) with a big improvement in perceptual metrics (LPIPS, MUSIQ, CLIPIQA). Our results in Table 7 are consistent with the analysis for comparison between ResShift and I2SB on simpler bicubic degradations, which is given in Appendix B.2 of ResShift. The same conclusion is quantitatively validated in Table 5 and visually supported in Figure 8 of ResShift, respectively. The results of Table 5 in ResShift also show a significant improvement in terms of perceptual quality for ResShift compared to I2SB for bicubic degradations with the same NFE = 15. We explain these results by the specific design of ResShift, which applies the diffusion process in

discrete time in the latent space of VAE and uses the non-uniform geometric noise schedule (Section 2 in ResShift).

**Comparison between students, IBMD and RSD**. For a fair comparison, we compare our RSD model without supervised losses and the IBMD model, because IBMD originally is a data-free distillation method (see contribution 3 in IBMD). The results show that RSD is better in **all evaluation metrics even without supervised losses** (PSNR, SSIM, LPIPS, MUSIQ, CLIPIQA) with a significant improvement in perceptual metrics (LPIPS, MUSIQ, CLIPIQA). The addition of supervised losses in RSD even more increases the margin between all evaluation metrics. In practice, we also found that the IBMD model requires a significant computational budget, which is in line with the complexity reported in Table 9 of IBMD. We trained the IBMD model on 8 A100 for 23 hours, which is $> 4$ times more than the training time of RSD. IBMD also has $> 3$ times bigger number of parameters and $> 8$ times bigger required GPU memory for inference.

**Summary**. For a quantitative comparison, the RSD model without supervised losses outperforms the IBMD model in all evaluation metrics. For a computational comparison, the RSD model has a much faster distillation time, requires less number of parameters, GPU memory, and has a faster generation time during inference compared to IBMD. We will add these results to the revised version of the text. Thus, task-specific features of RSD used for Real-ISR problems are essential for high perceptual quality compared to the diffusion distillation method of IBMD, which is developed for general image-to-image translation problems.

Table 7: Comparison on ImageNet-Test between I2SB (Liu et al., 2023b), ResShift (Yue et al., 2023), and their 1-step distillation versions, IBMD (Gushchin et al., 2025) and RSD, respectively. The best and second best results are highlighted in **bold** and underline.

| Methods | NFE | PSNR↑ | SSIM↑ | LPIPS↓ | CLIPIQA↑ | MUSIQ↑ |
|---|---|---|---|---|---|---|
| I2SB | 15 | 24.80 | 0.663 | 0.302 | 0.444 | 49.584 |
| I2SB | 1 | **25.52** | **0.690** | 0.412 | 0.405 | 34.439 |
| ResShift | 15 | 25.01 | 0.677 | 0.231 | 0.592 | 53.660 |
| IBMD | 1 | 23.91 | 0.619 | 0.284 | 0.505 | 54.667 |
| RSD (**Ours**, distill only) | 1 | 23.97 | 0.643 | 0.217 | 0.660 | 57.831 |
| RSD (**Ours**) | 1 | 24.31 | 0.657 | **0.193** | **0.681** | **58.947** |

Table 8: Training and inference complexity between RSD and IBMD (Gushchin et al., 2025). All methods are tested with an LR image of size $64 \times 64$ for SR factor $\times 4$, and the inference is done on an NVIDIA A100 GPU. The best values are highlighted in **bold**.

| Methods | RSD (**Ours**) | IBMD |
|---|---|---|
| Inference Step (NFE) | **1** | 1 |
| Inference Time (s) | **0.059** | 0.077 |
| # Total Param (M) | **174** | 553 |
| Maximum GPU memory (MB) | **539** | 4676 |
| Training time (hours / # GPU) | **5 / 4 A100** | 23 / 8 A100 |

## A.4 GENERALIZATION OF RSD TO OTHER METHODS.

It is noteworthy that the proof of Proposition (3.1) (see Appendix K), as well as the formulation of our loss function, does not rely on a specific form of processes used in the ResShift model. The only difference from other approaches is how they sample from the joint distribution $p_\theta(\hat{x}_0, y_0, x_t)$ during the training procedure. Usually, $p_\theta(\hat{x}_0, y_0, x_t)$ is written in the following way:

$$p_\theta(\hat{x}_0, y_0, x_t) = q(x_t \mid \hat{x}_0, y_0) p_\theta(\hat{x}_0 \mid y_0) p(y_0) \qquad (22)$$

showing that the only thing that is different for each method is used distribution $q$. Thus, to adopt our approach to other processes, one only needs to change the distribution $q$. For example, to train I2SB or LDM models using RSD formulation, one can use their discrete formulations for both models (Liu et al., 2023a, Eq. 11) in I2SB or (Rombach et al., 2022, Eq. 4) in LDM and plug them into $\mathcal{L}_\theta$.

## B  ALGORITHM OF RSD

The pseudocode for our RSD training algorithm is presented in Algorithm 1.

---

**Algorithm 1:** Residual Shifting Distillation (RSD).

---

**Input:**
Training dataset $p_{\text{data}}(x_0, y_0)$;
Pretrained ResShift Teacher model $f^*$; frozen encoder and decoder of VAE: $\text{Enc}, \text{Dec}$;
Number of fake ResShift ($f_\phi$) training iterations $K$, $f_\phi^{\text{encoder}}$ - encoder part of fake ResShift model;

**Output:**
A trained generator $G_\theta$;

1 **func** SampleEverything()
3    Sample $(x_0, y_0) \sim p_{\text{data}}(x_0, y_0)$
5    $z_y \leftarrow \text{Enc}(\text{upsample}(y_0)); \quad z_0 \leftarrow \text{Enc}(x_0)$
7    Sample $t_n \sim \mathcal{U}\{1, \ldots, T\}$, $z_{t_n} \sim q(z_{t_n}|z_0, z_y)$, $\epsilon \sim \mathcal{N}(0, \mathbf{I})$ // Eq. (27)
9    $\widehat{z}_0^{t_n} \leftarrow G_\theta(z_{t_n}, y_0, t_n, \epsilon)$
11    Sample $t \sim \mathcal{U}\{1, \ldots, T\}$, $z_t \sim q(z_t|\widehat{z}_0^{t_n}, z_y)$ // Eq. (27)
13    **return** $(x_0, y_0, z_0, z_y, t_n, z_{t_n}, \widehat{z}_0^{t_n}, t, z_t)$
14

15 // Initialize generator from pretrained model
16 // Initalize fake ResShift from pretrained model
17 // and GAN discriminator head randomly
18 $G_\theta \leftarrow \text{copyWeightsAndUnfreeze}(f^*)$;
19 $f_\phi \leftarrow \text{copyWeightsAndUnfreezeAndAddNoiseChannels}(f^*)$ // See Appendix C
20 $D_\psi \leftarrow \text{randomInitOfDiscriminatorHead}()$
21

22 **while** *train* **do**
23    // Train fake ResShift model
24    **for** $k \leftarrow 1$ **to** $K$ **do**
25       $(x_0, y_0, z_0, z_y, t_n, z_{t_n}, \widehat{z}_0^{t_n}, t, z_t) \leftarrow \text{SampleEverything}()$ // Generate training data
26       $\mathcal{L}_{\text{fake}} \leftarrow w_t \| f_\phi(z_t, y_0, t) - \widehat{z}_0^{t_n} \|_2^2$ // Eq. (5)
27       $\mathcal{L}_{\text{GAN}} \leftarrow \text{calcGANLossD}(D_\psi(f_\phi^{\text{encoder}}(\widehat{z}_0^{t_n}, y_0, 0)), D_\psi(f_\phi^{\text{encoder}}(z_0, y_0, 0)))$ // Eq. (10)
28       $\mathcal{L}_\phi^{\text{total}} \leftarrow \mathcal{L}_{\text{fake}} + \lambda_2 \mathcal{L}_{\text{GAN}}$ // Eq. (11)
29       Update $\phi$ by using $\frac{\partial \mathcal{L}_\phi^{\text{total}}}{\partial \phi}$
30       Update $\psi$ by using $\frac{\partial \mathcal{L}_{\text{GAN}}}{\partial \psi}$
31    **end for**
32    // Train generator model
33    // Generate training data
34    $(x_0, y_0, z_0, z_y, t_n, z_{t_n}, \widehat{z}_0^{t_n}, t, z_t) \leftarrow \text{SampleEverything}()$
35

36    // Compute $\mathcal{L}_\theta$ loss
37    $\mathcal{L}_\theta \leftarrow \text{calcThetaLoss}(f^*(z_t, y_0, t), f_\phi(z_t, y_0, t), \widehat{z}_0^{t_n})$ // Eq. (8)
38

39    // Compute $\mathcal{L}_{\text{LPIPS}}$ loss
40    Sample $z_T \sim \mathcal{N}(z_T|z_y, \kappa^2 \mathbf{I})$ // Eq. (27)
41    $\widehat{z}_0 \leftarrow G_\theta(z_T, y_0, T, \epsilon)$
42    $\mathcal{L}_{\text{LPIPS}} \leftarrow \text{LPIPS}(x_0, \text{Dec}(\widehat{z}_0))$
43

44    // Compute generator $\mathcal{L}_{\text{GAN}}$ loss
45    $\mathcal{L}_{\text{GAN}} \leftarrow \text{calcGANLossG}(D_\psi(f_\phi^{\text{encoder}}(\widehat{z}_0^{t_n}, y_0, 0)))$ // Eq. (10)
46

47    $\mathcal{L}_\theta^{\text{total}} \leftarrow \mathcal{L}_\theta + \lambda_1 \mathcal{L}_{\text{LPIPS}} + \lambda_2 \mathcal{L}_{\text{GAN}}$ // Eq. (11)
48    Update $\theta$ by using $\frac{\partial \mathcal{L}_\theta^{\text{total}}}{\partial \theta}$
49 **end while**

---

The pseudocode for the baseline ResShift-VSD training algorithm is presented in Algorithm 2, while the foundational theoretical framework is detailed in Appendix A.1. To ensure a fair comparison with the distillation loss in OSEDiff (Wu et al., 2024a), specifically the VSD Loss, under an identical experimental setup (i.e., ResShift), we adapted it to the ResShift framework using the same implementation details.

---

**Algorithm 2:** ResShift-VSD.

---

**Input:**
Training dataset $p_{\text{data}}(x_0, y_0)$;
Pretrained ResShift Teacher model $f^*$; frozen encoder and decoder of VAE: Enc, Dec;
Number of fake ResShift ($f_\phi$) training iterations $K$;

**Output:**
A trained generator $G_\theta$;

```
1  func SampleEverything()
3      Sample (x_0, y_0) ~ p_data(x_0, y_0);
5      z_y ← Enc(upsample(y_0))
7      Sample z_T ~ N(z_y, κ²η_T I) // Eq.   (27)
9      ẑ_0 ← G_θ(z_T, y_0, T)
11     Sample t ~ U{1,...,T},  z_t ~ q(z_t|ẑ_0, z_y) // Eq.   (27)
13     return (y_0, t, z_t, ẑ_0)
14
15 // Initialize generator from pretrained model
16 // Initialize fake ResShift from pretrained model
17 G_θ ← copyWeightsAndUnfreeze(f*);
18 f_φ ← copyWeightsAndUnfreeze(f*);
19
20 while train do
21     // Train fake ResShift model
22     for k ← 1 to K do
23         (y_0, t, z_t, ẑ_0) ← SampleEverything() // Generate training data
24         L_fake ← w_t||f_φ(z_t, y_0, t) − ẑ_0||²₂ // Eq.   (5)
25         Update φ by using ∂L_fake/∂φ
26     end for
27
28     // Train generator model
29     (y_0, t, z_t, ẑ_0) ← SampleEverything() // Generate training data
30     L_θ ← calcThetaLoss(stopgrad( f*(z_t, y_0, t)), stopgrad( f_φ(z_t, y_0, t)), ẑ_0) // Eq.   (8)
31     Update θ by using ∂L_θ/∂θ
32 end while
```

---

Table 9: Notation used in our paper. Pixel-space refers to the image domain, while latent-space refers to the internal representation domain.

| Symbol | Description | Space |
|---|---|---|
| $x_0$ | Original high-resolution image | Pixel-space |
| $\hat{x}_0$ | Reconstructed high-resolution image from $\hat{z}_0$ | Pixel-space |
| $y_0$ | Low-resolution input image | Pixel-space |
| $z_0$ | Latent representation of $x_0$ | Latent-space |
| $z_y$ | Latent representation of $y_0$ | Latent-space |
| $z_{t_n}$ | Noised latent sampled from $z_0$ | Latent-space |
| $z_T$ | Noised latent sampled from $\mathcal{N}(z_T|z_y, \kappa^2 I)$ | Latent-space |
| $\hat{z}_0$ | Denoised latent output of generator $G_\theta(z_T, y_0, T, \epsilon)$ | Latent-space |
| $z_t$ | Noised latent sampled from $q(z_t|\hat{z}_0^{t_n}, z_y)$ | Latent-space |
| $\hat{z}_0^{t_n}$ | Generator output from $G_\theta(z_{t_n}, y_0, t_n, \epsilon)$ | Latent-space |
| $f^*(z_t, y_0, t)$ | Frozen teacher network output | Latent-space |
| $f_\phi(z_t, y_0, t)$ | Student network output (trained) | Latent-space |
| $\epsilon$ | Noise variable sampled from $\mathcal{N}(0, I)$ | Latent-space |

## C EXPERIMENTAL DETAILS

**Noise condition**. By default, fake ResShift and generator models are initialized with teacher weights. Furthermore, for noise conditioning, as described in §3.2, we implement an additional convolutional channel to expand the generator's first convolutional layer to accept noise as an additional input. The noise is concatenated with the encoded low-resolution image and is processed by a separate zero-initialized convolutional layer.

**Training hyperparameters**. We use the same hyperparameters as SinSR for training, including the batch size, EMA rate, and optimizer type. To achieve smoother convergence, we replace the learning rate scheduler with a constant learning rate of $5 \times 10^{-5}$, matching the base learning rate of SinSR. Additionally, we adjust the AdamW (Loshchilov & Hutter, 2019) optimizer's $\beta$ parameters to $[0.9, 0.95]$ to further stabilize training. To ensure controlled adaptation between the generator and the fake ResShift models, we update the generator's weights once for every $K = 5$ updates of the fake model, following the strategy of DMD2 (Yin et al., 2024a). Furthermore, we adopt the loss normalization technique proposed in (Zhou et al., 2024) to improve training stability. In the final loss function (Eq. 11), we set $\lambda_1 = 2$ and $\lambda_2 = 3 \cdot 10^{-3}$ following OSEDiff (Wu et al., 2024a) and DMD2, respectively.

**Training time**. The complete training process, performed on 4 NVIDIA A100 GPUs, takes approximately 5 hours. During this time, the student model undergoes around 3000 gradient update iterations, while the fake model completes 15000 iterations. In practice, we found that SinSR (Wang et al., 2024b) requires around 60 hours on a single A100 GPU for 30000 iterations (2.57 days in Table 7 of SinSR (Wang et al., 2024b)) and SinSR converges roughly 3 times slower than RSD. We explain this difference by **simulation-free property** of RSD, which SinSR does not have. We recall that SinSR is a knowledge distillation method, which runs full teacher ResShift model for all $T = 15$ steps during the training according to Equations 5 and 6 in SinSR (Wang et al., 2024b):

$$x_{t-1} = k_t f^*(x_t, y_0, t) + m_t x_t + j_t y_0, \quad t \in \{T, T-1, \ldots, 2, 1\}, \tag{23}$$

$$F(x_T, y_0) = x_0, \quad x_T = y_0 + \kappa\sqrt{\eta_T}\epsilon, \quad \epsilon \sim \mathcal{N}(0, \mathbf{I}), \tag{24}$$

$$\mathcal{L}_{distill,SinSR} = L_{MSE}(f_\theta(x_T, y_0, T), F(x_T, y_0)), \tag{25}$$

where $f_\theta(x_T, y_0, T)$ is the student network in SinSR, which directly predicts the HR image in only one step, $F(x_T, y_0)$ represents the deterministic sampling with the ResShift teacher model using Equation (23), and $f^*(x_t, y_0, t)$ is the teacher model in ResShift. During training, RSD does not require full teacher simulation as SinSR in Equation (23). However, in the training of RSD, an additional $K = 5$ updates for the fake model are required, while SinSR does not have any fake model. Thus, RSD achieves an acceleration of around $\times 3$ for training compared to SinSR.

**Codebase**. Our method is implemented based on the original SinSR repository (Wang et al., 2024b), which serves as the primary code source for our experiments. We build upon this framework to integrate our training algorithm, which is described in Appendix B.

**Teacher checkpoint**. Following SinSR repository (Wang et al., 2024b), we also distill the same ResShift checkpoint `resshift_realsrx4_s15_v1.pth`, which was trained with 300k iterations.

**Datasets and baselines**. Table 10 lists details on the datasets used for training and testing, including their sources and download links. Table 11 provides associated licenses for considered datasets. Table 12 lists the models used for training and quality comparison and links to access them.

**Metrics calculation of SR models**. For calculating SR metrics, we use PyTorch Toolbox for Image Quality Assessment and `pyiqa` package (Chen & Mo, 2022). We also used the image quality assessment script provided in the OSEDiff GitHub repository.

Table 10: The used datasets and their sources

| Name | URL | Citation |
|---|---|---|
| RealSR-V3 | GitHub Link | (Cai et al., 2019) |
| RealSet65 | GitHub Link | (Yue et al., 2023) |
| DRealSR | GitHub Link | (Wei et al., 2020) |
| ImageNet | Website Link | (Deng et al., 2009) |
| ImageNet-Test | Google Drive Link | (Yue et al., 2023) |
| DIV2K-Val-512 | Hugging Face Link | (Agustsson & Timofte, 2017; Wang et al., 2024a) |
| DRealSR-512 | Hugging Face Link | (Wang et al., 2024a; Wei et al., 2020) |
| RealSR-512 | Hugging Face Link | (Wang et al., 2024a; Cai et al., 2019) |
| RealLR200 | Google Drive Link | (Wu et al., 2024b) |
| RealLQ250 | Google Drive Link | (Ai et al., 2024) |

Table 11: The used datasets and their licenses

| Name | License |
|---|---|
| RealSR-V3 | NTU S-Lab License 1.0 |
| DRealSR | Unknown |
| ImageNet | Custom (research, non-commercial) |
| ImageNet-Test | NTU S-Lab License |
| DIV2K-Val-512 | NTU S-Lab License |
| DRealSR-512 | NTU S-Lab License |
| RealSR-512 | NTU S-Lab License |
| RealLR200 | Apache 2.0 License |
| RealLQ250 | Apache 2.0 License |

Table 12: Baselines used for comparison. In each case, we used original code from GitHub repositories and model weights.

| Name | URL | Citation | License |
|---|---|---|---|
| Real-ESRGAN | GitHub Link | (Wang et al., 2021) | BSD 3-Clause License |
| BSRGAN | GitHub Link | (Zhang et al., 2021) | Apache-2.0 license |
| SwinIR | GitHub Link | (Liang et al., 2021) | Apache-2.0 license |
| ResShift | GitHub Link | (Yue et al., 2023) | NTU S-Lab License 1.0 |
| SinSR | GitHub Link | (Wang et al., 2024b) | CC BY-NC-SA 4.0 |
| SUPIR | GitHub Link | (Yu et al., 2024) | SUPIR Software License |
| OSEDiff | GitHub Link | (Wu et al., 2024a) | Apache License 2.0 |
| AdcSR | GitHub Link | (Chen et al., 2025) | Apache License 2.0 |
| PiSA-SR | GitHub Link | (Sun et al., 2025) | Apache License 2.0 |
| TSD-SR | GitHub Link | (Dong et al., 2025) | Apache License 2.0 |
| CTMSR | GitHub Link | (You et al., 2025) | MIT License |
| CCSR | GitHub Link | (Sun et al., 2024) | Apache License 2.0 |
| InvSR | GitHub Link | (Yue et al., 2025) | NTU S-Lab License 1.0 |

# D  ADDITIONAL QUANTITATIVE RESULTS

We present an additional set of quantitative results, including more baselines and evaluations on full-size DRealSR (Wei et al., 2020), RealLR200 (Wu et al., 2024b), and RealLQ250 (Ai et al., 2024), which were not included in the main text due to space limitations:

- Table 13 provides results on full-size images from the DRealSR dataset (Wei et al., 2020).

- Table 14 provides non-reference results on full-size images from the RealLR200 (Wu et al., 2024b) and RealLQ250 (Ai et al., 2024) datasets.

- Table 15 presents an extension version of Table 1 on RealSR (Cai et al., 2019) and RealSet65 (Yue et al., 2023) datasets with additional baselines.

- Table 16 presents an extension version of Table 2 on the ImageNet-Test dataset (Yue et al., 2023) with additional baselines.

- Table 17 presents an extension version of Table 3 on crops from DIV2K (Agustsson & Timofte, 2017), RealSR, and DRealSR used in StableSR (Wang et al., 2024a) with additional baselines.

**Table 13**. We evaluated the following models for Table 13 and followed their official implementations listed in Table 12:

1. **Diffusion-based SR models**. We ran pre-trained models of ResShift (Yue et al., 2023), SinSR (Wang et al., 2024b), OSEDiff (Wu et al., 2024a), and SUPIR (Yu et al., 2024) as representative members of diffusion-based SR models. We used the following checkpoints from the respective official repositories listed in Table 12: `resshift_realsrx4_s15_v1.pth`, `SinSR_v2.pth`, `osediff.pkl`, and `SUPIR-v0Q.ckpt`. Due to the high resolution of DRealSR images and the high demands for GPU memory for the SUPIR model, we ran it with tiled VAE using the flag `--use_tile_vae`.

2. **State-of-the-art diffusion-based one-step SR models**. In addition to ResShift, SinSR, OSEDiff, and SUPIR models, we also ran pre-trained recent state-of-the-art one-step diffusion SR models, including TSD-SR (Dong et al., 2025), PiSA-SR (Sun et al., 2025), CTMSR (You et al., 2025), CCSR (Sun et al., 2024), and InvSR (Yue et al., 2025). We used the following checkpoints from the respective repositories listed in Table 12: 1) TSD-SR - LoRA weights from the folder `checkpoint/tsdsr-mse`, embedding weights from the folder `dataset/default`, and the teacher SD3-medium model from Hugging Face Link; 2) PiSA-SR - `pisa_sr.pkl`; 3) CTMSR - `CTMSR.pth`; 4) InvSR - `noise_predictor_sd_turbo_v5_diftune.pth`; 5) CCSR - to follow CCSR GitHub repository, we used ControlNet weights from Google Drive Link, VAE weights from Google Drive Link, and pre-trained ControlNet weights from Google Drive Link, and Dino models from Google Drive Link, respectively.

3. **GAN-based SR models**. We ran pre-trained Real-ESRGAN (Wang et al., 2021) and BSRGAN (Zhang et al., 2021) GAN-based SR models with the checkpoint names `RealESRGAN_x4plus.pth` and `BSRGAN.pth`, provided in the respective GitHub repositories listed in Table 12.

4. **Transformer-based SR models**. We ran pre-trained SwinIR model (Liang et al., 2021) with the checkpoint name `003_realSR_BSRGAN_DFOWMFC_s64w8_SwinIR-L_x4_GAN.pth` as the representative model from transformer-based SR models using the respective GitHub repository listed in Table 12.

We compute the same set of metrics as in Table 3 - PSNR, SSIM, LPIPS, CLIPIQA, MUSIQ, DISTS, NIQE, and MANIQA-PIPAL.

**Table 14**. We evaluated RSD and the following baselines on real-world benchmarks, namely RealLR200 (Wu et al., 2024b) and RealLQ250 (Ai et al., 2024), using no-reference perceptual metrics (CLIPIQA, MUSIQ, NIQE, MANIQF), which follows the evaluation protocol of SeeSR (Wu et al., 2024a, Table 2) and DreamClear (Ai et al., 2024, Table 1):

1. **Diffusion-based SR models without T2I models**. We evaluated methods which were trained on the ImageNet data, namely ResShift, SinSR, CTMSR, and RSD.

2. **T2I-based diffusion SR models**. We evaluated SUPIR, OSEDiff, AdcSR, PiSA-SR, TSD-SR, InvSR, and CCSR. For AdcSR (Chen et al., 2025), we used the weights from the checkpoint `net_params_200.pkl` from the respective GitHub repository.

**Table 15** . We report an extended version of Table 1 with additional baselines used in ResShift and SinSR papers:

1. **GAN-based SR models**. We evaluated Real-ESRGAN (Wang et al., 2021) and BSRGAN (Zhang et al., 2021) on RealSR and RealSet65.

2. **SwinIR**. We also evaluated SwinIR on RealSR and RealSet65.

3. **State-of-the-art diffusion-based one-step SR models**. In addition to TSD-SR, PiSA-SR, CTMSR, and AdcSR, we also evaluated InvSR and CCSR using the same pre-trained models as for Table 13.

**Table 16** . We report an extended version of Table 2 with additional baselines used in ResShift and SinSR papers:

1. **Diffusion-based SR models**. We borrow the results of Table 2 from SinSR for LDM-15 and LDM-30 (Rombach et al., 2022), and SinSR (Wang et al., 2024b). We borrow the results of Table 3 from (Yue et al., 2023) for ResShift.

2. **GAN-based SR models**. We borrow the results of Table 2 from SinSR for ESRGAN (Wang et al., 2019), RealSR-JPEG (Ji et al., 2020), Real-ESRGAN (Wang et al., 2021), BSRGAN (Zhang et al., 2021).

3. **DASR and SwinIR**. We also borrow the results of Table 2 from SinSR for DASR (Liang et al., 2022b) and SwinIR (Liang et al., 2021).

4. **State-of-the-art diffusion-based one-step SR models**. In addition to TSD-SR, PiSA-SR, CTMSR, and AdcSR, we also evaluated InvSR and CCSR using the same pre-trained models as for Table 13.

**Table 17** . We report an extended version of Table 3 with additional baselines used in the OSEDiff paper:

1. **Diffusion-based SR models**. We borrow the results of Table 1 from OSEDiff for StableSR (Wang et al., 2024a), DiffBIR (Lin et al., 2025), SeeSR (Wu et al., 2024b), PASD (Yang et al., 2025), ResShift (Yue et al., 2023), and SinSR (Wang et al., 2024b).

2. **GAN-based SR models**. We borrow the results of Table 1 from OSEDiff for Real-ESRGAN (Wang et al., 2021), BSRGAN (Zhang et al., 2021), LDL (Liang et al., 2022a), and FeMASR (Chen et al., 2022).

3. **State-of-the-art diffusion-based one-step SR models**. In addition to TSD-SR, PiSA-SR, CTMSR, and AdcSR, we also evaluated InvSR and CCSR using the same pre-trained models as for Table 13.

These results demonstrate that our RSD model achieves performance comparable to the state-of-the-art diffusion SR models across a broad range of metrics and methods under the requirements of a restricted computational budget. We discuss a comparison between RSD and recent one-step diffusion SR models, namely CTMSR, TSD-SR, PiSA-SR, AdcSR, InvSR, and CCSR, in Appendix E.

Table 13: Quantitative results of models on full size images from DRealSR (Wei et al., 2020). The best and second best results are highlighted in **bold** and underline.

| Methods | NFE | PSNR↑ | SSIM↑ | LPIPS↓ | CLIPIQA↑ | MUSIQ↑ | DISTS↓ | NIQE↓ | MANIQA↑ |
|---|---|---|---|---|---|---|---|---|---|
| BSRGAN (Zhang et al., 2021) | 1 | 28.34 | 0.8206 | 0.2929 | 0.5704 | 35.500 | 0.1636 | 4.6811 | 0.4682 |
| Real-ESRGAN (Wang et al., 2021) | 1 | 27.91 | 0.8249 | 0.2818 | 0.5180 | 35.255 | 0.1464 | 4.7142 | 0.4756 |
| SwinIR (Liang et al., 2021) | 1 | 28.31 | **0.8272** | **0.2741** | 0.5072 | 35.826 | **0.1387** | 4.6665 | 0.4617 |
| SUPIR (Yu et al., 2024) | 50 | 25.73 | 0.7224 | 0.3906 | 0.5862 | 36.089 | 0.1944 | 4.4685 | 0.5720 |
| OSEDiff (Wu et al., 2024a) | 1 | 26.67 | 0.7922 | 0.3123 | 0.7264 | 37.761 | 0.1617 | 4.1768 | 0.5883 |
| PiSA-SR (Sun et al., 2025) | 1 | 27.43 | 0.8119 | 0.2844 | 0.6878 | 35.060 | 0.1537 | 4.4783 | 0.5615 |
| TSD-SR (Dong et al., 2025) | 1 | 26.53 | 0.7637 | 0.3084 | **0.7517** | 37.395 | 0.1567 | **3.6624** | 0.5549 |
| InvSR (Yue et al., 2025) | 1 | 26.06 | 0.7455 | 0.3578 | 0.7485 | 33.878 | 0.1838 | 3.7279 | **0.5928** |
| CCSR (Sun et al., 2024) | 1 | 27.71 | 0.8022 | 0.3208 | 0.7104 | 35.716 | 0.1816 | 4.3081 | 0.5720 |
| ResShift (Yue et al., 2023) | 15 | **28.76** | 0.7863 | 0.4310 | 0.5838 | 32.042 | 0.2314 | 6.6335 | 0.4297 |
| CTMSR (You et al., 2025) | 1 | 28.28 | 0.8017 | 0.3355 | 0.6821 | 33.206 | 0.1946 | 4.7795 | 0.4702 |
| SinSR (Wang et al., 2024b) | 1 | 27.32 | 0.7233 | 0.4452 | 0.7223 | 32.800 | 0.2368 | 5.5748 | 0.4757 |
| RSD (**Ours**) | 1 | 27.66 | 0.7864 | 0.3105 | 0.7398 | **38.340** | 0.1868 | 4.6098 | 0.5314 |

Table 14: Quantitative results of models on RealLR200 (Wu et al., 2024b) an RealLQ250 (Ai et al., 2024) datasets. The best and second best results are highlighted in **bold** and underline.

| Methods | T2I prior | NFE | Datasets | | | | | | | |
|---|---|---|---|---|---|---|---|---|---|---|
| | | | RealLR200 | | | | RealLQ250 | | | |
| | | | CLIPIQA↑ | MUSIQ↑ | NIQE↓ | MANIQA↑ | CLIPIQA↑ | MUSIQ↑ | NIQE↓ | MANIQA↑ |
| SUPIR (Yu et al., 2024) | | 50 | 0.6188 | 64.79 | 4.1862 | 0.6120 | 0.5746 | 65.72 | **3.6607** | 0.5969 |
| OSEDiff (Wu et al., 2024a) | | 1 | 0.6728 | 69.45 | 4.0506 | 0.6153 | 0.6724 | 69.56 | 3.9682 | 0.5889 |
| AdcSR (Chen et al., 2025) | yes, | 1 | 0.7047 | 70.35 | 3.8792 | 0.6174 | 0.6889 | 69.98 | 3.7181 | 0.5944 |
| PiSA-SR (Sun et al., 2025) | > 450M params | 1 | 0.7039 | 70.90 | 3.9594 | 0.6419 | 0.7054 | 71.25 | 3.9162 | **0.6190** |
| TSD-SR (Dong et al., 2025) | | 1 | **0.7335** | **72.06** | **3.8352** | 0.6248 | **0.7368** | **73.22** | 3.6996 | 0.6037 |
| InvSR (Yue et al., 2025) | | 1 | 0.6774 | 68.15 | 4.0378 | **0.6461** | 0.6499 | 64.77 | 4.6505 | 0.5810 |
| CCSR (Sun et al., 2024) | | 1 | 0.6937 | 70.49 | 4.3108 | 0.6319 | 0.6850 | 70.80 | 4.4760 | 0.6021 |
| ResShift (Yue et al., 2023) | | 15 | 0.6368 | 61.80 | 5.7016 | 0.5436 | 0.6348 | 61.99 | 5.7622 | 0.5364 |
| SinSR (Wang et al., 2024b) | no, | 1 | 0.7089 | 64.90 | 5.3329 | 0.5561 | 0.7142 | 65.29 | 5.4630 | 0.5294 |
| CTMSR (You et al., 2025) | < 200M params | 1 | 0.6754 | 67.63 | 4.2943 | 0.5426 | 0.6701 | 68.07 | 4.5831 | 0.5130 |
| RSD (**Ours**) | | 1 | 0.7151 | 68.66 | 4.7074 | 0.5949 | 0.7252 | 69.63 | 4.5531 | 0.5826 |

Table 15: Extended quantitative results of models on two real-world datasets. The best and second best results are highlighted in **bold** and underline.

| Methods | NFE | Datasets | | | | | | |
|---|---|---|---|---|---|---|---|---|
| | | RealSR | | | | | RealSet65 | |
| | | PSNR↑ | SSIM↑ | LPIPS↓ | CLIPIQA↑ | MUSIQ↑ | CLIPIQA↑ | MUSIQ↑ |
| BSRGAN (Zhang et al., 2021) | 1 | **26.51** | 0.775 | 0.269 | 0.5439 | 63.586 | 0.6163 | 65.582 |
| Real-ESRGAN (Wang et al., 2021) | 1 | 25.85 | 0.773 | 0.273 | 0.4898 | 59.678 | 0.5995 | 63.220 |
| SwinIR (Liang et al., 2021) | 1 | 26.43 | **0.786** | **0.251** | 0.4654 | 59.636 | 0.5782 | 63.822 |
| SUPIR (Yu et al., 2024) | 50 | 24.38 | 0.698 | 0.331 | 0.5449 | 63.676 | 0.6133 | 66.460 |
| OSEDiff (Wu et al., 2024a) | 1 | 25.25 | 0.737 | 0.299 | 0.6772 | 67.602 | 0.6836 | 68.853 |
| AdcSR (Chen et al., 2025) | 1 | 25.63 | 0.735 | 0.300 | 0.7033 | 67.550 | 0.7044 | 69.185 |
| PiSA-SR (Sun et al., 2025) | 1 | 25.59 | 0.750 | **0.271** | 0.6678 | 67.993 | 0.7062 | 70.208 |
| TSD-SR (Dong et al., 2025) | 1 | 24.88 | 0.723 | 0.281 | 0.7336 | **69.871** | 0.7263 | **70.958** |
| InvSR (Yue et al., 2025) | 1 | 24.73 | 0.731 | 0.275 | 0.6798 | 66.403 | 0.6990 | 67.770 |
| CCSR (Sun et al., 2024) | 1 | 25.99 | 0.752 | 0.287 | 0.6656 | 67.991 | 0.7150 | 70.731 |
| ResShift (Yue et al., 2023) | 15 | 26.49 | 0.754 | 0.360 | 0.5958 | 59.873 | 0.6537 | 61.330 |
| CTMSR (You et al., 2025) | 1 | 26.18 | **0.765** | 0.294 | 0.6449 | 64.782 | 0.6871 | 67.032 |
| SinSR (distill only) (Wang et al., 2024b) | 1 | 26.14 | 0.732 | 0.357 | 0.6119 | 57.118 | 0.6822 | 61.267 |
| SinSR (Wang et al., 2024b) | 1 | 25.83 | 0.717 | 0.365 | 0.6887 | 61.582 | 0.7150 | 62.169 |
| ResShift-VSD (Appendix A) | 1 | 23.96 | 0.616 | 0.466 | 0.7479 | 63.298 | **0.7606** | 66.701 |
| RSD (**Ours**, distill only) | 1 | 24.92 | 0.696 | 0.355 | **0.7518** | 66.430 | 0.7534 | 68.383 |
| RSD (**Ours**) | 1 | 25.91 | 0.754 | 0.273 | 0.7060 | 65.860 | 0.7267 | 69.172 |

Table 16: Extended quantitative results of models on ImageNet-Test (Yue et al., 2023). The best and second best results are highlighted in **bold** and underline.

| Methods | NFE | PSNR↑ | SSIM↑ | LPIPS↓ | CLIPIQA↑ | MUSIQ↑ |
|---|---|---|---|---|---|---|
| ESRGAN (Wang et al., 2019) | 1 | 20.67 | 0.448 | 0.485 | 0.451 | 43.615 |
| RealSR-JPEG (Ji et al., 2020) | 1 | 23.11 | 0.591 | 0.326 | 0.537 | 46.981 |
| BSRGAN (Zhang et al., 2021) | 1 | 24.42 | 0.659 | 0.259 | 0.581 | 54.697 |
| SwinIR (Liang et al., 2021) | 1 | 23.99 | 0.667 | 0.238 | 0.564 | 53.790 |
| Real-ESRGAN (Wang et al., 2021) | 1 | 24.04 | 0.665 | 0.254 | 0.523 | 52.538 |
| DASR (Liang et al., 2022b) | 1 | 24.75 | 0.675 | 0.250 | 0.536 | 48.337 |
| LDM (Rombach et al., 2022) | 30 | 24.49 | 0.651 | 0.248 | 0.572 | 50.895 |
| LDM (Rombach et al., 2022) | 15 | 24.89 | 0.670 | 0.269 | 0.512 | 46.419 |
| SUPIR (Yu et al., 2024) | 50 | 22.56 | 0.574 | 0.302 | **0.786** | 60.487 |
| OSEDiff (Wu et al., 2024a) | 1 | 23.02 | 0.619 | 0.253 | 0.677 | 60.755 |
| AdcSR (Chen et al., 2025) | 1 | 22.99 | 0.615 | 0.252 | 0.711 | 63.218 |
| PiSA-SR (Sun et al., 2025) | 1 | 24.29 | 0.670 | 0.213 | 0.629 | 62.137 |
| TSD-SR (Dong et al., 2025) | 1 | 23.58 | 0.645 | 0.197 | 0.673 | **65.299** |
| InvSR (Yue et al., 2025) | 1 | 21.31 | 0.604 | 0.293 | 0.641 | 54.870 |
| CCSR (Sun et al., 2024) | 1 | 24.79 | **0.677** | 0.238 | 0.602 | 61.789 |
| ResShift (Yue et al., 2023) | 15 | **25.01** | **0.677** | 0.231 | 0.592 | 53.660 |
| CTMSR (You et al., 2025) | 1 | 24.73 | 0.666 | 0.197 | 0.691 | 60.142 |
| SinSR (distill only) (Wang et al., 2024b) | 1 | 24.69 | 0.664 | 0.222 | 0.607 | 53.316 |
| SinSR (Wang et al., 2024b) | 1 | 24.56 | 0.657 | 0.221 | 0.611 | 53.357 |
| ResShift-VSD (Appendix A) | 1 | 23.69 | 0.624 | 0.230 | 0.665 | 58.630 |
| RSD (**Ours**, distill only) | 1 | 23.97 | 0.643 | 0.217 | 0.660 | 57.831 |
| RSD (**Ours**) | 1 | 24.31 | 0.657 | **0.193** | 0.681 | 58.947 |

Table 17: Extended quantitative results of models on crops from StableSR (Wang et al., 2024a). The best and second best results are highlighted in **bold** and underline.

| Datasets | Methods | NFE | PSNR↑ | SSIM↑ | LPIPS↓ | DISTS↓ | NIQE↓ | MUSIQ↑ | MANIQA↑ | CLIPIQA↑ | FID↓ |
|---|---|---|---|---|---|---|---|---|---|---|---|
| DIV2K-Val | BSRGAN (Zhang et al., 2021) | 1 | 24.58 | 0.6269 | 0.3351 | 0.2275 | 4.7518 | 61.20 | 0.5071 | 0.5247 | 44.23 |
| | Real-ESRGAN (Wang et al., 2021) | 1 | 24.29 | **0.6371** | 0.3112 | 0.2141 | 4.6786 | 61.06 | 0.5501 | 0.5277 | 37.64 |
| | LDL (Liang et al., 2022a) | 1 | 23.83 | 0.6344 | 0.3256 | 0.2227 | 4.8554 | 60.04 | 0.5350 | 0.5180 | 42.29 |
| | FeMASR (Chen et al., 2022) | 1 | 23.06 | 0.5887 | 0.3126 | 0.2057 | 4.7410 | 60.83 | 0.5074 | 0.5997 | 35.87 |
| | StableSR (Wang et al., 2024a) | 200 | 23.26 | 0.5726 | 0.3113 | 0.2048 | 4.7581 | 65.92 | 0.6192 | 0.6771 | **24.44** |
| | DiffBIR (Lin et al., 2025) | 50 | 23.64 | 0.5647 | 0.3524 | 0.2128 | 4.7042 | 65.81 | 0.6210 | 0.6704 | 30.72 |
| | SeeSR (Wu et al., 2024b) | 50 | 23.68 | 0.6043 | 0.3194 | 0.1968 | 4.8102 | 68.67 | 0.6240 | 0.6936 | 25.90 |
| | PASD (Yang et al., 2025) | 20 | 23.14 | 0.5505 | 0.3571 | 0.2207 | 4.3617 | 68.95 | **0.6483** | 0.6788 | 29.20 |
| | SUPIR (Yu et al., 2024) | 50 | 22.13 | 0.5280 | 0.3923 | 0.2314 | 5.6758 | 63.82 | 0.5933 | 0.7147 | 31.46 |
| | OSEDiff (Wu et al., 2024a) | 1 | 23.72 | 0.6108 | 0.2941 | 0.1976 | 4.7097 | 67.97 | 0.6148 | 0.6683 | 26.32 |
| | AdcSR (Chen et al., 2025) | 1 | 23.74 | 0.6017 | 0.2853 | 0.1899 | 4.3579 | 68.00 | 0.6073 | 0.6764 | 25.52 |
| | PiSA-SR (Sun et al., 2025) | 1 | 23.87 | 0.6058 | 0.2823 | 0.1934 | 4.5565 | 69.68 | 0.6375 | 0.6928 | 25.09 |
| | TSD-SR (Dong et al., 2025) | 1 | 23.02 | 0.5808 | **0.2673** | **0.1821** | **4.3244** | **71.69** | 0.6192 | **0.7416** | 29.16 |
| | InvSR (Yue et al., 2025) | 1 | 23.10 | 0.5985 | 0.3045 | 0.1985 | 4.7056 | 68.43 | 0.6385 | 0.7117 | 28.45 |
| | CCSR (Sun et al., 2024) | 1 | 24.30 | 0.6283 | 0.2979 | 0.2020 | 5.3367 | 69.52 | 0.6145 | 0.6752 | 30.86 |
| | ResShift (Yue et al., 2023) | 15 | 24.65 | 0.6181 | 0.3349 | 0.2213 | 6.8212 | 61.09 | 0.5454 | 0.6071 | 36.11 |
| | SinSR (Wang et al., 2024b) | 1 | 24.41 | 0.6018 | 0.3240 | 0.2066 | 6.0159 | 62.82 | 0.5386 | 0.6471 | 35.57 |
| | CTMSR (You et al., 2025) | 1 | **24.88** | 0.6265 | 0.3026 | 0.2040 | 5.1146 | 65.62 | 0.5165 | 0.6601 | 34.15 |
| | RSD (**Ours**) | 1 | 23.91 | 0.6042 | 0.2857 | 0.1940 | 5.1987 | 68.05 | 0.5937 | 0.6967 | 34.84 |
| DRealSR | BSRGAN (Zhang et al., 2021) | 1 | **28.75** | 0.8031 | 0.2883 | 0.2142 | 6.5192 | 57.14 | 0.4878 | 0.4915 | 155.63 |
| | Real-ESRGAN (Wang et al., 2021) | 1 | 28.64 | 0.8053 | 0.2847 | **0.2089** | 6.6928 | 54.18 | 0.4907 | 0.4422 | 147.62 |
| | LDL (Liang et al., 2022a) | 1 | 28.21 | **0.8126** | **0.2815** | 0.2132 | 7.1298 | 53.85 | 0.4914 | 0.4310 | 155.53 |
| | FeMASR (Chen et al., 2022) | 1 | 26.90 | 0.7572 | 0.3169 | 0.2235 | 5.9073 | 53.74 | 0.4420 | 0.5464 | 157.78 |
| | StableSR (Wang et al., 2024a) | 200 | 28.03 | 0.7536 | 0.3284 | 0.2269 | 6.5239 | 58.51 | 0.5601 | 0.6356 | 148.98 |
| | DiffBIR (Lin et al., 2025) | 50 | 26.71 | 0.6571 | 0.4557 | 0.2748 | 6.3124 | 61.07 | 0.5930 | 0.6395 | 166.79 |
| | SeeSR (Wu et al., 2024b) | 50 | 28.17 | 0.7691 | 0.3189 | 0.2315 | 6.3967 | 64.93 | 0.6042 | 0.6804 | 147.39 |
| | PASD (Yang et al., 2025) | 20 | 27.36 | 0.7073 | 0.3760 | 0.2531 | **5.5474** | 64.87 | 0.6169 | 0.6808 | 156.13 |
| | SUPIR (Yu et al., 2024) | 50 | 24.93 | 0.6360 | 0.4263 | 0.2823 | 7.4336 | 59.39 | 0.5537 | 0.6799 | 164.86 |
| | OSEDiff (Wu et al., 2024a) | 1 | 27.92 | 0.7835 | 0.2968 | 0.2165 | 6.4902 | 64.65 | 0.5899 | 0.6963 | 135.30 |
| | AdcSR (Chen et al., 2025) | 1 | 28.10 | 0.7726 | 0.3046 | 0.2200 | 6.4467 | 66.27 | 0.5916 | 0.7049 | 134.05 |
| | PiSA-SR (Sun et al., 2025) | 1 | 28.32 | 0.7804 | 0.2960 | 0.2169 | 6.1766 | 66.11 | 0.6161 | 0.6968 | **130.61** |
| | TSD-SR (Dong et al., 2025) | 1 | 27.77 | 0.7559 | 0.2967 | 0.2136 | 5.9131 | **66.62** | 0.5874 | **0.7343** | 134.98 |
| | InvSR (Yue et al., 2025) | 1 | 25.79 | 0.7176 | 0.3471 | 0.2381 | 5.8627 | 64.92 | **0.6212** | 0.7185 | 166.51 |
| | CCSR (Sun et al., 2024) | 1 | 28.24 | 0.7818 | 0.3201 | 0.2327 | 6.7901 | 66.28 | 0.6056 | 0.6632 | 157.23 |
| | ResShift (Yue et al., 2023) | 15 | 28.46 | 0.7673 | 0.4006 | 0.2656 | 8.1249 | 50.60 | 0.4586 | 0.5342 | 172.26 |
| | SinSR (Wang et al., 2024b) | 1 | 28.36 | 0.7515 | 0.3665 | 0.2485 | 6.9907 | 55.33 | 0.4884 | 0.6383 | 170.57 |
| | CTMSR (You et al., 2025) | 1 | 28.65 | 0.7834 | 0.3238 | 0.2358 | 6.1828 | 59.78 | 0.4861 | 0.6497 | 163.63 |
| | RSD (**Ours**) | 1 | 27.40 | 0.7559 | 0.3042 | 0.2343 | 6.2577 | 62.03 | 0.5625 | 0.7019 | 167.47 |
| RealSR | BSRGAN (Zhang et al., 2021) | 1 | **26.39** | **0.7654** | **0.2670** | 0.2121 | 5.6567 | 63.21 | 0.5399 | 0.5001 | 141.28 |
| | Real-ESRGAN (Wang et al., 2021) | 1 | 25.69 | 0.7616 | 0.2727 | 0.2063 | 5.8295 | 60.18 | 0.5487 | 0.4449 | 135.18 |
| | LDL (Liang et al., 2022a) | 1 | 25.28 | 0.7567 | 0.2766 | 0.2121 | 6.0024 | 60.82 | 0.5485 | 0.4477 | 142.71 |
| | FeMASR (Chen et al., 2022) | 1 | 25.07 | 0.7358 | 0.2942 | 0.2288 | 5.7885 | 58.95 | 0.4865 | 0.5270 | 141.05 |
| | StableSR (Wang et al., 2024a) | 200 | 24.70 | 0.7085 | 0.3018 | 0.2288 | 5.9122 | 65.78 | 0.6221 | 0.6178 | 128.51 |
| | DiffBIR (Lin et al., 2025) | 50 | 24.75 | 0.6567 | 0.3636 | 0.2312 | 5.5346 | 64.98 | 0.6246 | 0.6463 | 128.99 |
| | SeeSR (Wu et al., 2024b) | 50 | 25.18 | 0.7216 | 0.3009 | 0.2223 | 5.4081 | 69.77 | 0.6442 | 0.6612 | 125.55 |
| | PASD (Yang et al., 2025) | 20 | 25.21 | 0.6798 | 0.3380 | 0.2260 | 5.4137 | 68.75 | 0.6487 | 0.6620 | 124.29 |
| | SUPIR (Yu et al., 2024) | 50 | 23.61 | 0.6606 | 0.3589 | 0.2492 | 5.8877 | 63.21 | 0.5895 | 0.6709 | 128.35 |
| | OSEDiff (Wu et al., 2024a) | 1 | 25.15 | 0.7341 | 0.2921 | 0.2128 | 5.6476 | 69.09 | 0.6326 | 0.6693 | 123.49 |
| | AdcSR (Chen et al., 2025) | 1 | 25.47 | 0.7301 | 0.2885 | 0.2128 | 5.3477 | 69.90 | 0.6353 | 0.6730 | 118.41 |
| | PiSA-SR (Sun et al., 2025) | 1 | 25.50 | 0.7418 | 0.2672 | **0.2044** | 5.5046 | 70.15 | 0.6551 | 0.6696 | 124.09 |
| | TSD-SR (Dong et al., 2025) | 1 | 24.81 | 0.7172 | 0.2743 | 0.2105 | **5.1266** | **71.18** | 0.6346 | **0.7160** | **114.45** |
| | InvSR (Yue et al., 2025) | 1 | 24.30 | 0.7145 | 0.2775 | 0.2060 | 5.7168 | 67.31 | **0.6572** | 0.6734 | 129.52 |
| | CCSR (Sun et al., 2024) | 1 | 25.92 | 0.7485 | 0.2799 | 0.2122 | 5.7324 | 69.18 | 0.6398 | 0.6336 | 122.98 |
| | ResShift (Yue et al., 2023) | 15 | 26.31 | 0.7421 | 0.3421 | 0.2498 | 7.2365 | 58.43 | 0.5285 | 0.5442 | 141.71 |
| | SinSR (Wang et al., 2024b) | 1 | 26.28 | 0.7347 | 0.3188 | 0.2353 | 6.2872 | 60.80 | 0.5385 | 0.6122 | 135.93 |
| | CTMSR (You et al., 2025) | 1 | 25.98 | 0.7546 | 0.2897 | 0.2208 | 5.5546 | 64.26 | 0.5270 | 0.6318 | 135.35 |
| | RSD (**Ours**) | 1 | 25.61 | 0.7420 | 0.2675 | 0.2205 | 5.7500 | 66.02 | 0.5930 | 0.6793 | 138.23 |

# E    PERFORMANCE-EFFICIENCY TRADE-OFF FOR RSD AND RECENT STATE-OF-THE-ART ONE-STEP DIFFUSION SR METHODS

In this section, we discuss the comparison between RSD and very recent SOTA one-step diffusion SR methods - CTMSR (You et al., 2025) and T2I-based SR models, including PiSA-SR (Sun et al., 2025), TSD-SR (Dong et al., 2025), AdcSR (Chen et al., 2025), InvSR (Yue et al., 2025) and CCSR (Sun et al., 2024). We support the comparison with visual results of these models in Figure 5 for the RealLR200 dataset (Wu et al., 2024b) and Figure 6 for the RealLQ250 dataset (Ai et al., 2024), respectively.

## E.1    COMPARISON WITH CTMSR

**CTMSR method**. CTMSR proposed a distillation-free method for one-step diffusion SR, which is based on consistency training (Song et al., 2023; Song & Dhariwal, 2024). Their training scheme is split into two stages.

**Stage 1**. In the first stage, they formulate the ResShift forward stochastic diffusion process in Equation (1) as the deterministic trajectory of PF-ODE (Song et al., 2021b); see Equations 8 and 9 in the CTMSR paper. They trained the respective consistency model with 500k iterations using the proposed PF-ODE trajectories and the consistency loss $\mathcal{L}_{CT}$ according to Equation 10 in the CTMSR paper.

**Stage 2**. After the first stage, CTMSR additionally optimizes the model with the proposed Distribution Trajectory Matching objective. Its idea is to minimize the Distribution Trajectory Distance ($\mathcal{L}_{DTD}$ in Equations 15 and 16 in the CTMSR paper) between the end points of the real PF-ODE trajectory, which starts from the real HR images, and the fake PF-ODE trajectory, which starts from the predicted fake HR images; see Equations 11, 12, 13, 14 in the CTMSR paper. Computation of the gradient $\nabla_\theta \mathcal{L}_{DTD}$ with respect to the original CTMSR parameters $\theta$ requires calculating the U-Net Jacobian term. Inspired by SDS (Poole et al., 2023) and VSD (Wang et al., 2023c), CTMSR omits this U-Net Jacobian term. The authors optimized the CTMSR model using the gradients $\nabla_\theta \mathcal{L}_{CT} + \nabla_\theta \mathcal{L}_{DTD}$ with additional 2k iterations.

CTMSR uses the same training scheme on the ImageNet dataset with Real-ESRGAN degradations, which is detailed in Section 4.1, as ResShift, SinSR and RSD. Thus, CTMSR can be fairly comparable with these models.

**Perceptual-fidelity comparison**. According to the quantitative results in Tables 1, 3, 13, and 14, RSD has stable improvement over CTMSR for **all real-world datasets (RealSR, RealSet65, RealSR and DRealSR $512 \times 512$ crops, DRealSR, RealLR200, RealLQ250) in most perceptual metrics (LPIPS, DISTS, CLIPIQA, MUSIQ, MANIQA)**. We observe the most notable gaps between the perceptual quality of CTMSR and RSD on the following datasets:

1. RealSR and DRealSR $512 \times 512$ crops in Table 3 - improvement in MANIQA in 0.0660 and 0.0764, respectively, and in CLIPIQA in 0.0475 and 0.0522, respectively.

2. Full-size DRealSR in Table 13 - improvement in MUSIQ in 5.134 and MANIQA in 0.0612, respectively.

3. RealLR200 and RealLQ250 in Table 14 - improvement in MANIQA in 0.0523 and 0.0696, respectively, and in CLIPIQA in 0.0397 and 0.0551, respectively.

These results highlight the strong competitive perceptual performance of RSD among one-step diffusion SR models using the similar UNet architecture with Swin Transformer blocks (Liu et al., 2021) - ResShift, SinSR and CTMSR. However, CTMSR sometimes has better NIQE values and also achieves slightly better CLIPIQA and MUSIQ on the synthetic ImageNet-Test dataset from Table 2. We note that NIQE (Mittal et al., 2013) has been shown to have a worse correlation with human preference compared to recent IQA measures, including MUSIQ, MANIQA, and CLIPIQA, as evident in (Wang et al., 2023b, Tables 1 and 5), (Ke et al., 2021, Table 1), (Yang et al., 2022, Table 3). CTMSR also achieves better fidelity measures (PSNR and SSIM) compared to RSD, which are close to the results of SinSR. Unfortunately, this results in the blur problem of CTMSR, which is similar to SinSR behavior and is evident in Figure 3; see the severely blurred details of the roof of the house, broccoli and bear in Figure 3.

**Complexity comparison**. The CTMSR can be fairly compared to the RSD in computational complexity due to the similar architecture following ResShift. For the inference complexity, we evaluated the pre-trained CTMSR model using its official implementation listed in Table 12 on the same setup as RSD in Table 4. According to Table 4, CTMSR has a similar number of parameters as ResShift, SinSR, and RSD (172 millions for CTMSR and 174 millions for RSD) and a similar inference time per LR image with resolution $64 \times 64$. However, we also found that CTMSR requires $> 1.5$ GPU memory during inference compared to RSD. As the distillation-free method, CTMSR states that ResShift and its distilled version, SinSR, are limited in two points:

1. **Considerable training costs**. Training SinSR requires the training of the teacher ResShift model and the student SinSR model, while CTMSR is able to train one-step diffusion SR model without an additional distillation stage.

2. **Limitations of the teacher model**. The performance of the student model is limited with the performance of the teacher model.

In Appendices J and G, we agree with the second statement. To verify the first statement, we trained CTMSR with 500k iterations for the first stage and additional 2k iterations for the second stage using their official training code on recommended GPUs (4 NVIDIA A100 GPUs). Following the pre-trained ResShift model, which we used for the distillation with RSD and was used for the distillation with SinSR, we also trained the ResShift model with 300k iterations using its official training code on the same 4 A100 GPUs to compare the CTMSR training time and the total training time of ResShift and RSD. The results are given in Table 18. Surprisingly, we found that the total training time for ResShift and its distillation with our RSD requires **less training time** than the training time for the distillation-free CTMSR method using the same resources. The training efficiency of the RSD model is supported by its simulation-free property, which is detailed in Appendix C. Compared to the training of the original teacher ResShift model, its distillation with our RSD method requires only $\approx 15\%$ training time of ResShift, which leads to faster convergence than the distillation-free CTMSR even if we count the training time of ResShift. As noted in Appendix F, the training time of our RSD can be further halved using $K = 1$ without sacrificing quality. These results highlight the strong computational efficiency of RSD compared to CTMSR both in training and inference.

Table 18: Training and inference complexity for RSD, ResShift (Yue et al., 2023) and CTMSR (You et al., 2025). All models are trained on the same 4 NVIDIA A100 GPUs using the respective official training code and tested with an LR image of size $64 \times 64$ for SR factor $\times 4$. The inference is done on an NVIDIA A100 GPU. The best values are highligted in **bold**.

| Methods | ResShift | RSD (**Ours**) | RSD with ResShift training | CTMSR |
|---|---|---|---|---|
| Inference Step (NFE) | 15 | **1** | **1** | **1** |
| Inference Time (s) | 0.643 | **0.059** | **0.059** | **0.059** |
| # Total Param (M) | **174** | **174** | **174** | 172 |
| Maximum GPU memory (MB) | 1167 | **539** | **539** | 904 |
| Training time (hours) | 36 | 5 | 41 | 58 |

### E.2 COMPARISON WITH PISA-SR, TSD-SR, ADCSR, INVSR AND CCSR

PiSA-SR (Sun et al., 2025), TSD-SR (Dong et al., 2025), AdcSR (Chen et al., 2025), InvSR (Yue et al., 2025), and CCSR (Sun et al., 2024) are recent SOTA T2I-based SR models, which attempt to resolve the limitations of the OSEDiff model in different aspects.

**Adjustable perception-distortion trade-off and slow training convergence**. OSEDiff does not provide perception-distortion control without re-training, while the training requires 24 hours on 4 NVIDIA A100 GPUs according to Section 4.1 in OSEDiff paper. PiSA-SR proposed a decoupled training approach to train pixel and semantic level LoRA modules (Hu et al., 2022), which allows to adjust the perception-distortion trade-off by different pixel-semantic guidance scales (Ho & Salimans, 2021) during inference without the need of re-training. PiSA-SR uses $\ell_2$ loss for the training of the LoRA module of pixel-level regression and CSD loss (Ho & Salimans, 2021) combined with the LPIPS loss for the training of the LoRA module of semantic level. The CSD loss is computed using the pre-trained Stable Diffusion 2.1-base model and does not require an additional fake model used in OSEDiff, leading to faster training (Ma et al., 2025) according to Figure 9 in the PiSA-SR paper.

**Limitations of the VSD objective**. As shown in the TSD-SR paper, the VSD objective used in OSEDiff has two limitations. The first limitation is that the guidance of the teacher is unreliable

in scenarios where the initial SR outputs are suboptimal, as visualized in Figure 3 of TSD-SR. To solve this problem, TSD-SR proposed Target Score Matching, which aligns the predictions made by the teacher model on both synthetic and HR latents. The second limitation is that the matching of the score functions predicted by the teacher model and the LoRA model is inconsistent across different timesteps, which is shown in Figure 5 of TSD-SR. To address this issue, TSD-SR proposed the Distribution-Aware Sampling Module, which accumulates optimization gradients for earlier timestep samples in a single iteration, enabling the backpropagation of more gradients focused on detail optimization. TSD-SR initialized all training models from the Stable Diffusion 3 model (Esser et al., 2024).

**Large computational costs of T2I-based SR models**. As observed in AdcSR, the complexity of OSEDiff in terms of parameter number and inference time can still be too high for real deployments, especially on resource-limited edge devices. To reduce the complexity of OSEDiff while maintaining its high perceptual quality, AdcSR proposed an adversarial diffusion compression framework to OSEDiff. The idea of the framework is to train a smaller network after removing unnecessary OSEDiff modules and pruning the remaining modules. The training of AdcSR consists of two stages: 1) pretraining channel-pruned VAE decoder; 2) usage of knowledge distillation for OSEDiff with adversarial loss to train a smaller student model.

**Extension to multistep models**. The OSEDiff approach is developed only for the one-step diffusion model, which limits its generation capacity and flexibility for varying perception-distortion requirements. InvSR and CCSR proposed different approaches to enable multistep diffusion models without retraining. InvSR introduces a trainable noise prediction network and reformulates the SR problem as diffusion inversion (Chihaoui et al., 2024). The noise predictor is trained to estimate the noise maps for multiple pre-selected steps via time embedding. To enable arbitrary-step inversion for the inference, InvSR uses the noise map prediction for the initialization the reverse sampling process, where the starting timestep can be freely chosen during inference, resulting in perception-distortion trade-off. CCSR achieves multistep diffusion modeling with another idea. Inspired by the StableSR (Wang et al., 2024a) perception-distortion analysis depending on the diffusion reverse time step in Figure 2 of CCSR, it proposed to disentangle the SR process into structure generation and detail enhancement by GAN and DM, respectively.

**Quantitative comparison**. In Tables 13, 14, 15, 16, 17, we observe that our RSD combines the high fidelity of relatively small models (ResShift, SinSR, CTMSR) and the good perceptual quality of T2I-based SR models (TSD-SR, PiSA-SR, InvSR, CCSR, AdcSR). TSD-SR, PiSA-SR, InvSR, AdcSR and CCSR further develop T2I-based SR models to improve perceptual and fidelity quality compared to OSEDiff. Compared to these methods, RSD achieves mostly better fidelity consistency with HR images, which is evident by PSNR and SSIM metrics, with yet competitive perceptual metrics (LPIPS, CLIPIQA, MUSIQ).

**Complexity comparison**. Despite the good perceptual performance of the TSD-SR, PiSA-SR, AdcSR, CCSR, and InvSR models, these T2I-based models require more computational costs compared to the other one-step T2I-based SR model, OSEDiff, and much more computational costs compared to RSD. In Table 4, we highlight that most T2I-based SR models require much more GPU memory for inference and much longer training time compared to RSD. This analysis supports our claim that RSD aims to compromise between fidelity, perceptual quality, and computational efficiency.

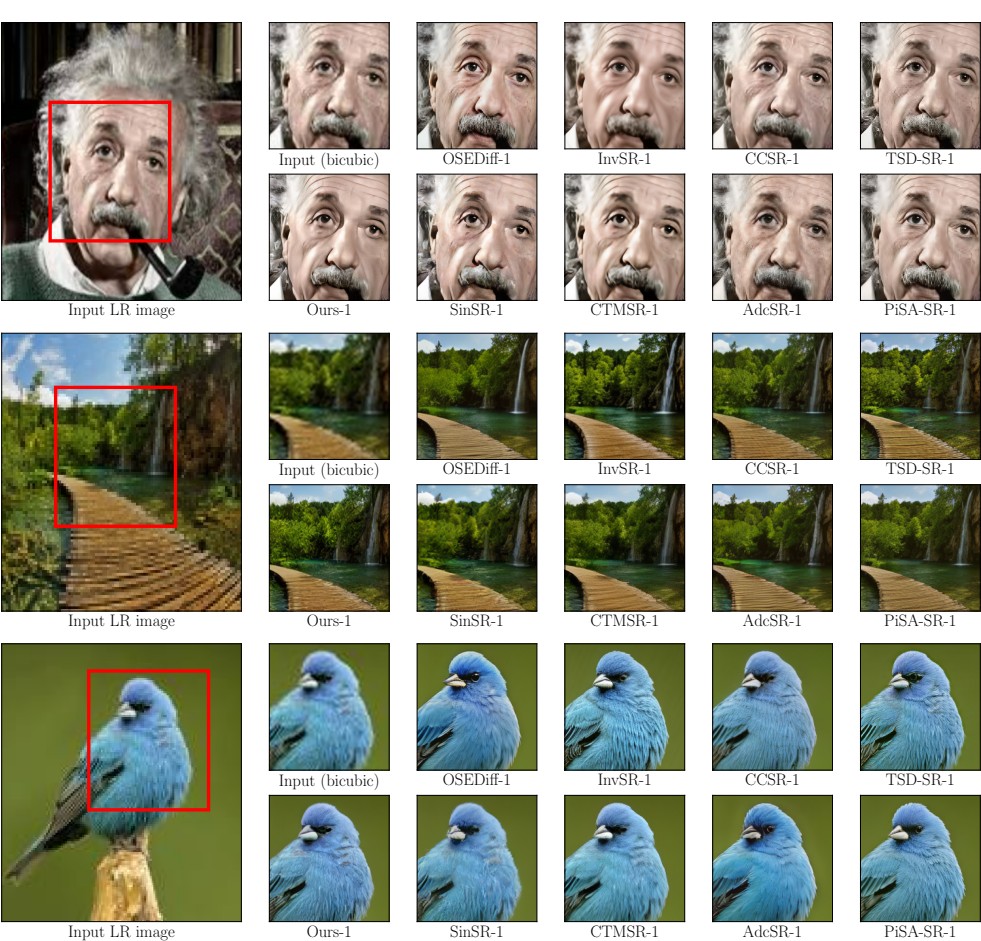

Figure 5: Visual results of recent one-step diffusion SR models (RSD, SinSR, CTMSR, OSEDiff, AdcSR, CCSR, InvSR, PiSA-SR, TSD-SR) on full-size images from RealLR200 (Wu et al., 2024b). Please zoom in for a better view.

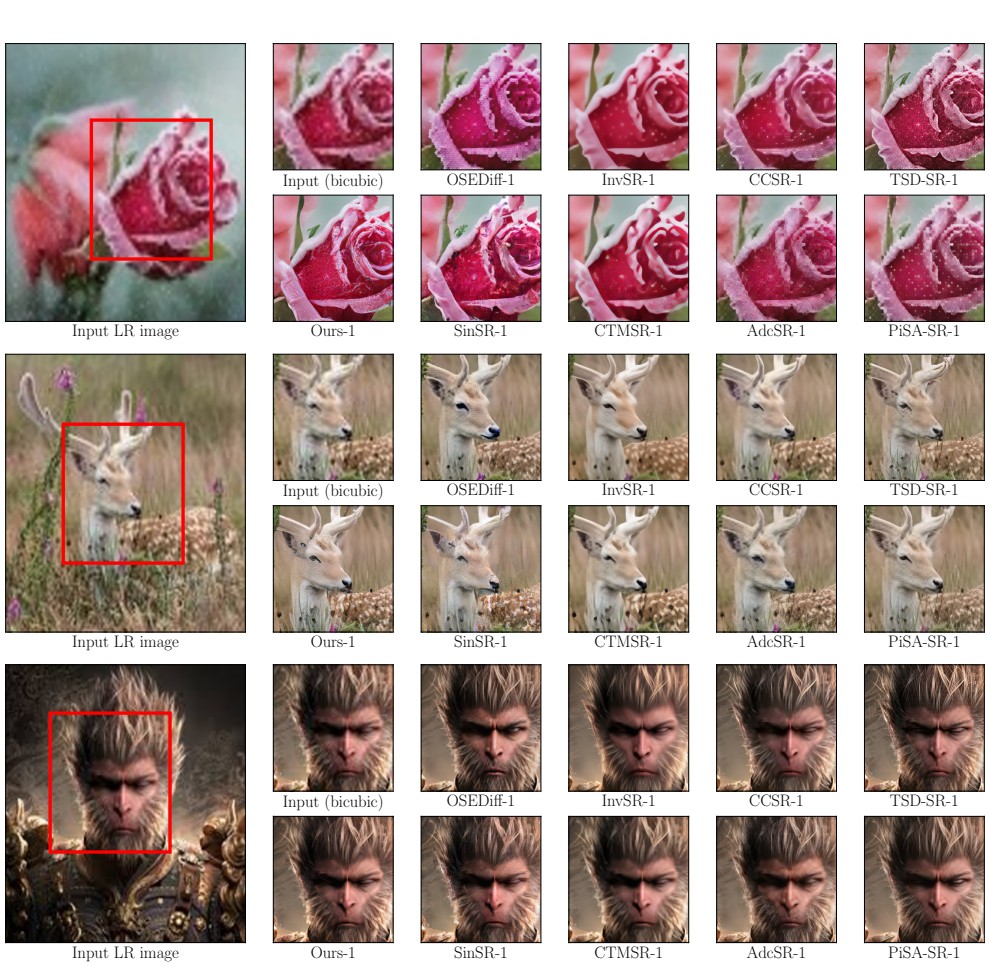

Figure 6: Visual results of recent one-step diffusion SR models (RSD, SinSR, CTMSR, OSEDiff, AdcSR, CCSR, InvSR, PiSA-SR, TSD-SR) on full-size images from RealLQ250 (Ai et al., 2024). Please zoom in for a better view.

# F    ADDITIONAL ABLATION STUDIES

**Ablation on the hyperparameter $K$.** To assess the sensitivity of RSD to the hyperparameter $K$ in Algorithm 1, the number of fake ResShift updates per student update, we performed experiments with the final RSD configuration, which is reported in Tables 1, 2 and 3, with additional supervised losses for $K \in \{1, 3, 5, 10\}$. We evaluated the trained models on both the full-size RealSR dataset (Table 19, as in Section 4.3) and the ImageNet-Test dataset (Table 20, following Table 2). Across both datasets, all choices of $K$ yield very similar performance: $K = 1$ slightly improves or matches the metrics of $K = 5$ while roughly halving the training time due to fewer fake-model updates. These results indicate that, in the presence of additional ground-truth losses, RSD is largely insensitive to the exact choice of $K$ in this range, so $K$ can be used to optimize computation at the cost of only minor performance changes. We used $K = 5$ to follow the DMD2 strategy for the number of updates of the fake model per student update (Yin et al., 2024a); see Figure 9 in Appendix C of DMD2 for the analysis of the impact of $K$ on the training stability for image generation problems. Our results show that RSD training with $K = 1$ and supervised losses can also be beneficial for Real-ISR problems while not breaking the good performance of $K = 5$.

To isolate the effect of $K$ on optimization stability, we further repeated the ablation in a *distill only* setup ((**Ours**, distill only) in Tables 1 and 2) without supervised losses. Figure 7 shows the convergence of PSNR and CLIPIQA on the ImageNet-Test dataset for $K \in \{1, 3, 5, 10\}$ in this case. For $K = 1$, the training dynamics become highly unstable, with large oscillations and clearly degraded final metrics, whereas configurations with $K \geq 3$ converge smoothly to similar quality levels. This suggests that supervised losses play a stabilizing role when using small $K$, and that $K = 5$ remains a robust choice in more challenging or purely distillation-based settings, while $K = 1$ is a viable and more efficient alternative in the supervised Real-ISR configuration used in the main experiments. This behavior was also reported in Figure 9 of DMD2.

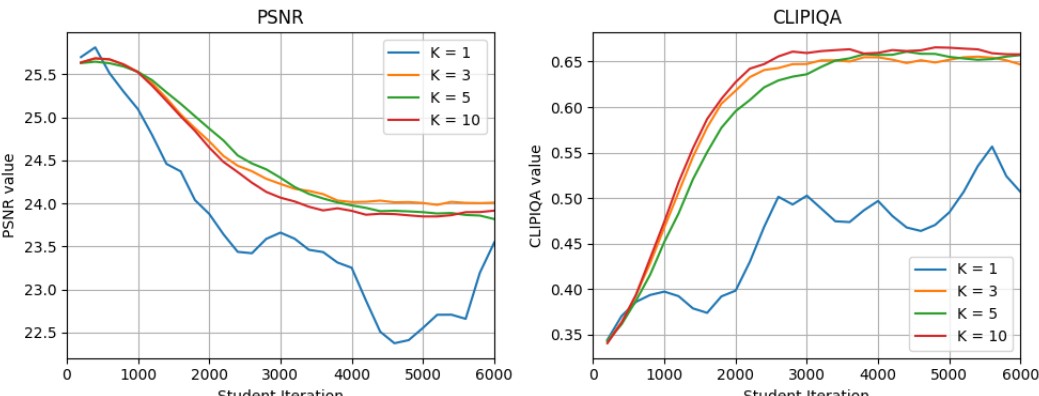

Figure 7: Convergence of PSNR and CLIPIQA on the ImageNet-Test dataset for $K \in \{1, 3, 5, 10\}$ when training *distill only* RSD. For $K = 1$, the optimization becomes unstable and fails to reach the quality of configurations with $K \geq 3$.

Table 19: Ablation on the hyperparameter $K$ on the RealSR validation set. All runs use the final RSD configuration with supervised losses.

| $K$ | PSNR↑ | SSIM↑ | LPIPS↓ | CLIPIQA↑ | MUSIQ↑ |
|---|---|---|---|---|---|
| 1 | 25.99 | 0.756 | 0.2713 | 0.7159 | 66.247 |
| 3 | 25.86 | 0.752 | 0.2701 | 0.7093 | 66.140 |
| 5 | 25.91 | 0.754 | 0.2726 | 0.7060 | 65.860 |
| 10 | 26.27 | 0.749 | 0.2732 | 0.7135 | 65.233 |

Table 20: Ablation on the hyperparameter $K$ on the ImageNet-Test dataset. All runs use the final RSD configuration with supervised losses.

| $K$ | PSNR↑ | SSIM↑ | LPIPS↓ | CLIPIQA↑ | MUSIQ↑ |
|---|---|---|---|---|---|
| 1 | 24.28 | 0.657 | 0.196 | 0.697 | 59.499 |
| 3 | 24.11 | 0.644 | 0.191 | 0.675 | 59.535 |
| 5 | 24.31 | 0.657 | 0.193 | 0.681 | 58.947 |
| 10 | 24.01 | 0.639 | 0.193 | 0.671 | 59.110 |

**Ablation on supervised losses.** We support the quantitative results of Table 6 on the RealSR dataset (Cai et al., 2019) with visual results in Figure 8. Without any supervised losses, RSD produces images with excessively sharp features, which are not related to the reference image; see the bottom image of Figure 8. As shown in Table 6, the LPIPS loss improves the correspondence to the reference image in terms of reference-based metrics (PSNR and LPIPS), but also spoils no-reference metrics (CLIPIQA and MUSIQ). This is visualized as smoother skin in the top image of Figure 8. Incorporating the GAN loss leads to the opposite effect and increases the number of details, such as skin textures. Adding both LPIPS and GAN losses allows us to balance between the number of excessive details and blurriness. For example, in the top image of Figure 8, both supervised losses improve the realism of the female eye.

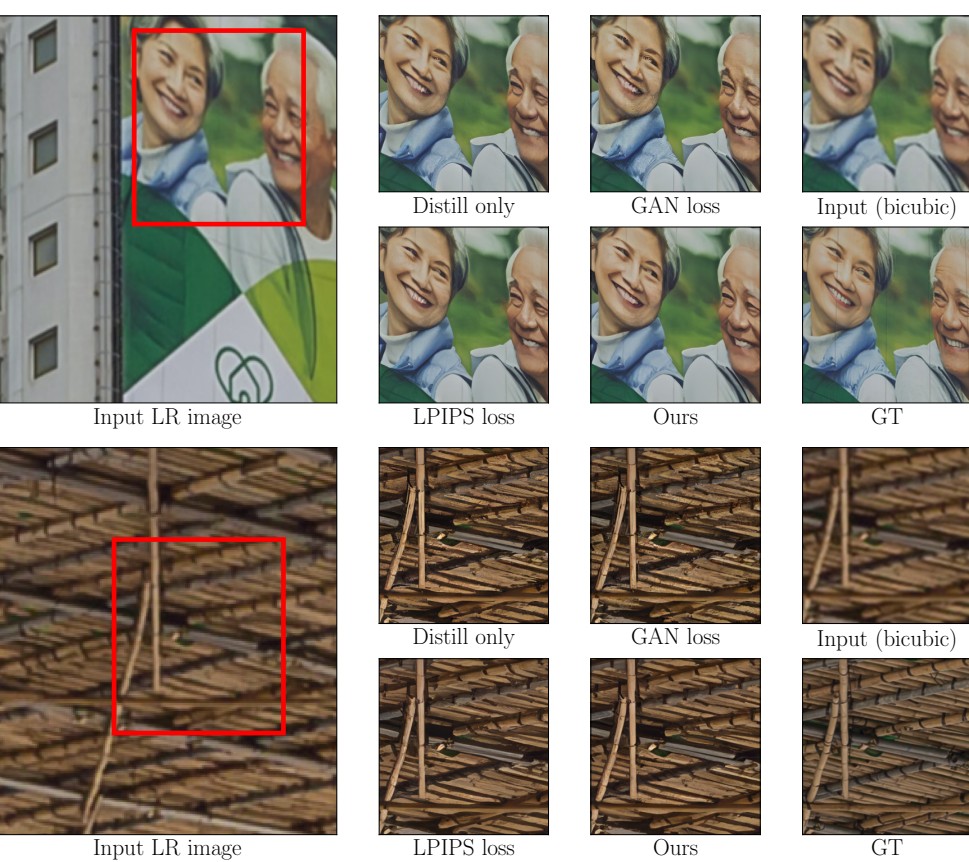

Figure 8: Effect of incorporating supervised losses on RealSR (Cai et al., 2019). Please zoom in for a better view.

# G RESULTS OF RSD TRAINED ON BIGGER RESOLUTION

To compare the performance of the RSD, SinSR, and ResShift models for training on high resolution images, we followed the training setup of OSEDiff, which was trained on $512 \times 512$ HR images randomly cropped from LSDIR (Li et al., 2023) with LR images generated via the Real-ESRGAN degradations with the $\times 4$ SR factor. Since the original ResShift model was trained only on $256 \times 256$ HR images, we first trained the ResShift model on $512 \times 512$ HR random crops from LSDIR (Li et al., 2023) for 300k iterations using the source training ResShift code and then distilled it with the RSD and SinSR methods. For a fair comparison, we used the same hyperparameters for RSD and SinSR, which were used for their training on $256 \times 256$ HR images in Table 1, Table 2, Table 3.

We evaluated trained ResShift, SinSR and RSD models on full-size images from RealSR (Cai et al., 2019) (left part of Table 1) and provided quantitative results in Table 21, where we also provide results of Real-ESRGAN, BSRGAN, SUPIR and OSEDiff from Table 15. We provide visual results of those models in Figure 9.

**Comparison with SinSR.** We observe that the RSD achieves better perceptual results, especially in CLIPIQA and MUSIQ, with competitive PSNR and SSIM compared to SinSR. Although ResShift and SinSR have better fidelity metrics, the gap between the visual quality of these models compared with the RSD can be observed in image details, which are sharper for RSD (compare the jackets in the top of Figure 9, and trees in the middle of Figure 9).

**Comparison with OSEDiff.** Compared to OSEDiff, RSD trained on the same images from the LSDIR dataset (Li et al., 2023) using HR crops of the same resolution $512 \times 512$ achieves better reference metrics (PSNR, SSIM, LPIPS), but worse no-reference metrics (CLIPIQA, MUSIQ). This is evident in Figure 9, where both OSEDiff and SUPIR models provide not relevant to the LR image details (see non-existing inscriptions in the bottom of Figure 9). We highlight that since we did not finetune hyperparameters of the teacher ResShift model, the quality of RSD is limited by the quality of the ResShift model. The main goal of this study is to show that **RSD achieves better perceptual results compared with SinSR even if trained on images with higher resolutions**. Improving the perceptual image quality of the RSD when trained on images of higher resolutions to make it closer to the visual quality of T2I-based SR models is a promising future work.

**The performance of the RSD model closely mirrors that of its teacher model**. Notably, the RSD model trained at a resolution of $256 \times 256$ demonstrates better performance on no-reference image quality metrics, such as CLIPIQA and MUSIQ (see Table 1). In contrast, the RSD model trained at $512 \times 512$ resolution achieves better results on reference-based metrics, including PSNR, SSIM, and LPIPS (see Table 21). We hypothesize that the observed decline in no-reference metrics, alongside the improvement in reference-based metrics at higher resolutions, is primarily attributed to the behavior of the teacher model, ResShift. Specifically, the ResShift model trained on $256 \times 256$ images yields higher scores on no-reference perceptual quality metrics, whereas the model trained on $512 \times 512$ images performs better on reference-based metrics. The RSD model exhibits the same pattern, which explains the observed trade-off between the two types of evaluation metrics across resolutions.

Table 21: Results on full-size images from RealSR (Cai et al., 2019). ResShift, SinSR and RSD were trained on $512 \times 512$ HR random crops from LSDIR (Li et al., 2023). The best and second best results are highlighted in **bold** and underline.

| Methods | NFE | PSNR↑ | SSIM↑ | LPIPS↓ | CLIPIQA↑ | MUSIQ↑ |
|---|---|---|---|---|---|---|
| Real-ESRGAN (Wang et al., 2021) | 1 | 25.85 | 0.773 | 0.273 | 0.4898 | 59.678 |
| BSRGAN (Zhang et al., 2021) | 1 | 26.51 | 0.775 | 0.269 | 0.5439 | 63.586 |
| SUPIR (Yu et al., 2024) | 50 | 24.38 | 0.698 | 0.331 | 0.5449 | 63.679 |
| OSEDiff (Wu et al., 2024a) | 1 | 25.25 | 0.737 | 0.299 | **0.6772** | **67.602** |
| ResShift (Yue et al., 2023) | 15 | **27.53** | **0.790** | 0.277 | 0.4988 | 58.034 |
| SinSR (Wang et al., 2024b) | 1 | 27.27 | 0.780 | 0.268 | 0.5503 | 59.478 |
| RSD (**Ours**) | 1 | 26.89 | 0.773 | **0.260** | 0.6103 | 64.987 |

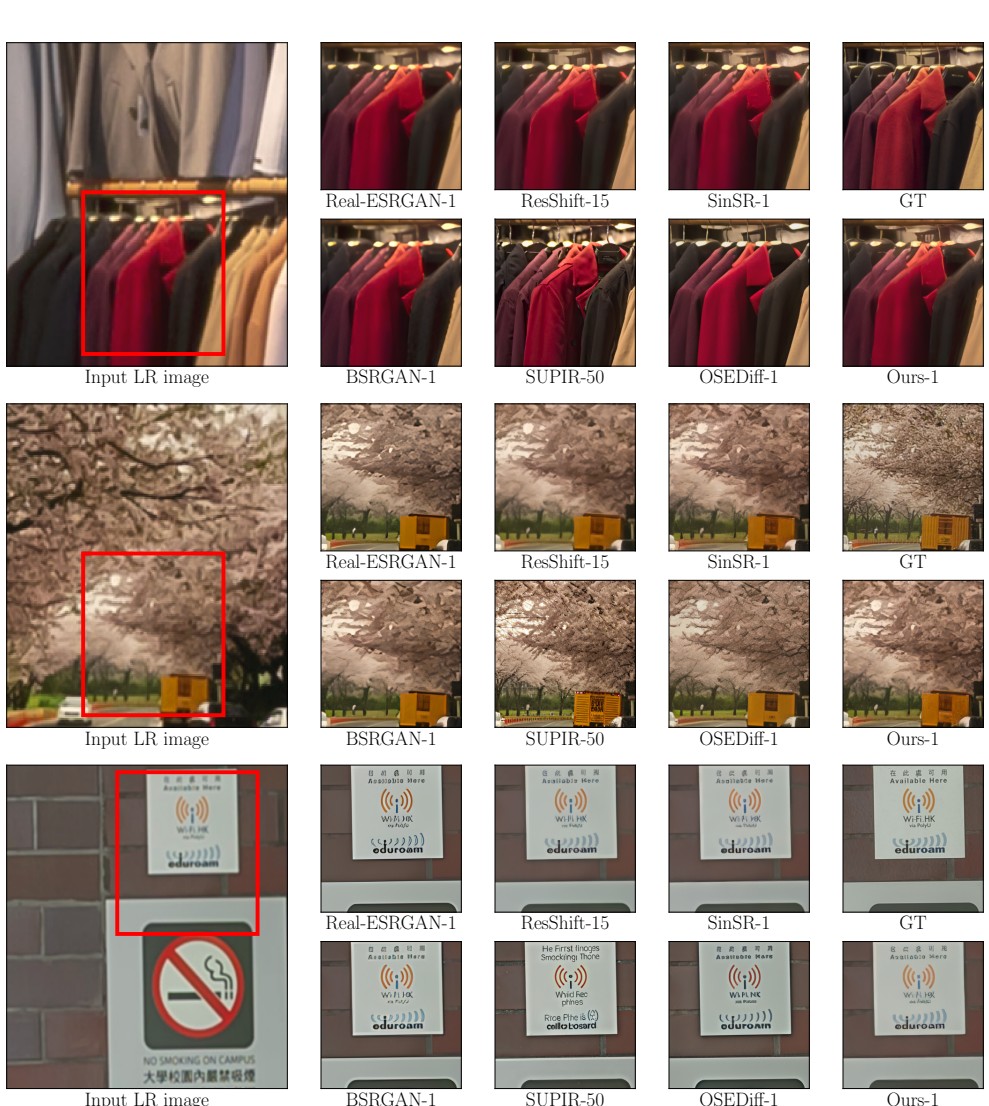

Figure 9: Visual results of RSD, ResShift, and SinSR models trained on $512 \times 512$ HR images from LSDIR dataset (Li et al., 2023) and other baselines (Real-ESRGAN, BSRGAN, SUPIR, OSEDiff) on full-size images from RealSR (Yue et al., 2023). Please zoom in for a better view.

## H    COMPARISON WITH ADDSR

It may be seem that our primary distillation loss $\mathcal{L}_\theta$ (7) is similar to the SDS loss adopted in ADD (Sauer et al., 2025) $\mathcal{L}_{distill}$ and AddSR (Xie et al., 2024) (Appendix A, (Sauer et al., 2025) and $\mathcal{L}_{ta-dis}$, Equation 1, (Xie et al., 2024)). However, we show that this similarity is only on the surface.

**Conceptual difference in objective functions**. In our work, we propose the RSD loss (20), and it is fundamentally different from SDS in both its formulation and practical implications. RSD introduces an auxiliary model to more accurately estimate the score function of the model distribution. This allows for a tighter and lower-variance approximation of the true KL gradient. Moreover, instead of treating each timestep independently as in SDS and VSD (i.e., using marginal KL divergences over $x_t$), our RSD loss is formulated over **the entire trajectory** $x_{0:T}$, leading to a more holistic distillation of the teacher's reverse process.

To facilitate a clear comparison, we summarize the key differences between the loss objectives used in SDS and our proposed RSD in Table 22. We denote by $p^*$ the reverse process of the teacher ResShift model and by $p$ the reverse process of ResShift trained on generator data.

Table 22: Comparison of distillation objectives between SDS (Poole et al., 2023), ADD (Sauer et al., 2025), AddSR (Xie et al., 2024) and RSD

| Methods | Objective Function |
|---|---|
| SDS (Poole et al., 2023), ADD (Xie et al., 2024), AddSR (Xie et al., 2024) | $\mathbb{E}_{p(y_0)}\left[\sum_{t=1}^{T} w_t \mathcal{D}_{\text{KL}}\left(p(x_t\|y_0)\|\|p^*(x_t\|y_0)\right)\right]$ |
| RSD (**Ours**, distill-only) | $\mathbb{E}_{p(y_0)}\left[\mathcal{D}_{\text{KL}}\left(p(x_{0:T}\|y_0)\,\|\,p^*(x_{0:T}\|y_0)\right)\right]$ |

**Practical difference**.  In addition to theoretical differences between RSD and ADD objectives discussed above, we list practical differences between implementations of RSD and AddSR for real-world SR.

**1. Objective implementation**. The implementation of objective in Eq. 1 in AddSR (Xie et al., 2024) is different from RSD objective in Eq. (11) in many aspects:

1. **RSD does not have any hyperparameters in the distillation loss**. The distillation loss of AddSR, $\mathcal{L}_{ta-dis}$ (Equation 1, (Xie et al., 2024)), requires a weighting function $d(s,t)$ (Equation 3, (Xie et al., 2024)), which is defined by two hyperparameters, $\mu$ and $\nu$. The choice $\mu$ and $\nu$ is based solely on empirical analysis of performance results, as shown in Table 7 and Table 8 of AddSR. In contrast, RSD loss in Eq. (8) relies only on weights $w_t$, which are used for the training of the ResShift model (Eq. 8 in (Yue et al., 2023)). These weights are derived from the theory of DDPM (Ho et al., 2020) and in practice are omitted by ResShift and RSD following the conclusion of DDPM (see Appendix I).

2. **Different supervised losses**. The adversarial loss of AddSR, $\mathcal{L}_{ta-dis}$, follows the hinge loss used in ADD (Sauer et al., 2025). We follow the adversarial loss of DMD2 (Yin et al., 2024a) and use the standard non-saturating loss. We also use LPIPS loss following OSEDiff (Wu et al., 2024a), while AddSR omits it.

3. **Fake model**. Contrary to AddSR, the objective of RSD involves a trainable fake model.

**Architecture**. The major architectural differences are as follows:

1. **RSD does not have any networks related to text-to-image models**. The architecture of generator and fake model in RSD is a UNet model (Ronneberger et al., 2015) following ResShift. We avoid ControlNet (Zhang et al., 2023) and other models used in AddSR.

2. **The sizes of the AddSR and RSD architectures differ by 1 order of magnitude**. In total, the architecture of AddSR requires 2.28B parameters, while RSD requires 174M parameters.

**Empirical results**. We show the comparison between results of 1-step AddSR and RSD in Table 23. It shows that RSD outperforms AddSR in most fidelity and perceptual metrics while having $\times 10$ much fewer parameters.

Table 23: Quantitative results of AddSR and RSD models on crops $512 \times 512$ from (Wang et al., 2024a). The best results are highlighted in **bold**.

| Datasets | Methods | PSNR↑ | SSIM↑ | LPIPS↓ | DISTS↓ | NIQE↓ | MUSIQ↑ | MANIQA↑ | CLIPIQA↑ |
|---|---|---|---|---|---|---|---|---|---|
| DIV2K-Val | AddSR (Xie et al., 2024) | 23.26 | 0.5902 | 0.3623 | 0.2123 | **4.7610** | 63.39 | 0.5657 | 0.5734 |
| | RSD (**Ours**) | **23.91** | **0.6042** | **0.2857** | **0.1940** | 5.1987 | **68.05** | **0.5937** | **0.6967** |
| DrealSR | AddSR (Xie et al., 2024) | **27.77** | **0.7722** | 0.3196 | **0.2242** | 6.9321 | 60.85 | 0.5490 | 0.6188 |
| | RSD (**Ours**) | 27.40 | 0.7559 | **0.3042** | 0.2343 | **6.2577** | **62.03** | **0.5625** | **0.7019** |
| RealSR | AddSR (Xie et al., 2024) | 24.79 | 0.7077 | 0.3091 | **0.2191** | **5.5440** | **66.18** | **0.6098** | 0.5722 |
| | RSD (**Ours**) | **25.61** | **0.7420** | **0.2675** | 0.2205 | 5.7500 | 66.02 | 0.5930 | **0.6793** |

# I  DETAILS OF RESSHIFT

As a part of the diffusion model class, ResShift can be described by specifying the forward (degradation) process, the parametrization of the reverse (restoration) process, and the objective for training the reverse process.

**Forward process.** Consider a pair of $(\mathrm{LR}, \mathrm{HR})$ images $(y_0, x_0) \sim p_{\text{data}}(y_0, x_0)$. For a residual $e_0 = y_0 - x_0$, ResShift proposes to transit from $x_0$ to $y_0$ with the Markov chain $\{x_t\}_{t=1}^T$ of length $T$ through the following Gaussian transition distribution:

$$q(x_t|x_{t-1}, y_0) = \mathcal{N}(x_t|x_{t-1} + \alpha_t e_0, \kappa^2 \alpha_t \mathbf{I}), \tag{26}$$

where:

- $\alpha_t = \eta_t - \eta_{t-1}$ for $t > 1$ and $\alpha_1 = \eta_1$ are defined by the shifting sequence $\{\eta_t\}_{t=1}^T$, $\mathbf{I}$ denotes the identity matrix.
- $\kappa$ is a hyper-parameter controlling the noise variance and the shifting sequence $\{\eta_t\}_{t=1}^T$ monotonically increases with the timestep $t$.

The transition distribution (1) leads to analytically tractable marginal distribution of $q(x_t|x_0, y_0)$ at any timestep $t$:

$$q(x_t|x_0, y_0) = \mathcal{N}(x_t|x_0 + \eta_t e_0, \kappa^2 \eta_t \mathbf{I}), t \in [1, T], \tag{27}$$

The shifting sequence satisfies $\eta_1 \approx 0$ and $\eta_T \approx 1$, which guarantee the convergence of marginal distributions of $x_1$ and $x_T$ to approximate distributions of the HR image and the LR image respectively. Notably, the posterior distribution $q(x_{t-1}|x_t, x_0, y_0)$ for the transition distribution (1) is tractable and can be derived using the Bayes's rule:

$$q(x_{t-1}|x_t, x_0, y_0) = \mathcal{N}\left(x_{t-1}\Big|\frac{\eta_{t-1}}{\eta_t}x_t + \frac{\alpha_t}{\eta_t}x_0, \kappa^2\frac{\eta_{t-1}}{\eta_t}\alpha_t\mathbf{I}\right). \tag{28}$$

**Reverse process.** ResShift suggests the construction of the reverse process to estimate the posterior distribution $p(x_0|y_0)$ in the following parametrized form:

$$p_\theta(x_0|y_0) = \int p(x_T|y_0)\prod_{t=1}^T p_\theta(x_{t-1}|x_t, y_0)dx_{1:T} \tag{29}$$

Here $p(x_T|y_0) \approx \mathcal{N}(x_T|y_0, \kappa^2 I)$ and $p_\theta(x_{t-1}|x_t, y_0)$ is the inverse transition kernel from $x_{t-1}$ to $x_t$ with learnable parameters $\theta$. Following DDPM (Ho et al., 2020), ResShift parametrizes this transition kernel with the Gaussian:

$$p_\theta(x_{t-1}|x_t, y_0) = \mathcal{N}(x_{t-1}|\mu_\theta(x_t, y_0, t), \Sigma_\theta(x_t, y_0, t)) \tag{30}$$

**Objective.** To derive the minimization objective for parameters $\theta$, ResShift applies the variational bound estimation on negative log-likelihood for the $p_\theta(x_0|y_0)$, as in DDPM:

$$\min_\theta \mathbb{E}_{(x_0, y_0)} \sum_{t=1}^T \mathbb{E}_{x_t \sim q(x_t|x_0, y_0)}\left[\mathcal{D}_{KL}(q(x_{t-1}|x_t, x_0, y_0)||p_\theta(x_{t-1}|x_t, y_0))\right] \tag{31}$$

Inspired by the tractable formula for the posterior $q(x_{t-1}|x_t, x_0, y_0)$ in (28), ResShift sets the variance parameter $\Sigma_\theta(x_t, y_0, t)$ to be independent of $x_t$ and $y_0$ and reparametrized the parameter $\mu_\theta(x_t, y_0, t)$ as follows:

$$\Sigma_\theta(x_t, y_0, t) = \kappa^2\frac{\eta_{t-1}}{\eta_t}\alpha_t\mathbf{I} \tag{32}$$

$$\mu_\theta(x_t, y_0, t) = \frac{\eta_{t-1}}{\eta_t}x_t + \frac{\alpha_t}{\eta_t}f_\theta(x_t, y_0, t), \tag{33}$$

where $f_\theta$ is a deep neural network with parameter $\theta$, aiming to predict $x_0$. Given the Gaussian form of the distributions $q(x_{t-1}|x_t, x_0, y_0)$ (28) and $p_\theta(x_{t-1}|x_t, y_0)$ (30), the objective (31) simplifies as follows:

$$\min_\theta\left[\sum_{t=1}^T w_t\mathbb{E}_{(x_0, y_0, x_t)}\|f_\theta(x_t, y_0, t) - x_0\|^2\right], \tag{34}$$

where $w_t = \frac{\alpha_t}{2\kappa^2\eta_t\eta_{t-1}}$. Empirically, the omitting the weight $w_t$ leads to a noticeable improvement in performance, which aligns with the conclusion in DDPM.

## J    LIMITATIONS, FAILURE CASES, POTENTIAL SOCIETAL IMPACT

**Limitations and failure cases**. Below, we present failure cases for image restoration for RSD and baselines. Our method may produce images with mistakes since the teacher model is imperfect. However, we stress that T2I-based SR models with rich priors also have such problems. Specifically, in Figure 10 (top), we observe that the teacher model produces an indistinguishable image from the simple bicubic upsampling image. A similar occurs with OSEDiff, while all other methods, including ours, SinSR, SUPIR, and GAN-based models, produce images with visible artifacts. The other typical failure case of diffusion-based methods, ResShift, SinSR, and RSD, includes images with rich background details, which are hard to predict given insufficient contextual information on the LR image. As we show in Figure 10 (bottom), hallucination properties of T2I-based methods, SUPIR and OSEDiff, provide realistic continuation of the road and cars with greater and richer details compared to results of ResShift, SinSR, and RSD. In Figure 11, we show failure cases of considered SR methods on the real-world RealSR benchmark (Cai et al., 2019) with available ground truth images. All methods struggle when running on images with many small details, like computer details and bush patterns. Hallucinations of diffusion-based methods don't coincide with the original HR image in small details. Figure 12 demonstrates several failure cases for real-world SR methods on synthetic low-resolution crops of StableSR (Wang et al., 2024a), which were obtained from DIV2K dataset (Agustsson & Timofte, 2017) using Real-ESRGAN degradations (Wang et al., 2021). Due to the lack of context diffusion-based methods, ResShift, SinSR, and RSD failed to reconstruct details of the squirrel's body and forest, as well as T2I-based SUPIR and OSEDiff methods with richer prior. Their predictions have significant visual differences with ground truth HR images.

**Potential societal impact**. Our proposed distillation method of the diffusion-based image super-resolution model, RSD, presents potential positive and negative societal impacts. On the positive side, the practical effect of the techniques developed to improve the quality and efficiency of the real-world SR model ranges from improving medical imaging for diagnostic purposes and assisting in disaster response to improving remote sensing and autonomous driving performance. However, there are concerns regarding the generation of fake content. Our model is generative and can generate deepfakes for disinformation. But we note that our model was trained using only one dataset, ImageNet (Deng et al., 2009), which is known to be standard in SR research (Wang et al., 2024b; Yue et al., 2023; Gushchin et al., 2025). Thus, we don't expect any high risk for misuse of the trained model as long as the training data does not contain unsafe images.

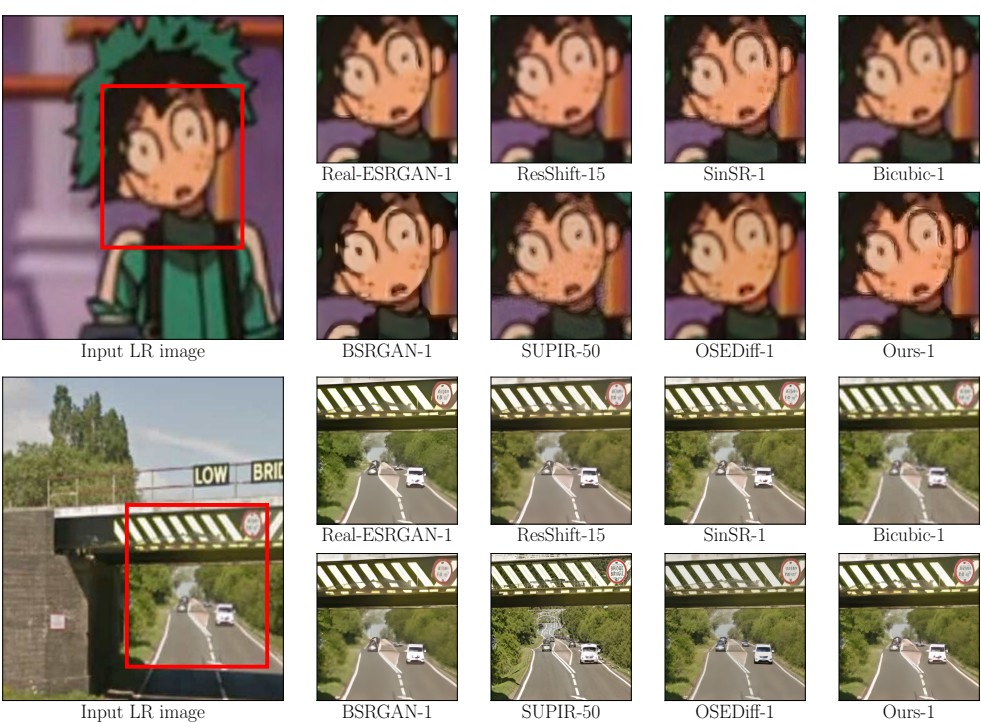

Figure 10: Failure cases on images from RealSet65 (Yue et al., 2023). Please zoom in for a better view.

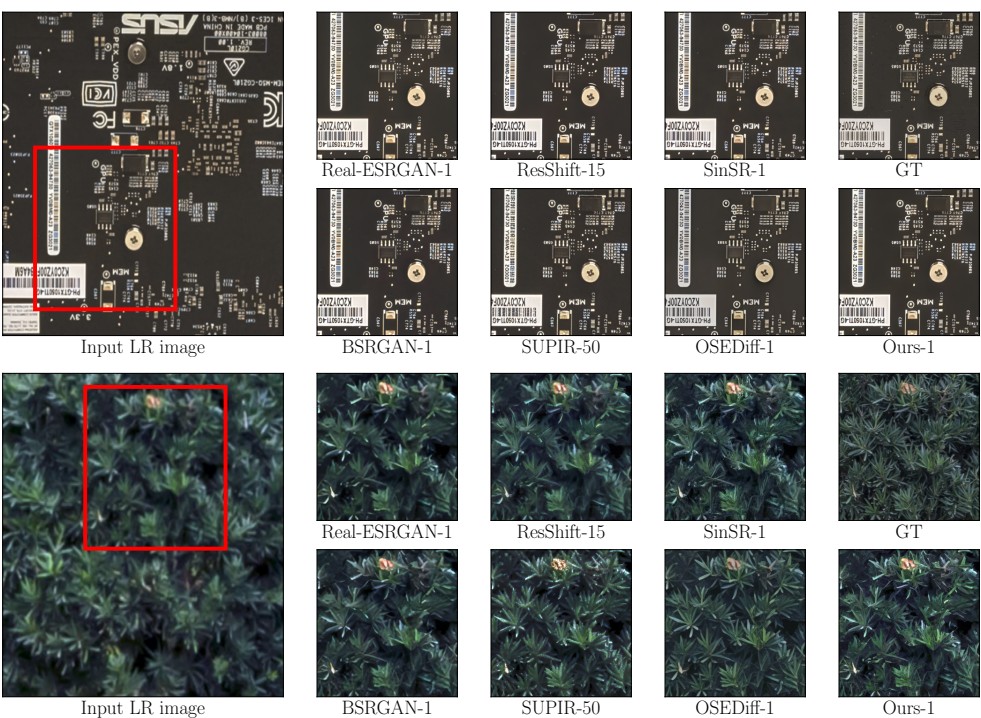

Figure 11: Failure cases on images from RealSR (Yue et al., 2023). Please zoom in for a better view.

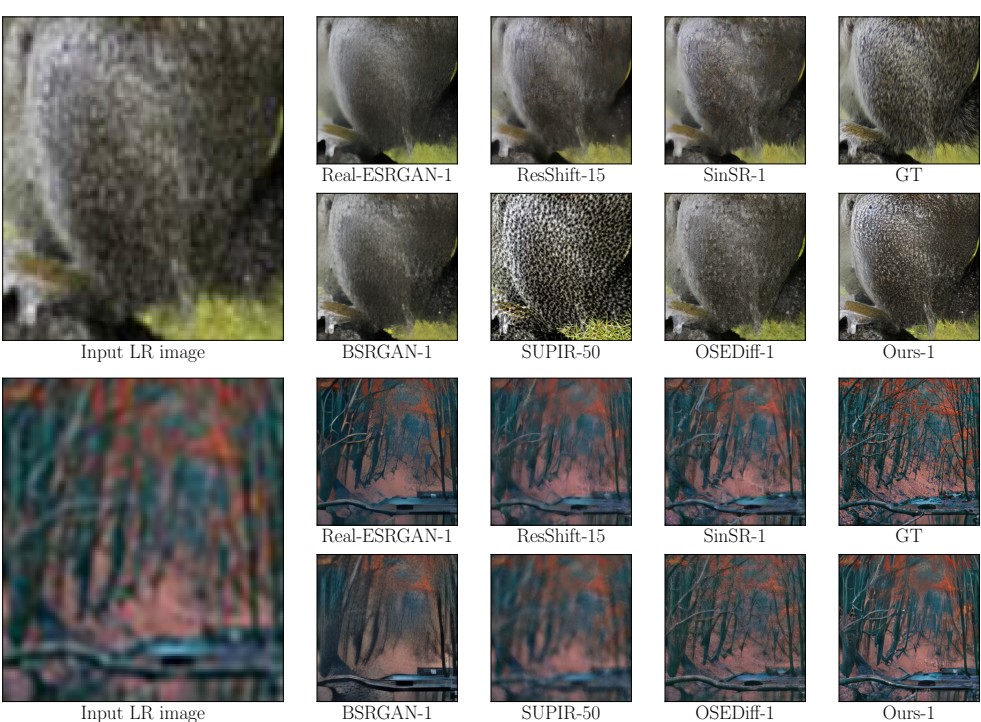

Figure 12: Failure cases on synthetic images from DIV2K crops (Agustsson & Timofte, 2017; Wang et al., 2024a). Please zoom in for a better view.

## K  PROOFS

*Proof of Proposition 3.1.* **First stage.** We first prove that using objective $\mathcal{L}_{\text{fake}}$ (see Eq. (8)) is equivalent to training a fake model $f_\phi$ with objective (5). We recall the $\mathcal{L}_{\text{fake}}$ minimization objective:

$$\arg\min_\phi \mathcal{L}_{\text{fake}}, \tag{35}$$

where

$$\mathcal{L}_{\text{fake}} = \Big( \sum_{t=1}^T w_t \mathbb{E}_{p_\theta(\widehat{x}_0,y_0,x_t)} \Big\{ \|f_\phi(x_t,y_0,t)\|_2^2 - 2\langle f_\phi(x_t,y_0,t) + \underbrace{f^*(x_t,y_0,t)}_{\text{Does not depend on }\phi}, \widehat{x}_0 \rangle \Big\} \Big) \tag{36}$$

Then we prove:

$$\arg\min_\phi \Big( \underbrace{\sum_{t=1}^T w_t \mathbb{E}_{p_\theta(\widehat{x}_0,y_0,x_t)} \|f_\phi(x_t,y_0,t) - \widehat{x}_0\|_2^2}_{\text{Training a fake model } f_\phi \text{ with objective (5)}} \Big) =$$

$$\arg\min_\phi \Big( \sum_{t=1}^T w_t \mathbb{E}_{p_\theta(\widehat{x}_0,y_0,x_t)} \Big\{ \|f_\phi(x_t,y_0,t)\|_2^2 - 2\langle f_\phi(x_t,y_0,t), \widehat{x}_0 \rangle \Big\} +$$

$$\underbrace{\sum_{t=1}^T w_t \mathbb{E}_{p_\theta(\widehat{x}_0,y_0,x_t)} \|\widehat{x}_0\|_2^2}_{\text{Does not depend on }\phi} \Big) =$$

$$\arg\min_\phi \Big( \sum_{t=1}^T w_t \mathbb{E}_{p_\theta(\widehat{x}_0,y_0,x_t)} \Big\{ \|f_\phi(x_t,y_0,t)\|_2^2 - 2\langle f_\phi(x_t,y_0,t), \widehat{x}_0 \rangle \Big\} \Big) =$$

$$\arg\min_\phi \Big( \sum_{t=1}^T w_t \mathbb{E}_{p_\theta(\widehat{x}_0,y_0,x_t)} \Big\{ \|f_\phi(x_t,y_0,t)\|_2^2 - 2\langle f_\phi(x_t,y_0,t), \widehat{x}_0 \rangle \Big\} -$$

$$\underbrace{\sum_{t=1}^T w_t \mathbb{E}_{p_\theta(\widehat{x}_0,y_0,x_t)} \Big\{ 2\langle f^*(x_t,y_0,t), x_0 \rangle \Big\}}_{\text{Does not depend on }\phi} \Big) =$$

$$\arg\min_\phi \Big( \sum_{t=1}^T w_t \mathbb{E}_{p_\theta(\widehat{x}_0,y_0,x_t)} \Big\{ \|f_\phi(x_t,y_0,t)\|_2^2 -$$

$$2\langle f_\phi(x_t,y_0,t) + \underbrace{f^*(x_t,y_0,t)}_{\text{Does not depend on }\phi}, \widehat{x}_0 \rangle \Big\} \Big) =$$

$$\arg\min_\phi \mathcal{L}_{\text{fake}}. \tag{37}$$

**Second stage.** Now we prove that:

$$\underbrace{\sum_{t=1}^T w_t \mathbb{E}_{p_\theta(\widehat{x}_0,y_0,x_t)} \|f_{G_\theta}(x_t,y_0,t) - f^*(x_t,y_0,t)\|_2^2}_{\mathcal{L}_\theta} =$$

$$-\min_\phi \Big\{ \sum_{t=1}^T w_t \mathbb{E}_{p_\theta(\widehat{x}_0,y_0,x_t)} \Big( \|f_\phi(x_t,y_0,t)\|_2^2 - \|f^*(x_t,y_0,t)\|_2^2 +$$

$$2\langle f^*(x_t,y_0,t) - f_\phi(x_t,y_0,t), \widehat{x}_0 \rangle \Big) \Big\} \tag{38}$$

Note, that since ResShift objective (34) is an MSE, the solution $f_{G_\theta}$ for the data produced by generator $G_\theta$ is given by the conditional expectation as:

$$f_{G_\theta}(x_t,y_0,t) = \mathbb{E}_{p_\theta(\widehat{x}_0|y_0,x_t)}[\widehat{x}_0]. \tag{39}$$

We start from the right part of (38) and transform it back to the left part:

$$-\min_{\phi}\Big\{\sum_{t=1}^{T}w_t\mathbb{E}_{p_\theta(\widehat{x}_0,y_0,x_t)}\Big(-\|f^*(x_t,y_0,t)\|_2^2+\|f_\phi(x_t,y_0,t)\|_2^2+$$
$$2\langle f^*(x_t,y_0,t)-f_\phi(x_t,y_0,t),\widehat{x}_0\rangle\Big)\Big\}=$$

$$\sum_{t=1}^{T}w_t\mathbb{E}_{p_\theta(\widehat{x}_0,y_0,x_t)}\Big\{\|f^*(x_t,y_0,t)\|_2^2-2\langle f^*(x_t,y_0,t),\widehat{x}_0\rangle\Big\}-$$

$$\min_{\phi}\Big\{\sum_{t=1}^{T}w_t\mathbb{E}_{p_\theta(\widehat{x}_0,y_0,x_t)}\Big(\|f_\phi(x_t,y_0,t)\|_2^2-2\langle f_\phi(x_t,y_0,t),\widehat{x}_0\rangle\Big)\Big\}=$$

$$\sum_{t=1}^{T}w_t\mathbb{E}_{p_\theta(y_0,x_t)}\Big(\|f^*(x_t,y_0,t)\|_2^2-2\langle f^*(x_t,y_0,t),\underbrace{\mathbb{E}_{p_\theta(\widehat{x}_0|y_0,x_t)}\widehat{x}_0}_{f_{G_\theta(x_t,y_0,t)}}\rangle\Big)-$$

$$\min_{\phi}\Big\{\sum_{t=1}^{T}w_t\mathbb{E}_{p_\theta(\widehat{x}_0,y_0,x_t)}\Big(\|f_\phi(x_t,y_0,t)\|_2^2-2\langle f_\phi(x_t,y_0,t),\widehat{x}_0\rangle\Big)\Big\}=$$

$$\sum_{t=1}^{T}w_t\mathbb{E}_{p_\theta(y_0,x_t)}\Big(\|f^*(x_t,y_0,t)\|_2^2-2\langle f^*(x_t,y_0,t),f_{G_\theta(x_t,y_0,t)}\rangle\Big)-$$

$$\sum_{t=1}^{T}w_t\mathbb{E}_{p_\theta(y_0,x_t)}\Big(\|f_{G_\theta}(x_t,y_0,t)\|_2^2-2\underbrace{\langle f_{G_\theta}(x_t,y_0,t),f_{G_\theta(x_t,y_0,t)}\rangle}_{\|f_{G_\theta}\|_2^2}\Big)=$$

$$\sum_{t=1}^{T}w_t\mathbb{E}_{p_\theta(y_0,x_t)}\Big(\|f^*(x_t,y_0,t)\|_2^2-2\langle f^*(x_t,y_0,t),f_{G_\theta(x_t,y_0,t)}\rangle+\|f_{G_\theta}(x_t,y_0,t)\|_2^2\Big)=$$

$$\sum_{t=1}^{T}w_t\mathbb{E}_{p_\theta(y_0,x_t)}\underbrace{\|f_{G_\theta}(x_t,y_0,t)-f^*(x_t,y_0,t)\|_2^2}_{\text{Does not depend on }\widehat{x}_0\text{ so we can add }\widehat{x}_0\text{ in expectation.}}=$$

$$\sum_{t=1}^{T}w_t\mathbb{E}_{p_\theta(\widehat{x}_0,y_0,x_t)}\|f_{G_\theta}(x_t,y_0,t)-f^*(x_t,y_0,t)\|_2^2.$$

$\square$

**Discussion**. We explain the intractability of the gradient for $\mathcal{L}_\theta$ in Equation (7) as follows. The gradient of $\mathcal{L}_\theta$ (7) over parameters $\theta$ of $G_\theta$ is given by the chain rule:

$$\frac{d\mathcal{L}_\theta}{d\theta}=\underbrace{\frac{\partial\mathcal{L}}{\partial\theta}}_{\text{direct}}+\underbrace{\frac{\partial\mathcal{L}}{\partial f_{G_\theta}}\cdot\frac{\partial f_{G_\theta}}{\partial\theta}}_{\text{implicit}}\tag{40}$$

$\frac{d\mathcal{L}_\theta}{d\theta}$ contains an implicit term, which requires a differentiation through the full ResShift training loop and is computationally infeasible (because one needs to backpropagate through all the gradient updates used to train $f_{G_\theta}$):

$$\frac{\partial f_{G_\theta}}{\partial\theta}=\frac{\partial}{\partial\theta}\underbrace{\Big[\arg\min_{\phi}[\mathcal{L}(\phi,\theta)]\Big]}_{\text{Training on the Generator outputs}}\tag{41}$$

In contrast, our Proposition 3.1 and Equation (8) resolve this by deriving the mathematically equivalent form of Equation (7), which does not require directly differentiating through the "re-training" step $f_{G_\theta}$ i.e., it does not contain terms with argmin operations.

## L  MORE VISUAL RESULTS

This section provides an additional qualitative visual comparisons between RSD and other baselines on two real-world full-size and two synthetic benchmarks with available ground truth data.

1. Full-size benchmark RealSR (Cai et al., 2019). The results are shown in Figure 13.

2. Full-size benchmark DRealSR (Wei et al., 2020). The results are shown in Figure 14.

3. Synthetic benchmark ImageNet-Test (Deng et al., 2009; Yue et al., 2023). The results are shown in Figure 15.

4. Synthetic benchmark DIV2K (Agustsson & Timofte, 2017; Wang et al., 2024a). The results are shown in Figure 16.

The baseline methods include multistep diffusion-based SR methods (ResShift (Yue et al., 2023), SUPIR (Yu et al., 2024)), 1-step diffusion-based SR methods (SinSR (Wang et al., 2024b), OSEDiff (Wu et al., 2024a)), and GAN-based SR methods (Real-ESRGAN (Wang et al., 2021) and BSRGAN (Zhang et al., 2021)).

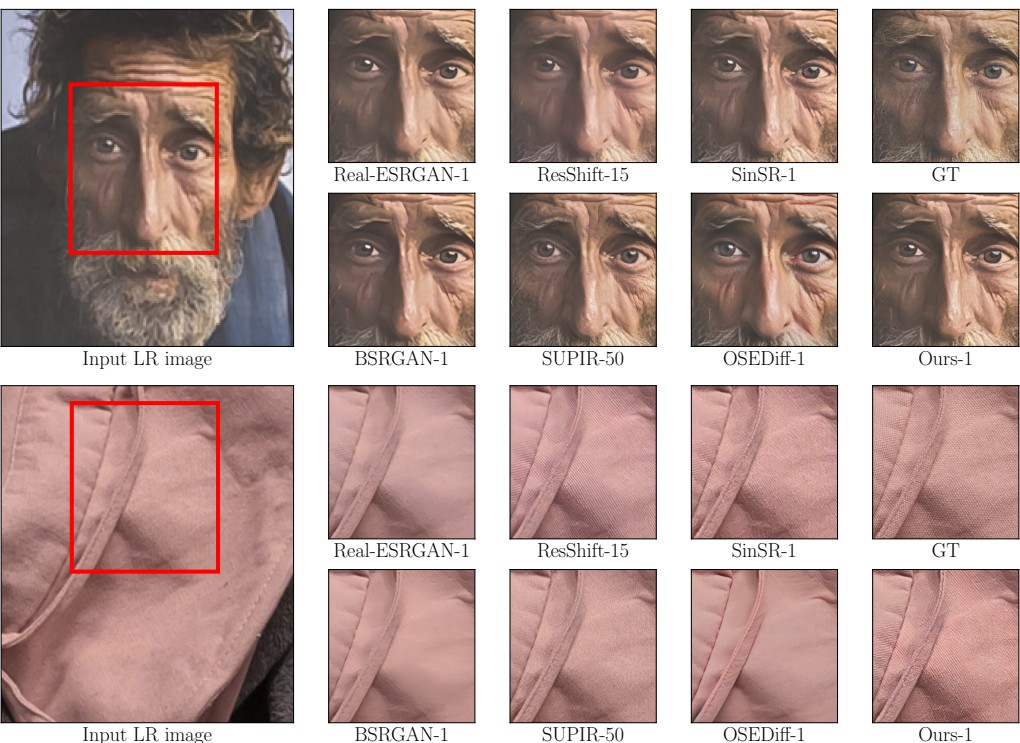

Figure 13: Visual comparison on real-world samples from RealSR (Cai et al., 2019). Please zoom in for a better view.

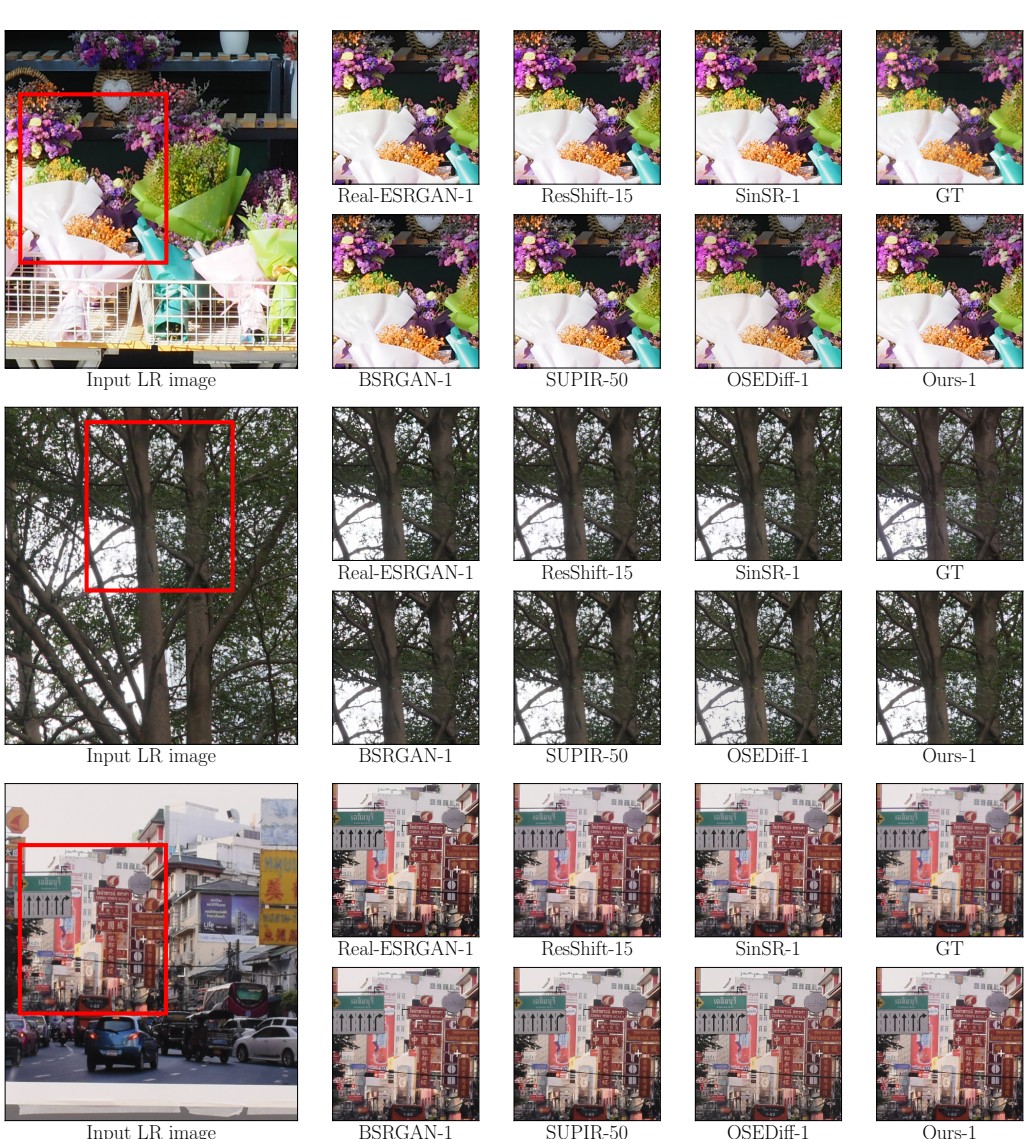

Figure 14: Visual comparison on real-world samples from DRealSR (Wei et al., 2020). Please zoom in for a better view.

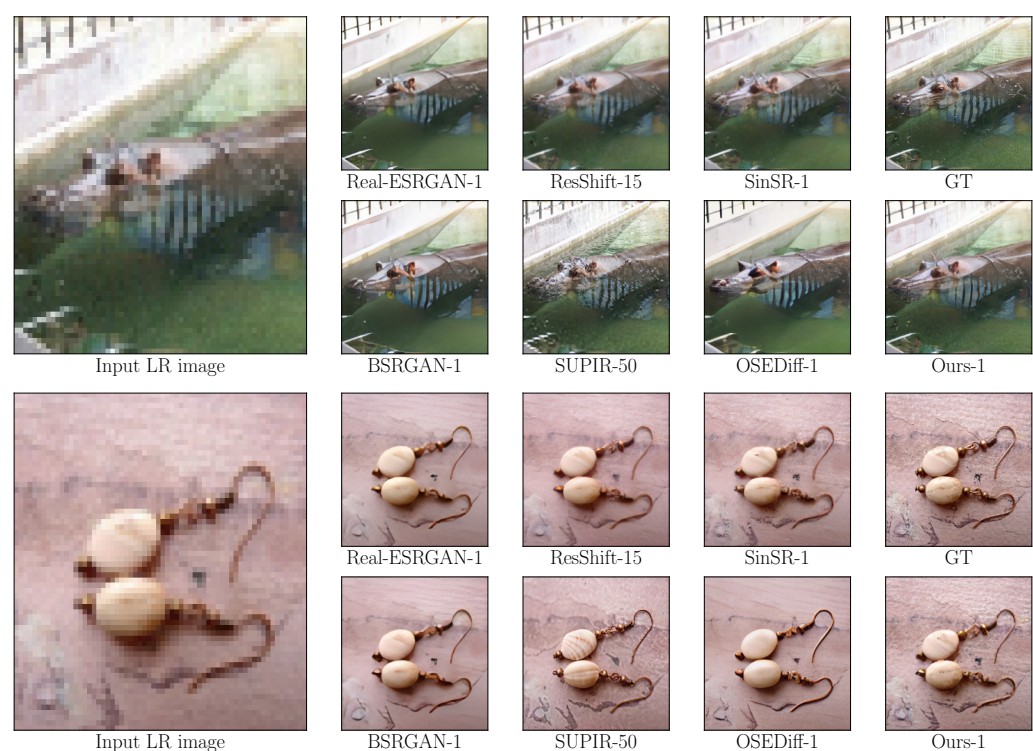

Figure 15: Visual comparison on synthetic samples from ImageNet-Test (Deng et al., 2009; Yue et al., 2023). Please zoom in for a better view.

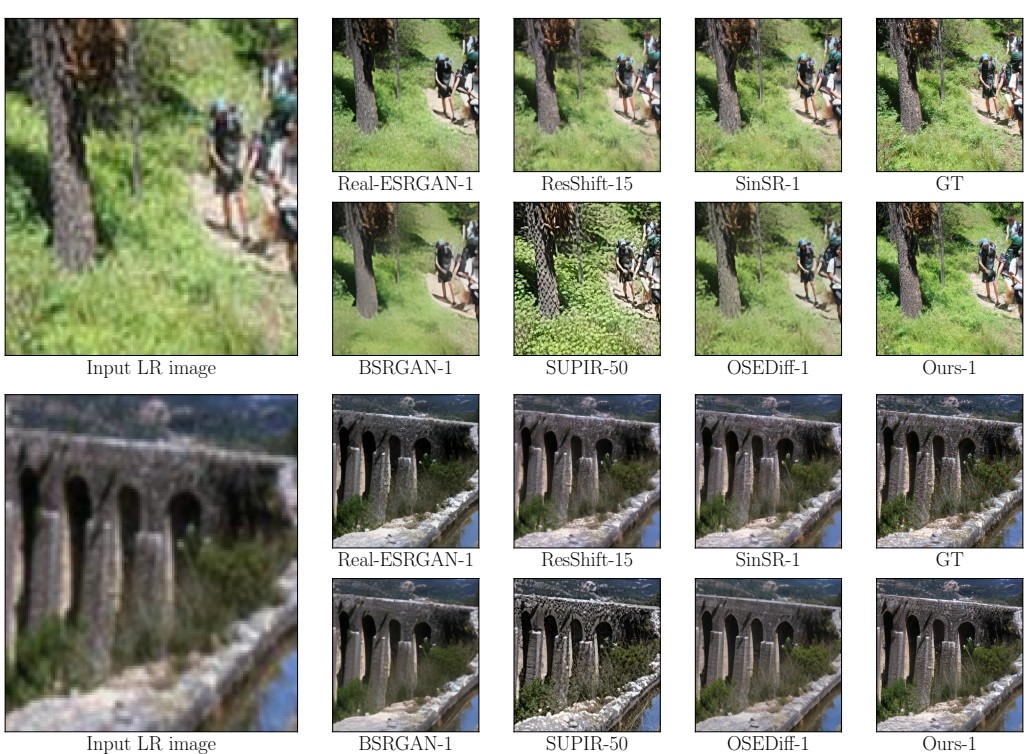

Figure 16: Visual comparison on synthetic samples from DIV2K crops (Agustsson & Timofte, 2017; Wang et al., 2024a). Please zoom in for a better view.

# M   STATEMENT ON LLM USAGE

The authors used the large language model (LLM) only to improve the writing and grammar of the text. All the results from the LLM were checked by the authors.

