# OpenReview forum: "One-Step Residual Shifting Diffusion for Image Super-Resolution via Distillation"
_ICLR.cc/2026/Conference — Submitted to ICLR 2026_

### Official Review · Reviewer_mz7A · 2025-10-30

**Soundness:** 3
**Presentation:** 3
**Contribution:** 3
**Rating:** 6
**Confidence:** 3

**Summary:**

The paper proposes Residual Shifting Distillation (RSD), a distillation framework for accelerating the ResShift diffusion-based super-resolution model to a single-step inference regime. RSD introduces a trainable “fake” ResShift model to align the joint trajectory distribution of the student-generated data with that of the teacher, in contrast to prior methods (e.g., VSD) that align only marginal distributions per timestep. The method is combined with LPIPS and GAN losses for improved perceptual quality. Extensive experiments on real-world (RealSR, DRealSR) and synthetic (ImageNet-Test, DIV2K) benchmarks demonstrate that RSD outperforms SinSR in perceptual metrics while maintaining competitive fidelity, and significantly reduces computational cost compared to T2I-based methods like OSEDiff and SUPIR.

**Strengths:**

1. Clear and principled formulation: The paper provides a well-motivated objective (Eq. 7–9) based on KL divergence over the full reverse trajectory, with a tractable surrogate derived via Proposition 3.1. The distinction between joint vs. marginal distribution alignment is clearly illustrated (Fig. 4, Appendix A).
2. Comprehensive experiments: The evaluation covers multiple datasets, degradation models, and SOTA baselines across GAN-based, diffusion-based, and T2I-based SR methods. Both quantitative (Tables 1–3, Appendix D–E) and qualitative results (Figs. 3, 5, 9–12) are thorough and convincing.
3. Strong practical impact: RSD achieves 1-step inference, 174M parameters, and <600MB GPU memory, making it highly deployable. Training is ~3× faster than SinSR due to simulation-free distillation.
4. Honest discussion of limitations: The authors acknowledge dependence on the ResShift teacher, failure cases on complex textures, and the gap with T2I models in hallucination-rich scenarios (Appendix H).
5. Excellent reproducibility: Pseudocode (Appendix B), training details (Appendix C), dataset licenses (Table 8), and codebase description are provided.

**Weaknesses:**

1. Incremental theoretical novelty: As acknowledged in Appendix A, RSD is closely related to IBMD (Gushchin et al., 2025) and can be viewed as its discrete, task-specific adaptation to ResShift. The core idea—using an auxiliary model to match joint distributions—has appeared in consistency distillation, diffusion bridges, and related works.
2. Missing comparison to very recent 1-step SR methods: For example, CCSR (Sun et al., 2024) and TSD-SR (Dong et al., 2024) are mentioned but not included in main tables.
3. Limited generalizability: The method is tightly coupled to the ResShift architecture (residual shifting, latent-space diffusion). It is unclear whether RSD can be applied to other diffusion frameworks (e.g., I2SB, LDM) without significant redesign.
4. Training-resolution mismatch in comparisons: RSD is trained on 256×256 crops, while OSEDiff uses 512×512. Although Appendix E shows RSD trained at 512×512, the main results are not fully comparable. This slightly favors T2I methods in perceptual metrics.

For clarity, the following should be addressed.
The paper is generally well organized, but the abstract contains a 76-word sentence that should be split. Figure fonts are smaller than 8 pt and become illegible when printed. A symbol table summarizing latent-space vs pixel-space notation would help readers. Additionally, the phrase “diffusion-based SR models” appears four times within three lines; replacing subsequent instances with “such approaches” or “these methods” would improve fluency.

For the experiments, the following should be addressed.
1. The current teacher ResShift was pretrained on 256² crops, limiting its modelling of 512² high-frequency details and causing RSD to lag behind OSEDiff/SUPIR on high-resolution data. Higher-resolution teacher distillation is deferred to future work.
2. To enable a fairer assessment of the performance gap with T2I-based methods (e.g., OSEDiff), it would be better including in the main experiments an additional set of results where RSD is trained and fully fine-tuned at 512×512 resolution.
3. To strengthen the empirical evaluation and better position RSD within the landscape of state-of-the-art one-step super-resolution methods, it would be more convincing that the authors include a direct quantitative and qualitative comparison with CCSR (Sun et al., 2024) and TSD-SR (Dong et al., 2024) in the main results.

**Questions:**

1. Could RSD be applied to a non-ResShift diffusion model (e.g., standard DDPM or LDM)? What architectural or training changes would be needed?
2. In Appendix E, RSD trained on 512×512 still lags behind OSEDiff in CLIPIQA/MUSIQ. Do the authors believe this gap is due to the teacher (ResShift) or the distillation objective?

---

> ### Author Response · Authors · 2025-11-21
> **Comments on questions**
>
> Dear Reviewer mz7A, thank you for your comments. We are working on the revised version of the text. Here are the answers to your questions and comments.
>
> **Question 1: theoretical novelty**.
>
> **(1) "... Incremental theoretical novelty: As acknowledged in Appendix A, RSD is closely related to IBMD (Gushchin et al., 2025) and can be viewed as its discrete, task-specific adaptation to ResShift. The core idea—using an auxiliary model to match joint distributions—has appeared in consistency distillation, diffusion bridges, and related works. ..."**
>
> Thank you for your comments. Here, we would like to highlight our theoretical contributions and key differences to prior works:
>
> (i) _Discrete time generalization._ First of all, our work generalizes the IBMD framework, which was initially developed for continuous time processes, to a discrete time domain. The crucial difference is that the optimization objective of discrete-time models is derived from other principles and represents ELBO on the target distribution density. We think that it was vital because a lot of work leverages discrete-time processes to build their models, such as DDPM, LDM, and ResShift. Indeed, it was not clear whether the proof that was initially introduced in IBMD would fit into these frameworks, mainly because in the work Girsanov theorem (see Proof of Proposition 3.1 in IBMD) was used, which is applicable for continuous time processes. We proved our theorem without relying on that fact.
>
> (ii) _Comparisons of VSD-like vs RSD-like loss functions._ The second contribution is that we theoretically show the **difference** between VSD-like (see Equation 16) and RSD-like (see Equation 20) losses, which is illustrated in Figure 4. In short, we show that the former one matches marginals between the distribution induced by fake ResShift model and distribution induced by teacher ResShift model for each time step, while the latter one matches the joint of these distributions (for more details, please see Appendix A.1). We hypothesize that this joint alignment is particularly beneficial for SR tasks, where maintaining consistency and accuracy across all image details and features is crucial for high-quality resolution (for justification in practice, see Table 1 and 2 ResShift-VSD vs RSD (Ours, distill only)).

---

> > ### Comment · Reviewer_mz7A · 2025-11-27
> >
> > We appreciate the authors’ detailed clarification regarding the theoretical contributions of RSD.
> > Although RSD’s discrete-time formulation has technical merit, its core idea—joint distribution alignment via an auxiliary model—closely aligns with IBMD, SiD, and FGM, as acknowledged by the authors (Appendix A). Thus, the theoretical contribution is best viewed as a task-specific adaptation and empirical validation within ResShift-based super-resolution, not a new distillation paradigm.
> > The rebuttal clarifies the method’s motivation and strengthens rigor, but does not fully address the limited theoretical novelty. The revised manuscript should frame RSD accordingly: not as a methodological breakthrough, but as an efficient, well-validated instantiation of IBMD in discrete diffusion-based SR.

---

> ### Author Response · Authors · 2025-11-21
> **Comments on questions**
>
> **Question 2: comparison with very recent 1-step SR methods**.
>
> **(2) "... Missing comparison to very recent 1-step SR methods: For example, CCSR (Sun et al., 2024) and TSD-SR (Dong et al., 2024) are mentioned but not included in main tables. ... To strengthen the empirical evaluation and better position RSD within the landscape of state-of-the-art one-step super-resolution methods, it would be more convincing that the authors include a direct quantitative and qualitative comparison with CCSR (Sun et al., 2024) and TSD-SR (Dong et al., 2024) in the main results. ..."**
>
> We thank the reviewer for pointing out the question of the comparison with these recent state-of-the-art methods. Following your comment, we extend the numerical comparison between the proposed RSD method and the TSD-SR, PiSA-SR methods in our Tables 1 and 2. We used the officially published pretrained models and inference code from the respective GitHub repositories reported in these works.
>
> Additional results on RealSR in Table 1 are the following:
>
> |**Method**|**NFE**|**PSNR**$\uparrow$|**SSIM**$\uparrow$|**LPIPS**$\downarrow$|**CLIPIQA**$\uparrow$|**MUSIQ**$\uparrow$|
> |-|-|-|-|-|-|-|
> |RSD|1|25.91|0.754|0.273|0.7060|65.860|
> |TSD-SR|1|24.88|0.723|0.281|0.7336|69.871|
> |CCSR|1|25.99|0.752|0.287|0.6656|67.991|
>
> Additional results on RealSet65 in Table 1 are the following:
>
> |**Method**|**NFE**|**CLIPIQA**$\uparrow$|**MUSIQ**$\uparrow$|
> |-|-|-|-|
> |RSD|1|0.7267|69.172|
> |TSD-SR|1|0.7263|70.958|
> |CCSR|1| 0.7150|70.731|
>
> Additional results on ImageNet in Table 2 are the following:
>
> |**Method**|**NFE**|**PSNR**$\uparrow$|**SSIM**$\uparrow$|**LPIPS**$\downarrow$|**CLIPIQA**$\uparrow$|**MUSIQ**$\uparrow$|
> |-|-|-|-|-|-|-|
> |RSD|1|24.31|0.657|0.193|0.681| 58.947|
> |TSD-SR|1|23.58|0.645|0.197|0.673|65.299|
> |CCSR|1|24.79|0.677|0.238|0.602|61.789|
>
> **1. Quantitative comparison**. We observe that RSD outperforms CCSR in terms of LPIPS and CLIPIQA metrics across all datasets, while exhibiting only marginal differences in the remaining metrics. In Tables 1 and 2, we observe that our RSD outperforms TSD-SR in reference-based fidelity (PSNR, SSIM) and perceptual (LPIPS) metrics, while TSD-SR achieves better no-reference perceptual metrics (CLIPIQA, MUSIQ) compared to RSD.
>
> **2. Complexity comparison**. Despite the good perceptual performance of TSD-SR, this T2I-based model requires more computational costs compared to the other one-step T2I-based SR model, OSEDiff, and much more computational costs compared to RSD. To show it, we updated Table 4 with additional information on the training time and the corresponding number of GPUs used for the training according to the information reported in this work:
>
> |**Methods**|RSD|TSD-SR|
> |-|-|-|
> | **Inference Step (NFE)** | 1 | 1 |
> | **Inference Time (s)**|0.059| 0.074 |
> | **# Total Parameters (M)** | 174 | 2207 |
> | **Maximum GPU memory (MB)** | 539 | 4611 |
> | **Training time (hours/ # GPUs)** | 5 / 4 A100 | 96 / 8 A100 |
>
> **Summary**. We highlight the training efficiency of our RSD compared to TSD-SR: RSD has $\times 19$ faster training with $\times 2$ times less GPUs compared to TSD-SR. During inference, TSD-SR requires $\times 13$ more parameters and $\times 8$ more GPU memory than RSD. Even with fast inference, the deployment of TSD-SR is limited by a significant computational budget and heavy architecture. Similar observations are applicable for CCSR: this model requires $\times 9$ more parameters compared to RSD during inference and also requires two-stage training process with additional finetuning of VAE.
>
> The analysis supports our claim in lines 101-102 that RSD aims to **compromise between fidelity, perceptual quality, and
> computational efficiency**. We will add these results to the revised version of the text.

---

> > ### Comment · Reviewer_mz7A · 2025-11-27
> >
> > The authors have addressed the concern about missing comparisons with recent one-step SR methods (CCSR and TSD-SR). Their new results provide a clear and fair empirical assessment.
> > Most notably, the complexity analysis strongly supports RSD’s practical value: 174M parameters vs. TSD-SR’s 2.2B, and 19× faster training—highlighting RSD as a highly efficient alternative without sacrificing core performance.
> > These results convincingly position RSD within the current SOTA landscape. We recommend incorporating this comparison into the main paper (e.g., a revised Table 1 or supplementary table) and explicitly discussing the performance–efficiency trade-off. This addition significantly strengthens the paper’s empirical contribution.

---

> ### Author Response · Authors · 2025-11-21
> **Comments on questions**
>
> **Question 3: generalization of other diffusion frameworks**.
>
> **(3) "... Limited generalizability: The method is tightly coupled to the ResShift architecture (residual shifting, latent-space diffusion). It is unclear whether RSD can be applied to other diffusion frameworks (e.g., I2SB, LDM) without significant redesign. ... Could RSD be applied to a non-ResShift diffusion model (e.g., standard DDPM or LDM)? What architectural or training changes would be needed? ...."**
>
> We appreciate your comment and hope to resolve your concerns regarding generalization of our approach to the other diffusion based methods. Indeed, although we mentioned that our framework can be generalized to a broader scope of approaches (see Remark on line 175), we have not clarified it further in the text. The following text shows how our approach can be generalized to other processes, including I2SB and LDM, and we will add it to the revised version of the text.
>
> It is noteworthy that the proof of Proposition 3.1 (see Appendix I), as well as the formulation of our loss function, does not rely on a specific form of forward process used in the ResShift model. The only difference from other approaches is how they sample from the joint distribution $p\_{\theta}(\widehat{x}\_{0}, y\_{0}, x\_{t})$ during the training procedure. Usually, $p\_{\theta}(\widehat{x}\_{0}, y\_{0}, x\_{t})$ is written in the following way:
> \begin{gather}
>     p\_{\theta}(\widehat{x}\_{0}, y\_{0}, x\_{t}) = q(x\_{t} \mid \widehat{x}\_{0}, y\_{0}) p\_{\theta}(\widehat{x}\_{0} \mid y\_{0}) p(y\_{0}),
> \end{gather}
> showing that the only thing that is different for each method is used distribution $q.$  Thus, to adopt our approach to other processes, one only needs to change the distribution $q$. For example, to train _I2SB_ or _LDM_ models using RSD formulation, one can use their discrete formulations for both models (see Equation 11 in I2SB or Equation 4 in LDM, respectively) and plug them into $\mathcal{L}\_{\theta}$. Answering your question, as you can see, no extra architectural or training changes would be needed.
>
> **Question 4: results of RSD trained on $512 \times 512$ images**.
>
> **(4) "... Training-resolution mismatch in comparisons: RSD is trained on 256×256 crops, while OSEDiff uses 512×512. Although Appendix E shows RSD trained at 512×512, the main results are not fully comparable. This slightly favors T2I methods in perceptual metrics. ... The current teacher ResShift was pretrained on 256² crops, limiting its modelling of 512² high-frequency details and causing RSD to lag behind OSEDiff/SUPIR on high-resolution data. Higher-resolution teacher distillation is deferred to future work. ...To enable a fairer assessment of the performance gap with T2I-based methods (e.g., OSEDiff), it would be better including in the main experiments an additional set of results where RSD is trained and fully fine-tuned at 512×512 resolution.
> ... In Appendix E, RSD trained on 512×512 still lags behind OSEDiff in CLIPIQA/MUSIQ. Do the authors believe this gap is due to the teacher (ResShift) or the distillation objective? ..."**
>
> We appreciate the reviewer’s concern about mismatch in training procedure between RSD and T2I models. As you mentioned, we trained the RSD on $512 \times 512$ crops and included the results in Appendix E. However, for a more fair comparison with the T2I models, we will move the results into the main text of the revised version. Due to the fact that our model relies on the pretrained teacher model, we trained our own ResShift model on $512 \times 512$ image. Answering your question, we suppose that this drop in CLIPIQA and MUSIQ metrics appears mainly due to the teacher ResShift model. To explain it, we observe that our RSD model reflects the behavior of the teacher in Tables 1 and 14. ResShift model trained on $256 \times 256$ resolution images tends to produce higher non-reference perceptual metrics, such as CLIPIQA and MUSIQ, while ResShift model trained on $512 \times 512$ images gives better results for reference based metrics, such as PSNR, SSIM, LPIPS. The same happens for RSD as well, explaining such a drop for both non-reference metrics.

---

> > ### Comment · Reviewer_mz7A · 2025-11-27
> >
> > The authors correctly note that RSD’s formulation is not strictly limited to ResShift, as Proposition 3.1 depends only on the sampling distribution . This suggests theoretical extensibility to other DDPM-based models.
> > However, while the formulation may be general, the practical implementation and performance are deeply intertwined with ResShift’s specific design (residual shifting, latent-space diffusion). The authors' claim that "no extra architectural or training changes would be needed" for models like I2SB or LDM is overly optimistic. Adapting RSD to these frameworks would likely require significant engineering effort to handle their distinct forward processes, noise schedules, and model architectures, even if the loss function structure remains similar.
> > In summary, the rebuttal provides a sound theoretical justification for broader applicability. To fully address the concern, the revised manuscript should more accurately frame this as a theoretical possibility rather than an immediate practical reality, acknowledging the non-trivial implementation challenges involved in adapting RSD to other diffusion paradigms.

---

> > ### Comment · Reviewer_mz7A · 2025-11-27
> >
> > Your rebuttal and Appendix E provide a clear, fair comparison: RSD trained on 512x512 crops still lags behind OSEDiff in CLIPIQA/MUSIQ, while outperforming it in fidelity metrics (PSNR/SSIM/LPIPS). This confirms the reviewer’s concern that the original 256x256 training setup favors T2I models.
> > The authors’ hypothesis—that the performance gap stems primarily from the teacher ResShift model—is well-supported by the data. The trend holds for both the teacher and its distilled student: higher-resolution training improves fidelity but not necessarily no-reference perceptual scores. This is a crucial insight, highlighting a fundamental limitation of the current distillation pipeline rather than a flaw in the RSD objective itself.
> > We recommend moving these results into the main paper to ensure a more complete and equitable evaluation against SOTA T2I methods. The analysis should explicitly frame this as a "teacher capacity" issue, which is a valid and important contribution to the discussion.

---

> ### Author Response · Authors · 2025-11-21
> **Comments on questions**
>
> **Question 5: clarity**.
>
> **(5) "... For clarity, the following should be addressed. The paper is generally well organized, but the abstract contains a 76-word sentence that should be split. Figure fonts are smaller than 8 pt and become illegible when printed. A symbol table summarizing latent-space vs pixel-space notation would help readers. Additionally, the phrase “diffusion-based SR models” appears four times within three lines; replacing subsequent instances with “such approaches” or “these methods” would improve fluency. ..."**
>
> We thank the reviewer for raising this point. We will fix the abstract, change the size captions of the figures, and add a table describing the variables in the initial pixel $x$ and latent $z$ spaces in the revised version of the text. We will also change frequently used terms for more clarity of the text.
>
> **Concluding remarks**. We would be grateful if you could let us know if our explanations have been satisfactory. If so, we kindly ask that you consider increasing your rating. We are also open to discussing any other questions you may have.

---

### Official Review · Reviewer_kZUH · 2025-10-31

**Soundness:** 3
**Presentation:** 4
**Contribution:** 2
**Rating:** 4
**Confidence:** 4

**Summary:**

This paper presents RSD (Residual Shifting Distillation), a one-step distillation method for diffusion-based image super-resolution. The approach trains a student generator to produce images such that a "fake" ResShift model trained on these generated images matches the original teacher model. RSD achieves competitive results compared to the 15-step teacher ResShift and claims to outperform existing distillation methods. Experiments are conducted on RealSR, RealSet65, DRealSR, ImageNet, and DIV2K datasets.

**Strengths:**

1.	The paper combines a distillation objective that aligns joint distributions rather than marginal distributions at each timestep (as in VSD) with the ResShift architecture.
2.	Extensive quantitative results are provided across multiple benchmarks.
3.	Comprehensive related work discussion and comparison with VSD, SiD, FGM, and IBMD frameworks.

**Weaknesses:**

1.	Missing baseline method comparison. Appendix A.3 acknowledges that RSD is essentially a "discrete variant of IBMD," yet IBMD is never empirically compared. Given this close relationship, quantitative comparison is essential to validate the claimed improvements from task-specific adaptations. Equation 21 explicitly shows RSD is IBMD applied to ResShift.
2.	Limited Novelty. The main contribution is essentially combining ResShift with VSD/IBMD frameworks. Moreover, the authors also acknowledge the relationship to IBMD (Appendix A.3, Lines 1242-1295).
3.	Misleading Experimental Claims. The authors claim to "outperform the teacher by a large margin" and . However, in Table 1 (RealSR), RSD loses on fidelity metric PSNR.
4.	The paper ignores recent state-of-the-art methods including:
[1] PiSASR (CVPR 2025)
[2] TSDSR (Dong et al., 2024)
[3] InvSR (CVPR 2025)
5.	These omissions weaken the comprehensiveness of the experimental comparison.
6.	FID is not reported, despite being widely used in diffusion-based SR papers such as StableSR and OSEDiff. This limits comparability with existing literature.
7.	Visual quality does not significantly outperform baseline methods like OSEDiff. For example, differences in Figure 10 (Lines 2295 and 2306) are minimal, which undermines claims of substantial perceptual improvements.

**Questions:**

See the weakness.

---

> ### Author Response · Authors · 2025-11-21
> **Comments on questions**
>
> Dear Reviewer kZUH, thank you for your comments. We are working on the revised version of the text. Here are the answers to your questions and comments.
>
> **Question 1: Quantitative comparison with IBMD**.
>
> **(1) "...Missing baseline method comparison. Appendix A.3 acknowledges that RSD is essentially a "discrete variant of IBMD," yet IBMD is never empirically compared. Given this close relationship, quantitative comparison is essential to validate the claimed improvements from task-specific adaptations. Equation 21 explicitly shows RSD is IBMD applied to ResShift. ..."**
>
> As noted in **lines 1212-1215**, IBMD originally distilled the I2SB teacher model (Liu et al., 2023a) for simple degradations, such as bicubic and pool (see Tables 1 and 3 in IBMD and Table 2 in I2SB). Following your comment, we conducted the following numerical experiments to show that both the teacher model of I2SB and the distilled student model of IBMD do not provide sufficient perceptual quality in Real-ISR problems compared to ResShift and RSD, respectively.
>
> **Step 1: training the I2SB teacher using Real-ESRGAN degradations**. For a fair comparison with ResShift, we trained an I2SB model on ImageNet using the Real-ESRGAN degradations following the training setup of ResShift and RSD detailed in **lines 369-373**. The model was trained using the same hyperparameters as the original I2SB model trained on bicubic degradations with $4000$ iterations. We used the official I2SB implementation published in the respective GitHub repository, which is provided by the I2SB authors.
>
> **Step 2: distillation of I2SB with IBMD**. We asked the authors of IBMD to provide the training and evaluation code of the IBMD method, which originally distilled I2SB with bicubic degradations. We adapt the provided implementation with the replacement of Real-ESRGAN degradations instead of original bicubic degradations and use the same hyperparameters, which were used for the training of IBMD model for bicubic degradations (first line in Table 7 of IBMD). We distilled the trained I2SB teacher with Real-ESRGAN degradations using IBMD method into one-step student model with 1500 gradient updates for the student model.
>
> For a fair comparison with ResShift and RSD, we evaluated the trained teacher I2SB with NFE = 15 and 1 and the trained IBMD student with NFE = 1, which follows the inference NFE of ResShift and RSD, respectively. We report on the results of their evaluation on the ImageNet-Test dataset, which follows the RSD evaluation setup reported in Table 2, and compare them with the ResShift, RSD with supervised losses (RSD (Ours)), and RSD with distillation loss only (RSD (Ours, distill only)) in the following table:
>
> |**Method**|**NFE**|**PSNR**$\uparrow$|**SSIM**$\uparrow$|**LPIPS**$\downarrow$|**CLIPIQA**$\uparrow$|**MUSIQ**$\uparrow$|
> |-|-|-|-|-|-|-|
> |I2SB|1| 25.52 | 0.690 | 0.412 | 0.405 | 34.439 |
> |I2SB|15| 24.80 | 0.663 | 0.302 | 0.444 | 49.584 |
> |ResShift|15| 25.01 | 0.677 | 0.231 | 0.592 | 53.660 |
> |IBMD|1| 23.91 | 0.619 | 0.284 | 0.505 | 54.667 |
> |RSD (Ours, distill only) | 1 | 23.97 | 0.643 | 0.217 | 0.660| 57.831 |
> |RSD (Ours) | 1 | 24.31 | 0.657 | 0.193 | 0.681 | 58.947 |
>
> We also extend the complexity comparison in our Table 4 of RSD with IBMD and I2SB with ResShift by adding information about the training time as follows:
>
> |**Methods**|RSD|IBMD|
> |-|-|-|
> | **Inference Step (NFE)** | 1 | 1 |
> | **Inference Time (s)**|0.059| 0.077 |
> | **# Total Parameters (M)** | 174 | 553 |
> | **Maximum GPU memory (MB)** | 539 | 4676 |
> | **Training time (hours/ # GPUs)** | 5 / 4 A100 | 23 / 8 A100 |

---

> ### Author Response · Authors · 2025-11-21
> **Comments on questions**
>
> **Continuation of discussion for Question 1**.
>
> **Comparison between teachers, I2SB and ResShift**. Our results are consistent with the analysis for comparison between ResShift and I2SB on simpler bicubic degradations, which is given in Appendix B.2 of ResShift: ResShift achieves better results on ImageNet dataset with complex Real-ESRGAN degradations compared to I2SB in **all evaluation metrics** (PSNR, SSIM, LPIPS, MUSIQ, CLIPIQA) with a big improvement in perceptual metrics (LPIPS, MUSIQ, CLIPIQA). The same conclusion is quantitatively validated in Table 5 and visually supported in Figure 8 of ResShift, respectively. The results of Table 5 in ResShift also show a significant improvement in terms of perceptual quality for ResShift compared to I2SB for bicubic degradations with the same NFE = 15. We explain these results by the specific design of ResShift, which applies the diffusion process in discrete time in the latent space of VAE and uses the non-uniform geometric noise schedule (Section 2 in ResShift).
>
> **Comparison between students, IBMD and RSD**. For a fair comparison, we compare our RSD model without supervised losses and the IBMD model, because IBMD originally is a data-free distillation method (see contribution 3 in IBMD). The results show that RSD is better in **all evaluation metrics even without supervised losses** (PSNR, SSIM, LPIPS, MUSIQ, CLIPIQA) with a significant improvement in perceptual metrics (LPIPS, MUSIQ, CLIPIQA). The addition of supervised losses in RSD even more increases the margin between all evaluation metrics. In practice, we also found that the IBMD model requires a significant computational budget, which aligns with the reported complexity in Table 9 of IBMD. We trained the IBMD model on 8 A100 for 23 hours, which is $>4$ times more than the training time of RSD. IBMD also has $>3$ times bigger number of parameters and $>8$ times bigger required GPU memory for inference.
>
> **Summary**. For a quantitative comparison, RSD model without supervised losses outperforms the IBMD model in all evaluation metrics. For a computational comparison, RSD model has a much faster distillation time, requires less number of parameters, GPU memory, and has a faster generation time during inference compared to IBMD. We will add these results to the revised version of the text. Thus, task-specific features of RSD used for Real-ISR problems are essential for high perceptual quality compared to the diffusion distillation method of IBMD, which is developed for general image-to-image translation problems.
>
> **Question 2: Limited novelty**.
>
> **(2) "... Limited Novelty. The main contribution is essentially combining ResShift with VSD/IBMD frameworks. Moreover, the authors also acknowledge the relationship to IBMD (Appendix A.3, Lines 1242-1295). ..."**
>
> Thank you for your comments. Here, we would like to highlight our theoretical contributions and key differences to prior works:
>
> (i) _Discrete time generalization._ First of all, our work generalizes the IBMD framework, which was initially developed for continuous time processes, to a discrete time domain. _The crucial difference is that the optimization objective of discrete-time models is derived from other principles and represents ELBO on the target distribution density_. We think that it was vital because a lot of work leverages discrete-time processes to build their models, such as DDPM, LDM, and ResShift. Indeed, it was not clear whether the proof that was initially introduced in IBMD would fit into these frameworks, mainly because in the work Girsanov theorem (see Proof of Proposition 3.1 in IBMD) was used which is applicable for continuous time processes. We proved our theorem without relying on that fact.
>
> (ii) _Comparisons of VSD-like vs RSD-like loss functions._ The second contribution is that we theoretically show the \underline{difference} between VSD-like (see Equation 16) and RSD-like (see Equation 20) losses, which is illustrated in the Figure 4. In short, we show that the former one matches marginals between the distribution induced by fake ResShift model and distribution induced by teacher ResShift model for each time step, while the latter one matches the joint of these distributions (for more details, please see Appendix A.1). We hypothesize that this joint alignment is particularly beneficial for SR tasks, where maintaining consistency and accuracy across all image details and features is crucial for high-quality resolution (for justification in practice, see Table 1 and 2 ResShift-VSD vs RSD (Ours, distill only)).

---

> ### Author Response · Authors · 2025-11-21
> **Comments on questions**
>
> **Question 3: Experimental claims and PSNR drop of RSD compared to ResShift**.
>
> **(3) "...Misleading Experimental Claims. The authors claim to "outperform the teacher by a large margin" and . However, in Table 1 (RealSR), RSD loses on fidelity metric PSNR. ..."**
>
> For clarity, we will add concrete evaluation perceptual metrics (LPIPS, CLIPIQA, MUSIQ, DISTS, NIQE, MANIQA)
> in the abstract to synchronize the abstract claim with precise statement in **lines 415-418**. We provide comments on experimental claims about RSD as follows.
>
> **Significance of quantitative improvement in perceptual quality**. As noted in **lines 415-418**, RSD outperforms the teacher ResShift model by a large margin for **all perceptual metrics** (LPIPS, CLIPIQA, MUSIQ, DISTS, NIQE, MANIQA) and **all test datasets** (full size RealSR, ImageNet, $512 \times 512$ crops from DIV2K, RealSR, DRealSR, full size DRealSR) while training on the same ImageNet images. This claim is quantitatively supported by the results presented in Table 1, Table 2, Table 3 of the main text, and Table 10 of Appendix D. These results are comparable in terms of improvement of perceptual reference-based and no-reference metrics of SinSR over ResShift (see Table 1 and Table 2 in SinSR (Wang et al., 2024b)) and OSEDiff over SinSR (see Table 1 in OSEDiff (Wu et al., 2024a)).
>
> **Significance of visual improvement in perceptual quality**. The perceptual improvements of RSD over ResShift are **qualitatively supported** by Figures 1, 3 and 11. Concretely, we note in **lines 437-439** that ResShift produces images, which may struggle with severely blurred details, such as the house’s roof in Figure 3. **In lines 147-150**, we also note that blurry images and worse perceptual metrics of ResShift compared to T2I-based Real-ISR models **were reported in related work** (Figure 3 and Section 4.2 in OSEDiff (Wu et al., 2024a), Figure 4 and Section 4.2 in SeeSR (Wu et al., 2024b)). Compared to ResShift, RSD achieves more detailed images.
>
> **Fidelity of RSD and perception-distortion trade-off**. As noted in **lines 464-465**, there is a perceptual-distortion trade-off for SR models (Blau & Michaeli, 2018). As noted in Eq. 9 and detailed in Appendix A.1, the distillation objective of RSD is to minimize KL divergence between the prediction distributions of the ResShift teacher and the RSD student. As shown in Theorem 2 and illustrated in Figures 9 and 10 of (Blau \& Michaeli, 2018), minimizing convex divergences, such as KL divergence, leads to worse fidelity (PSNR). Thus, **a drop in PSNR for RSD compared to the ResShift teacher is expected**. We highlight in **lines 425-427** that the drop for PSNR and SSIM of RSD compared to ResShift is **much smaller** than the drop for PSNR and SSIM of modern T2I-based SR methods, like OSEDiff and SUPIR, which is evident in Tables 1, 2 and 3.
>
> **Control of perception-distortion trade-off in RSD**. ResShift allows to control perception-distortion trade-off using the number of diffusion steps, $T$, see the section "Perception-Distortion Trade-off" in ResShift. RSD can also control a perception-distortion trade-off using the multistep training, which we introduce in Section 3.3. Table 5 shows the impact of $N$ on the results on RealSR dataset. We observe that reference-based metrics, including fidelity metric PSNR, is the best when we match the student and teacher distributions for all intermediate timesteps, which are used by ResShift during training and inference. We choose $N = 4$ for RSD, because it provides the optimal choice between models with good fidelity (ResShift and SinSR) and models with good perceptual and visual quality (OSEDiff and SUPIR).

---

> ### Author Response · Authors · 2025-11-21
> **Comments on questions**
>
> **Question 4: Comparison with recent state-of-the-art SR models**.
>
> **"... The paper ignores recent state-of-the-art methods including: [1] PiSASR (CVPR 2025) [2] TSDSR (Dong et al., 2024) [3] InvSR (CVPR 2025) ... These omissions weaken the comprehensiveness of the experimental comparison. ..."**
>
> We thank the reviewer for pointing out the question of the comparison with these recent state-of-the-art methods. Following your comment, we extend the numerical comparison between the proposed RSD method and PiSA-SR, TSD-SR, InvSR methods in our Tables 1 and 2. We used the officially published pretrained models and inference code from the respective GitHub repositories reported in these works.
>
> Additional results on RealSR in Table 1 are the following:
>
> |**Method**|**NFE**|**PSNR**$\uparrow$|**SSIM**$\uparrow$|**LPIPS**$\downarrow$|**CLIPIQA**$\uparrow$|**MUSIQ**$\uparrow$|
> |-|-|-|-|-|-|-|
> |RSD|1| 25.91 | 0.754|0.273|0.7060|65.860|
> |TSD-SR|1| 24.88 | 0.723|0.281|0.7336|69.871|
> |PiSA-SR|1|25.59|0.750|0.271|0.6678|67.993|
> |InvSR|1|24.93|0.735|0.270|0.6710|66.371|
>
> Additional results on RealSet65 in Table 1 are the following:
>
> |**Method**|**NFE**|**CLIPIQA**$\uparrow$|**MUSIQ**$\uparrow$|
> |-|-|-|-|
> |RSD|1| 0.7267 | 69.172|
> |TSD-SR|1| 0.7263| 70.958|
> |PiSA-SR|1|0.7062|70.208|
> |InvSR|1|0.6828|67.679|
>
> Additional results on ImageNet in Table 2 are the following:
>
> |**Method**|**NFE**|**PSNR**$\uparrow$|**SSIM**$\uparrow$|**LPIPS**$\downarrow$|**CLIPIQA**$\uparrow$|**MUSIQ**$\uparrow$|
> |-|-|-|-|-|-|-|
> |RSD|1| 24.31 | 0.657 | 0.193 | 0.681 | 58.947|
> |TSD-SR|1| 23.58 | 0.645 | 0.197 |	0.673 |	65.299 |
> |PiSA-SR|1| 24.29|0.670|0.213|0.629|62.137|
> |InvSR|1|21.16|0.598|0.304|0.635|54.251|
>
> **{Summary**. We will add these additional comparisons and their discussion in the revised version of the text. Here we summarize the results:
>
> **1. Quantitative comparison**. We observe that our RSD combines high fidelity of relatively small models (ResShift, SinSR) and good perceptual quality of T2I-based SR models (TSD-SR, PiSA-SR, InvSR). TSD-SR, PiSA-SR and InvSR further develops T2I-based SR models to resolve limitations of OSEDiff. Compared to these methods, RSD achieves mostly better fidelity consistency with HR images, which is evident by PSNR and SSIM metrics, with yet competitive perceptual metrics (LPIPS, CLIPIQA, MUSIQ).
>
> **2. Complexity comparison**. Despite the good perceptual performance of TSD-SR, this T2I-based model requires more computational costs compared to the other one-step T2I-based SR model, OSEDiff, and much more computational costs compared to RSD. To show it, we updated Table 4 with additional information on the training time and the corresponding number of GPUs used for the training according to the information reported in this work:
>
> |**Methods**|RSD|TSD-SR|
> |-|-|-|
> | **Inference Step (NFE)** | 1 | 1 |
> | **Inference Time (s)**|0.059| 0.074 |
> | **# Total Parameters (M)** | 174 | 2207 |
> | **Maximum GPU memory (MB)** | 539 | 4611 |
> | **Training time (hours/ # GPUs)** | 5 / 4 A100 | 96 / 8 A100 |
>
> We highlight the training efficiency of our RSD compared to TSD-SR: RSD has $\times 19$ faster training with $\times 2$ times less GPUs compared to TSD-SR. During inference, TSD-SR requires $\times 13$ more parameters and $\times 8$ more GPU memory than RSD. Even with fast inference, the deployment of TSD-SR is limited by a significant computational budget and heavy architecture.
>
> The analysis supports our claim in lines 101-102 that RSD aims to **compromise between fidelity, perceptual quality, and
> computational efficiency**. We will add these results to the revised version of the text.

---

> ### Author Response · Authors · 2025-11-21
> **Comments on questions**
>
> **Question 5: Report on FID**.
>
> **(4) "... FID is not reported, despite being widely used in diffusion-based SR papers such as StableSR and OSEDiff. This limits comparability with existing literature. ..."**
>
> **1. Quantitative comparison**. Following your comment to be consistent with related work (Table 1 in StableSR and Table 1 in OSEDiff), we provide FID for RSD, our main and closely related competitors (ResShift, SinSR), and T2I-based SR models (SUPIR, OSEDiff, TSD-SR) on $512 \times 512$ crops from DIV2K, RealSR and DRealSR, which were used for the evaluation in Table 3 according to lines 375-377.
>
> |**Method**|**NFE**|DIV2K|RealSR|DRealSR|
> |-|-|-|-|-|
> |SUPIR|50| 31.46|128.35|164.86|
> |OSEDiff|1|26.32|123.49|135.30|
> |TSD-SR|1|29.16|114.45|135.24|
> |ResShift|15|42.01|151.53|176.77|
> |SinSR|1|35.57|138.61|172.72|
> |RSD|1|34.84|138.23|167.47|
>
> We note that the original StableSR paper, which introduced the evaluation protocol for our Table 3 in their Table 1, reported FID **only on the DIV2K dataset**. It can be explained by the small number of images in RealSR (100 cropped images) and DRealSR (93 cropped images), while DIV2K consists of 3000 cropped images. FID is known to be a biased estimator, where the bias depends on the model being evaluated, and leads to inconsistent results when varying the sample size (Jayasumana et al., 2024; Chong et al., 2020). We conclude that FID values on RealSR and DRealSR crops, which were reported in OSEDiff, are not representative metrics, but we report them to be consistent with Table 1 in OSEDiff.
>
> In addition, we report FID values on ImageNet-Test dataset to extend our Table 2, which also consists of 3000 images as $512 \times 512$ crops from DIV2K:
>
> |**Method**|**NFE**|ImageNet|
> |-|-|-|
> |SUPIR|50|24.70|
> |OSEDiff|1|23.13|
> |TSD-SR|1|20.55|
> |ResShift|15|30.34|
> |SinSR|1|25.85|
> |RSD|1|25.46|
>
> **Analysis of results**. The results in terms of FID are consistent on all datasets: RSD achieves better distribution alignment compared to ResShift and SinSR, but worse compared to T2I-based SR models (SUPIR, OSEDiff, TSD-SR). The superiority of RSD over SinSR is expected due to its formulation of the distillation loss in Eq. 9 as the KL divergence between teacher and student distributions, while SinSR applies knowledge distillation loss for the teacher with MSE in Eq. 24. We explain good results of T2I-based SR models by their ability to rely on prior information of modern T2I models, such as Stable Diffusion 2.1 (Rombach et al., 2022), SDXL (Podell et al., 2024) and Stable Diffusion 3 (Esser et al., 2024) for OSEDiff, SUPIR and TSD-SR, respectively. However, as we show in Table 4, these models require a large computational budget during inference.
>
> **Summary**. In all testing datasets, RSD has a better FID compared to ResShift and SinSR, but worse compared to T2I-based SR models. The good results in terms of FID for T2I-based SR models are expected by their ability to utilize the prior of modern T2I models, but are also accompanied with the requirement of large computational budget.

---

> ### Author Response · Authors · 2025-11-21
> **Comments on questions**
>
> **Question 6: Visual comparison between RSD and baselines**.
>
> **(5) "... Visual quality does not significantly outperform baseline methods like OSEDiff. For example, differences in Figure 10 (Lines 2295 and 2306) are minimal, which undermines claims of substantial perceptual improvements. ..."**
>
> **Claim on the perceptual quality of RSD**. As noted in lines 098-109, we claim that the aim of RSD is to **improve the compromise between fidelity, perceptual quality, and computational efficiency for Real-ISR** and we do not claim that RSD significantly outperform T2I-based SR models in perceptual quality. For the perceptual quality of RSD, we note in **lines 419-427** that RSD is _competitive_ with OSEDiff. In lines **450-455** we highlight the computational limitations of T2I-based SR baselines compared to RSD due to their heavy architectures.
>
> **Visual comparison with OSEDiff**. Indeed, the results of RSD and OSEDiff on full-size DRealSR dataset are comparable in terms of visual quality, which is evident by Figure 10 and is quantitatively supported by Table 10. In lines **439-440** we note that OSEDiff can produce visual excessive details, which do not correspond well to real-world LR images; see the bear’s nose in Figure 3 and panda’s nose in Figure 1. We also support this claim with visual results on ImageNet and DIV2K; see the bottom image in Figure 11 and the top image in Figure 12.
>
> **Concluding remarks**. We would be grateful if you could let us know if our explanations have been satisfactory. If so, we kindly ask that you consider increasing your rating. We are also open to discussing any other questions you may have.
>
> References:
>
> (Rombach et al., 2022; Podell et al., 2024), the same as in the main text.
>
> Sadeep Jayasumana, Srikumar Ramalingam, Andreas Veit, Daniel Glasner, Ayan Chakrabarti, Sanjiv Kumar. Rethinking FID: towards a better evaluation metric for image generation. CVPR, 2024.
>
> Min Jin Chong and David A. Forsyth. Effectively unbiased FID and inception score and where to find them. CVPR, 2020.

---

> > ### Comment · Reviewer_kZUH · 2025-11-27
> >
> > The authors' rebuttal clarifies the methodological motivation and enhances the approach effectiveness, yet it does not fully resolve concerns regarding the limited theoretical novelty.

---

### Official Review · Reviewer_MSib · 2025-10-31

**Soundness:** 3
**Presentation:** 3
**Contribution:** 3
**Rating:** 6
**Confidence:** 3

**Summary:**

This paper proposes Residual Shifting Distillation (RSD), a novel one-step distillation framework for image super-resolution (SR) based on the ResShift diffusion model. Unlike previous knowledge distillation approaches such as SinSR, which require running the full teacher model through all diffusion steps, RSD introduces a “simulation-free” design by using an auxiliary fake ResShift model to estimate the teacher–student discrepancy through a theoretically derived loss. The method also combines perceptual and adversarial supervision (LPIPS + GAN losses) to further enhance visual realism. The appendices provide mathematical derivations, ablation studies, algorithmic details, proofs, and extensive visual comparisons.

**Strengths:**

1. The paper presents a clear derivation of the RSD loss from a probabilistic perspective, bridging diffusion-based knowledge distillation and joint distribution alignment. The inclusion of Proposition 3.1 and the equivalence to KL divergence (Eq. 9) adds strong theoretical grounding.

2. The proposed “simulation-free” training significantly reduces the computational overhead compared to SinSR, making one-step diffusion models more accessible for real-world scenes.

3. The paper is well-written and well-organized.

**Weaknesses:**

1. The paper does not compare with recent state-of-the-art methods such as AdcSR [1], PiSA-SR[2], CTMSR[3], and TSDSR [4]. This omission makes it difficult to fully evaluate the competitiveness of the proposed method in the context of the latest advances in this field.

2. Table 6 shows a significant decrease in the CLIPIQA score after introducing the GAN loss, but the authors have not discussed this in depth.

3. While the paper mentions the choice of $K=5$ for updating the fake model, it does not delve deeply into how hyperparameter sensitivity (e.g., $\lambda_1$, $\lambda_2$, $K$) affects training stability and final performance. More analysis of this sensitivity, including robustness to different configurations is needed.

4. The evaluation of the paper on real-world datasets is limited. Evaluating on larger and more diverse real-world datasets, such as RealLR200 in SeeSR or RealLQ250 in DreamClear, would strengthen the robustness and credibility of the experimental results.

[1] Adversarial Diffusion Compression for Real-World Image Super-Resolution. CVPR 2025

[2] Pixel-level and Semantic-level Adjustable Super-resolution: A Dual-LoRA Approach. CVPR 2025

[3] Consistency Trajectory Matching for One-Step Generative Super-Resolution. ICCV 2025

[4] TSD-SR: One-Step Diffusion with Target Score Distillation for Real-World Image Super-Resolution. CVPR 2025

**Questions:**

See the weaknesses above.

---

> ### Author Response · Authors · 2025-11-21
> **Comments on questions**
>
> Dear Reviewer MSib, thank you for your comments. We are working on the revised version of the text. Here are the answers to your questions and comments.
>
> **Question 1: Comparison with recent state-of-the-art SR models.**
>
> **(1) "... The paper does not compare with recent state-of-the-art methods such as AdcSR [1], PiSA-SR[2], CTMSR[3], and TSDSR [4]. This omission makes it difficult to fully evaluate the competitiveness of the proposed method in the context of the latest advances in this field. ..."**
>
> We thank the reviewer for pointing out the question of the comparison with these recent state-of-the-art methods. Following your comment, we extend the numerical comparison between the proposed RSD method and the TSD-SR, PiSA-SR methods in our Tables 1 and 2. We used the officially published pretrained models and inference code from the respective GitHub repositories reported in these works.
>
> Additional results on RealSR in Table 1 are the following:
>
> |**Method**|**NFE**|**PSNR**$\uparrow$|**SSIM**$\uparrow$|**LPIPS**$\downarrow$|**CLIPIQA**$\uparrow$|**MUSIQ**$\uparrow$|
> |-|-|-|-|-|-|-|
> |RSD|1|25.91|0.754|0.273|0.7060|65.860|
> |AdcSR|1|25.63|0.735|0.300|0.7033|67.550|
> |PiSA-SR|1|25.59|0.750|0.271|0.6678|67.993|
> |CTMSR|1|26.18|0.765|0.294|0.6449|64.782|
> |TSD-SR|1|24.88|0.723|0.281|0.7336|69.871|
>
> Additional results on RealSet65 in Table 1 are the following:
>
> |**Method**|**NFE**|**CLIPIQA**$\uparrow$|**MUSIQ**$\uparrow$|
> |-|-|-|-|
> |RSD|1|0.7267|69.172|
> |AdcSR|1|0.7044|69.185|
> |PiSA-SR|1|0.7062|70.208|
> |CTMSR|1|0.6871|67.032|
> |TSD-SR|1| 0.7263| 70.958|
>
> Additional results on ImageNet in Table 2 are the following:
>
> |**Method**|**NFE**|**PSNR**$\uparrow$|**SSIM**$\uparrow$|**LPIPS**$\downarrow$|**CLIPIQA**$\uparrow$|**MUSIQ**$\uparrow$|
> |-|-|-|-|-|-|-|
> |RSD|1|24.31|0.657|0.193|0.681| 58.947|
> |AdcSR|1|22.99|0.615|0.252|0.711|63.218|
> |PiSA-SR|1| 24.29|0.670|0.213|0.629|62.137|
> |CTMSR|1|24.72|0.666|0.197|0.692|60.198|
> |TSD-SR|1|23.58|0.645|0.197|0.673|65.299|
>
> **Summary**. We will add these additional comparisons and their discussion in the revised version of the text. Here we summarize the results:
>
> **1. Quantitative comparison**.
> We observe that our RSD combines high fidelity of relatively small models (CTMSR) and good perceptual quality of T2I-based SR models (AdcSR, PiSA-SR, TSD-SR). Compared to CTMSR, which was trained on the same ImageNet data, RSD has a better performance on real-world degradations from RealSR in all perceptual metrics (LPIPS, CLIPIQA, MUSIQ) and slightly worse CLIPIQA and MUSIQ on synthetic ImageNet. The better perceptual quality of RSD compared to CTMSR is also evident on real-world dataset RealSet65 in both perceptual metrics, CLIPIQA and MUSIQ. AdcSR, PiSA-SR and TSD-SR further develops T2I-based SR models to resolve limitations of OSEDiff. Compared to these methods, RSD achieves mostly better fidelity consistency with HR images, which is evident by PSNR and SSIM metrics, with yet competitive perceptual metrics (LPIPS, CLIPIQA, MUSIQ).
>
> **2. Complexity comparison**. Despite the good perceptual performance of TSD-SR, this T2I-based model requires more computational costs compared to the other one-step T2I-based SR model, OSEDiff, and much more computational costs compared to RSD. To show it, we updated Table 4 with additional information on the training time and the corresponding number of GPUs used for the training according to the information reported in this work:
>
> |**Methods**|RSD|TSD-SR|
> |-|-|-|
> | **Inference Step (NFE)** | 1 | 1 |
> | **Inference Time (s)**|0.059| 0.074 |
> | **# Total Parameters (M)** | 174 | 2207 |
> | **Maximum GPU memory (MB)** | 539 | 4611 |
> | **Training time (hours/ # GPUs)** | 5 / 4 A100 | 96 / 8 A100 |
>
> We highlight the training efficiency of our RSD compared to TSD-SR: RSD has $\times 19$ faster training with $\times 2$ times less GPUs compared to TSD-SR. During inference, TSD-SR requires $\times 13$ more parameters and $\times 8$ more GPU memory than RSD. Even with fast inference, the deployment of TSD-SR is limited by a significant computational budget and heavy architecture.
>
> The analysis supports our claim in lines 101-102 that RSD aims to **compromise between fidelity, perceptual quality, and
> computational efficiency**. We will add these results in the revised version of the text.

---

> ### Author Response · Authors · 2025-11-21
> **Comments on questions**
>
> **Question 2: Affect of GAN loss.**
>
> **(2) "... Table 6 shows a significant decrease in the CLIPIQA score after introducing the GAN loss, but the authors have not discussed this in depth. ..."**
>
> We thank the reviewer for pointing out this intriguing observation. We note that this decrease in CLIPIQA is accompanied by the increase of the other no-reference perceptual metric, namely, MUSIQ. The observation that no-reference metrics sometimes do not correlate well was also pointed out in other SR works (see Figure 6 in CTMSR (You et al., 2025)). As noted in **lines 279-283**, our motivation to integrate the GAN loss into the RSD framework was inspired by DMD2 (Yin et al., 2024). In practice, the addition of the GAN loss ($\lambda_{2} \neq 0)$ to the RSD distillation loss ($\lambda_{1,2} = 0)$ quantitatively leads to better correspondence between the predicted and reference images, which is evident in the LPIPS and PSNR metrics in Table 6. We will add visual results to the revised version of the text to support this claim.
>
> **Question 3: Ablation on hyperparameter sensitivity ($K$, $\lambda_{1}$, $\lambda_{2}$).**
>
> **(3) "... While the paper mentions the choice of  $K = 5$ for updating the fake model, it does not delve deeply into how hyperparameter sensitivity (e.g.,  $\lambda_{1}$,  $\lambda_{2}$, $K$) affects training stability and final performance. More analysis of this sensitivity, including robustness to different configurations is needed. ..."**
>
> **Ablation for $K$**. Following your comment, we provide an additional quantitative ablation study on the influence of the hyperparameter $K$ on the final metrics on the full-size RealSR dataset to be consistent with Section 4.3. For this, we chose our final RSD configuration with $K = 5$, which is reported in Tables 1, 2 and 3, and trained the RSD model with $K = 1, 3, 10$.
> The results are summarized as follows:
>
> |$K$|**PSNR**$\uparrow$|**SSIM**$\uparrow$|**LPIPS**$\downarrow$|**CLIPIQA**$\uparrow$|**MUSIQ**$\uparrow$|
> |-|-|-|-|-|-|
> |$K = 1$| 25.99|0.756|0.2713|0.7159|66.247|
> |$K = 3$| 25.86|0.752|0.2701|0.7093|66.140|
> |$K = 5$| 25.91|0.754|0.2726|0.7060|65.860|
> |$K = 10$| 26.27|0.749|0.2732|0.7135|65.233|
>
> Surprisingly, we found that $K = 1$ slightly improves all metrics compared to the originally used value $K = 5$. Moreover, $K = 1$ also reduces the training time of RSD approximately in $2$ times compared to training RSD with $K = 5$, which further improves the training efficiency of RSD compared to baselines. We also report a similar behavior of RSD on the ImageNet-Test dataset following Table 2, which supports our claim that $K = 1$ provides competitive results compared to $K = 5$:
>
> |$K$|**PSNR**$\uparrow$|**SSIM**$\uparrow$|**LPIPS**$\downarrow$|**CLIPIQA**$\uparrow$|**MUSIQ**$\uparrow$|
> |-|-|-|-|-|-|
> |$K = 1$|24.28|0.657|0.196|0.697|59.499|
> |$K = 3$|24.11|0.644|0.191|0.675|59.535|
> |$K = 5$|24.31|0.657|0.193|0.681|58.947|
> |$K = 10$|24.01|0.639|0.193|0.671|59.110|
>
> As noted in **lines 1361-1363**, we used $K = 5$ to follow the DMD2 strategy in number of updates for the fake model per student update (Yin et al., 2024a); see Figure 9 in DMD2 for the analysis of the impact of $K$ on the training stability for image generation problems. Our results show that RSD training  with $K = 1$ can also be beneficial for Real-ISR problems while not breaking the good performance of $K = 5$. Overall, the results for all $K = 1, 3, 5, 10$ are close to each other with small variation and will be added in revised version of the text .
> However, when we repeat the same experiments across different values of $K$ _without_ additional supervised losses, we observe that training with $K = 1$ becomes highly unstable and leads to clearly degraded quantitative metrics, which also was observed in DMD2. In the final version of the paper, we will add convergence plots for different $K$ to substantiate this observation.
>
> **Ablation for $\lambda_{1}$, $\lambda_{2}$**. As noted in **lines 1364-1366**, we set $\lambda\_{1} = 2$ and $\lambda\_{2} = 3 \cdot 10\^{-3}$ following OSEDiff and DMD2, respectively. In Table 6, we provide the numerical results of adding supervised losses in RSD on the RealSR dataset and discuss the improvements compared to RSD with distillation only loss in **lines 476-479**. There might be an area for additional performance improvements for RSD in finetuning hyperparameter values for $\lambda\_{1}, \lambda\_{2}$, but we found that the initial values already provide good performance. We plan to address this question in future work.
>
> **Summary**. RSD with $K = 1$ leads to similar performance compared to $K = 5$, also accelerates the training process and does not break the training stability of RSD. We used the values $\lambda\_{1} = 2$ and $\lambda\_{2} = 3 \cdot 10\^{-3}$ following OSEDiff and DMD2, which already provides good performance. Finetuning of hyperparameters $\lambda\_{1}$ and $\lambda\_{2}$ is left for the future work.

---

> ### Author Response · Authors · 2025-11-21
> **Comments on questions**
>
> **Question 4: Comparison on larger and more diverse real-world datasets.**
>
> **(4) "... The evaluation of the paper on real-world datasets is limited. Evaluating on larger and more diverse real-world datasets, such as RealLR200 in SeeSR or RealLQ250 in DreamClear, would strengthen the robustness and credibility of the experimental results. ..."**
>
> As we note in **lines 373-377**, we follow standard evaluation protocols of SinSR and OSEDiff to be consistent with the prior evaluation of competitors.
>
> Following your comment, we extended the experimental evaluation with two additional real-world datasets: RealLR200 (Wu et al., 2024b) and RealLQ250 (Yuang et al., 2024). We provide quantitative results for RSD, closest baselines, including the teacher ResShift model and SinSR, and T2I-based SR models, such as OSEDiff, SUPIR and TSD-SR. Since both datasets do not have reference HR images, we provided only no-reference perceptual metrics, including CLIPIQA, MUSIQ, NIQE, and MANIQA, which follow the evaluation protocol of SeeSR (Wu et al., 2024b) and DreamClear (Yuang et al., 2024).
>
> The results on RealLR200 are summarized as follows:
>
> |**Method**|**NFE**|**CLIPIQA**$\uparrow$|**MUSIQ**$\uparrow$|**NIQE**$\downarrow$|**MANIQA**$\uparrow$|
> |-|-|-|-|-|-|
> |SUPIR|50| 0.6188|64.79|4.1862|0.6120|
> |OSEDiff|1| 0.6728|69.45|4.0506|0.6153|
> |TSD-SR|1|0.7335|72.06|3.8352|0.6150|
> |ResShift|15| 0.6368|61.80|5.7016|0.5367|
> |SinSR|1| 0.7089|64.90|5.3327|0.5507|
> |RSD|1| 0.7151|68.66|4.7074|0.5865|
>
> The results on RealLQ250 are summarized as follows:
>
> |**Method**|**NFE**|**CLIPIQA**$\uparrow$|**MUSIQ**$\uparrow$|**NIQE**$\downarrow$|**MANIQA**$\uparrow$|
> |-|-|-|-|-|-|
> |SUPIR|50| 0.5746|65.72|3.6607|0.5969|
> |OSEDiff|1| 0.6724|69.56|3.9682|0.5889|
> |TSD-SR|1|0.7369|73.22|3.6996|0.5924|
> |ResShift|15| 0.6348|61.99|5.7622|0.5295|
> |SinSR|1| 0.7142|65.29|5.4629|0.5259|
> |RSD|1| 0.7252|69.63|4.5531|0.5739|
>
> We analyze the results as follows:
>
> **Comparison between RSD, SinSR, and ResShift**. Quantitative results on both additional real-world datasets support our claim in **lines 415-418** that RSD outperforms the teacher ResShift model and our closest competitor, SinSR, in all perceptual metrics (CLIPIQA, MUSIQ, NIQE, MANIQA).
>
> **Comparison between RSD, OSEDiff, SUPIR, and TSD-SR**.
> Compared to large T2I-based SR models, RSD achieves competitive CLIPIQA and MUSIQ values, but worse NIQE and MANIQA. Among all methods, TSD-SR achieves the best values in most perceptual no-reference metrics on both datasets. This result can be explained by a huge prior extracted using the distillation of the teacher model, which is the recent text-to-image SD3 model (Esser et al., 2024), and also accompanied by computational limitations, as we discuss in the response to Question 1: the biggest architecture size and required GPU memory during the inference, and the longest training time.
>
> **Summary**. Compared to the closest relevant competitors, ResShift and SinSR, RSD outperforms in all no-reference perceptual metrics (CLIPIQA, MUSIQ, NIQE, MANIQA), while having competitive CLIPIQA and MUSIQ and worse NIQE and MANIQA when compared to recent T2I-based SR models (OSEDiff, SUPIR, TSD-SR). At the same time, RSD requires much less computational inference budget compared to T2I-based SR models. We will add these quantitative results and support them with visual results on both RealLR200 and RealLQ250 datasets to the revised version of the text.
>
> **Concluding remarks**.
> We would be grateful if you could let us know if our explanations have been satisfactory. If so, we kindly ask that you consider increasing your rating. We are also open to discussing any other questions you may have.
>
> References:
>
> (Wu et al., 2024b; Yin et al., 2024), the same as in the main text.
>
> Ai Yuang, Zhou Xiaoqiang, Huang Huaibo, Han Xiaotian, Chen Zhengyu, You Quanzeng, Yang Hongxia. DreamClear: High-Capacity Real-World Image Restoration with Privacy-Safe Dataset Curation. In Advances in Neural Information Processing Systems, 2024.
>
> Patrick Esser, Sumith Kulal, Andreas Blattmann, Rahim
> Entezari, Jonas Muller, Harry Saini, Yam Levi, Dominik Lorenz, Axel Sauer, Frederic Boesel, et al. Scaling rectified flow transformers for high-resolution image synthesis.
> In Forty-first International Conference on Machine Learning, 2024.

---

> ### Comment · Reviewer_MSib · 2025-11-28
>
> Thank you to the authors for the detailed rebuttal. The authors' responses have addressed my main concerns, and I am maintaining my positive rating.

---

### Official Review · Reviewer_wG89 · 2025-11-02

**Soundness:** 2
**Presentation:** 2
**Contribution:** 2
**Rating:** 2
**Confidence:** 5

**Summary:**

This paper introduces "RSD," a novel distillation method for the ResShift super-resolution (SR) model, aiming to reduce high computational costs while improving quality over existing acceleration methods like SinSR (unrealistic details) and OSEDiff (hallucinated structures).

The method involves training a student network to generate images, which are then used to train a "fake ResShift model." The objective is to make this "fake" model's performance "coincide with" the original teacher model.

**Strengths:**

The indirect training approach (training a student to generate data to train a proxy model) is a novel contribution to knowledge distillation for generative models.

**Weaknesses:**

The paper’s ambitious claims are not adequately supported by evidence. The core contribution remains vague: the notions of a “new fake ResShift model” and making it “coincide with” the teacher are abstract, lacking mathematical formulation or clear motivation. The statement that a distilled student “outperforms the teacher by a large margin” is implausible without thorough justification—this raises concerns about whether the teacher was properly optimized or if “outperform” is defined using non-standard metrics.

The abstract also includes unqualified claims (“surpass,” “on par”) without specifying the evaluation criteria (e.g., PSNR, LPIPS, FID). Given the method’s complexity, the authors should consider a simpler baseline—quantifying ResShift directly to obtain a lightweight variant for comparison. Without such analyses, the claimed advantages remain unconvincing.

**Questions:**

See the weekness secton. In summary, the abstract proposes an intriguing and timely solution to a significant problem. However, its credibility is undermined by a vague methodology and exceptionally strong claims that lack clear evidence. The assertion that a student model can dramatically outperform its teacher is extraordinary and demands rigorous proof and explanation.

---

> ### Author Response · Authors · 2025-11-21
> **Comments on questions**
>
> Dear Reviewer wG89, thank you for your comments. We are working on the revised version of the text. Here are the answers to your questions and comments.
>
> **Question 1: Methodology and motivation of RSD.**
>
> **(1) " ... In summary, the abstract proposes an intriguing and timely solution to a significant problem. However, its credibility is undermined by a vague methodology ..."**
>
> Our methodology and motivation of RSD presented in the text are theoretically and practically justified.
>
> **Core motivation and result**. We provide our core motivation for the training of the one-step student model $G\_{\theta}$ in **lines 211–215 and Eq. 6**. We propose its mathematical formulation as the minimization problem for the student model $G_{\theta}$ with the objective $\mathcal{L}\_{\theta}$ in **lines 236-240 and Eq. 7**. We give a discussion about the computational intractability of the gradient for $\mathcal{L}\_{\theta}$ in **Appendix I and lines 2143-2159**. To alleviate this problem, we derive the equivalent form of $\mathcal{L}\_{\theta}$ in **Proposition 3.1 and Eq. 8**, which involves the training of an additional ("fake") ResShift model $f\_{\phi}$. In Section 4, we provide a numerical and qualitative comparison between the proposed RSD model, the teacher ResShift model, and the SinSR model (Wang et al., 2024b), which also distills the same teacher model of ResShift.
>
> **Additional motivation**. We also note about **the additional point of view** on the motivation for the proposed RSD objective in **lines 255-258 and Eq. 8 in the main text**, which is based on KL divergences and detailed in Appendix A.1 (see Eq. 20 and its derivation in lines 1018-1059 in supplementary material). In Section 3.6 and **lines 299-313**, we discuss the relation of the RSD objective to distillation losses, which were implemented in related works on diffusion distillation for the Real-ISR problem. In Section 4, we provide a numerical and visual comparison between the proposed RSD model and the OSEDiff model (Wu et al., 2024a), which also requires training of an additional fake model.
>
> **Summary**. Based on two motivation points of view on the proposed distillation objection in Eq. 7, mathematical derivation of Proposition 3.1, and quantitative, qualitative, and computational comparison in Section 4.1 between the proposed RSD method and related distillation methods for Real-ISR problems, we consider our methodology to be theoretically and empirically justified. _Could you explain, please, what you mean by "vague methodology"_?
>
> **Question 2: Notation in RSD.**
>
> **(2) " ... The core contribution remains vague: the notions of a “new fake ResShift model” and making it “coincide with” the teacher are abstract, lacking mathematical formulation or clear motivation ..."**
>
> **Explanation for our notation**. The claim of the abstract in **lines 014-016** is formally stated in **Proposition 3.1** and can be mathematically explained by the **proof of Proposition 3.1 in Appendix I**. The explanation of this claim is already given in the main text as follows.
>
> **Justification for "new fake ResShift model"**. In the abstract claim, we denote "a new fake ResShift model" as the model $f\_{\phi}$ in Proposition 3.1 and formalize the claim in **lines 251-252**. This notation is explained in the first stage of the proof for Proposition 3.1 in Appendix I; see **lines 2054-2092**. This explains the part "a new fake ResShift model trained on them (images generated by the student network)".
>
> **Justification for "making it coincide with the teacher model"**. The minimization problem in Eq. 7 mathematically explains the meaning of "making it coincide with the teacher model". As noted in **lines 236-241**, we propose this objective to achieve the matching $f\_{G\_{\theta}}(x\_{t}, y\_{0}, t) \approx f\^{*}(x\_{t}, y\_{0}, t)$ for all timesteps $t$, where $x\_{t}$ is generated using the posterior distribution $q(x\_{t}| \widehat{x}\_{0}, y\_{0})$ in Eq. 2 with $\widehat{x}\_{0} = G\_{\theta}(x\_{T}, y\_{0}, \epsilon)$.
>
> **Motivation for the objective proposed in Eq. 7**. Our motivation for the objective $\mathcal{L}\_{\theta}$, which was proposed in Eq. 7, is formulated in **lines 212-215 and Eq. 6**. The motivation for the proposed objective $\mathcal{L}\_{\theta}$ in Eq. 7 is also mathematically supported by its representation as the expectation of KL divergence between the joint distributions $p(x\_{0:T}|y\_{0})$ and $p\^{*}(x\_{0:T}|y\_{0})$, which is detailed in Appendix A.1 in **Eq. 20 and lines 1018-1059**.
>
> **Summary**. Based on the mathematical formulation of the proposed optimization problem in Eq. 7, its tractable solution based on the theoretically justified Proposition 3.1, and the additional motivation in Appendix A.1, we consider our core contributions to have precise mathematical formulation with detailed and motivated presentation. _Could you explain, please, what you mean by "vague core contribution"_?

---

> ### Author Response · Authors · 2025-11-21
> **Comments on questions**
>
> **Question 3: Comparison between RSD and the teacher ResShift models.**
>
> **(2.a) ... "The statement that a distilled student “outperforms the teacher by a large margin” is implausible without thorough justification—this raises concerns about whether the teacher was properly optimized or if “outperform” is defined using non-standard metrics." ...**
>
> This statement is precisely stated in **lines 415-418 in Section 4.2** as follows: RSD outperforms the teacher ResShift model by a large margin for **all perceptual metrics** (LPIPS, CLIPIQA, MUSIQ, DISTS, NIQE, MANIQA) and **all test datasets** (full size RealSR, ImageNet, $512 \times 512$ crops from DIV2K, RealSR, DRealSR, full size DRealSR) while training on the same ImageNet images. As noted in **lines 371-373** in the main text, **lines 1475-1480** of Appendix C, and Table 9 of Appendix D, for RSD distillation and a fair comparison with RSD we used the same officially trained weights of the teacher model, which are provided in the official GitHub repository by the authors of ResShift paper. As we note in **lines 404-411**, the **reported perceptual metrics are standard in related work** and follow Table 1 and Table 2 in SinSR, Table 1 in OSEDiff, and Table 3 and Table 4 in ResShift.
>
> **Quantitative support and metrics**. This claim is **quantitatively supported** by the results presented in Table 1, Table 2, Table 3 of the main text, and Table 10 of Appendix D. For example, on the RealSR dataset (Table 1), RSD achieves LPIPS = $0.273$ and CLIPIQA = $0.7060$, which are around 24% and 18%, respectively, improvements over ResShift results on the same dataset, which are LPIPS = $0.360$ and CLIPIQA = $0.5958$, respectively. We visualize these improvements in the right part of Figure 1. These results are comparable in terms of improvement of perceptual reference-based and no-reference metrics of SinSR over ResShift (see Table 1 and Table 2 in SinSR (Wang et al., 2024b)) and OSEDiff over SinSR (see Table 1 in OSEDiff (Wu et al., 2024a)).
>
> **Teacher optimization and fairness of comparison**.  According to Section 4.1 of the ResShift paper, this model was trained for 500K optimization steps. We also ran this checkpoint using the official ResShift implementation, which gave us reproducible results as are reported in Tables 3 and 4 in the ResShift paper. Thus, we consider the used teacher ResShift model to be properly optimized and consistent with the results reported in the ResShift paper. As noted in **lines 369-372**, RSD follows the training setup of ResShift on the same ImageNet dataset, which explains the fairness of the comparison between those models.
>
> **Summary**. For better clarification, we will add "in all perceptual metrics" to the claim in the abstract. Based on the results presented in Section 4.2, we consider the difference between the ResShift model and RSD to be significant, our comparison to be fair, and the evaluation protocol to be standard and follow related work. _Could you explain, please, what you mean concretely by "the statement without thorough justification"?_
>
> **(3.b) ... "Given the method’s complexity, the authors should consider a simpler baseline—quantifying ResShift directly to obtain a lightweight variant for comparison" ...**
>
> As noted in our previous comment, in the main text we directly provided quantitative results of ResShift in Tables 1, 2, 3 and visual results in Figures 1 and 2. _Can you explain, please, what you mean by "quantifying ResShift directly to obtain a lightweight variant for comparison"?_

---

> ### Author Response · Authors · 2025-11-21
> **Comments on questions**
>
> **(3.c) ... "The assertion that a student model can dramatically outperform its teacher is extraordinary and demands rigorous proof and explanation."**
>
> We note that the phenomenon of surpassing the teacher model by the student model was observed in related diffusion distillation literature (Zhou et al., 2024; Yin et al., 2024; Wang et al., 2024b) and **is not extraordinary**.
>
> **Surpassing the teacher by the student without supervised losses**. As reported in Section 5.1 of the SiD work (Zhou et al., 2024), the authors were able to distill the EDM teacher (Karras et al., 2022) into the one-step student model with their data-free diffusion distillation method, which outperforms the teacher diffusion model in the same image generation problem for the ImageNet dataset. This remarkable outcome implies that the teacher model, which utilizes the pretrained score function to generate images using multiple steps and **naturally accumulates discretization errors during reverse diffusion**, might not be as efficient as a one-step distillation student model. The one-step student, by sidestepping error accumulation, could align better with the target data distribution when the model-based score-matching loss is completely minimized. As shown in Table 1 and Table 2, our proposed one-step distillation model without supervised losses - RSD (Ours, distill only) - is also able to outperform the teacher ResShift model in no-reference perceptual metrics (CLIPIQA, MUSIQ).
>
> **Surpassing the teacher by the student with supervised losses**. In the SR diffusion distillation literature, our closest competitor, SinSR (Wang et al., 2024b), also distilled the ResShift model into the one-step student model. As stated in Section 5.2 of the SinSR paper, SinSR was able to "outperform the teacher model that we used by a large margin" using the additional supervised consistency preserving loss proposed in Eq. 8 of their paper; see Table 1 of the SinSR paper. This result also aligns with our Tables 1 and 3, which show that SinSR consistently outperforms the ResShift model in various perceptual metrics (LPIPS, DISTS, NIQE, MUSIQ, NIQE, MANIQA) and test datasets (RealSR, RealSet65, $512 \times 512$ crops from DIV2K, RealSR, DRealSR). As we discuss in Section 3.4, we also integrate additional LPIPS and GAN losses into the final RSD model following related diffusion distillation methods (Wu et al., 2024a; Yin et al., 2024a); see Eq. 11. Similarly to SinSR, since we utilized supervised losses, it is possible to outperform the teacher ResShift model with one-step student model. The surpassing of the multistep teacher model using the one-step student model with supervised GAN loss was also reported in Section 5 of DMD2 (Yin et al., 2024).
>
> **Summary**. We are able to outperform the teacher ResShift model in **all perceptual metrics** (LPIPS, CLIPIQA, MUSIQ, DISTS, NIQE, MANIQA) and **all test datasets** (full size RealSR, ImageNet, $512 \times 512$ crops from DIV2K, RealSR, DRealSR, full size DRealSR) by a large margin due to the utilization of supervised losses, which is consistent with SinSR and SiD results. We will add this discussion to the revised version of the text.
>
> **Question 4: Claims in abstract.**
>
> **(4) ... "The abstract also includes unqualified claims (“surpass,” “on par”) without specifying the evaluation criteria (e.g., PSNR, LPIPS, FID)." ... "In summary, the abstract proposes an intriguing and timely solution to a significant problem. However, its credibility is undermined by a vague methodology and exceptionally strong claims that lack clear evidence." ...**
>
> For clarity, we will add concrete evaluation perceptual metrics (LPIPS, CLIPIQA, MUSIQ, DISTS, NIQE, MANIQA) in the abstract to synchronize the abstract claim with **lines 415-418**. The evidence of empirical claims in the abstract is supported by experimental Section 4.
>
> **Concluding remarks.** We would be grateful if you could let us know if our explanations have been satisfactory. If so, we kindly ask that you consider increasing your rating. We are also open to discussing any other questions you may have.
>
> References:
>
> (Wang et al., 2024b; Wu et al., 2024a,b; Zhou et al., 2024; Yin et al., 2024a), the same as in the main text.
>
> Karras, T., Aittala, M., Aila, T., and Laine, S. Elucidating the design space of diffusion-based generative models. In Advances in Neural Information Processing Systems, 2022.

---

### Author Response · Authors · 2025-11-21
**Working on revised text**

**Author Final Remarks**.

We thank all reviewers for fruitful discussions. Based on our rebuttal and provided feedback, we are working on the list of changes and additional clarifications for the revised version of our text

---

### Comment · Area_Chair_JEbt · 2025-11-23
**The authors' rebuttal is available. Please read, comment, and discuss.**

Dear Reviewers,

Thanks for your time and effort in reviewing ICLR2026 submissions. The authors have provided their responses to your review. Please read and raise your further comments, and discuss with the authors.

Best regards,

Your AC

---

### Author Response · Authors · 2025-12-03
**Revised version of the text**

We thank all reviewers for their valuable comments and discussion.
Guided by the reviews and rebuttal discussion, we have revised the paper. **We have substantially strengthened both the theoretical analysis and the empirical evaluation of RSD, and we summarize the main revisions below**. All modifications in the revised manuscript are highlighted in orange.

**1. Abstract and clarity** (Reviewers wG89, kZUH, mz7A)

**We revised the abstract to state our claims more precisely and concretely**. We now explicitly report perceptual metrics (LPIPS, CLIPIQA, MUSIQ, DISTS, NIQE, MANIQA) that demonstrate the perceptual superiority of RSD over the teacher model ResShift and SinSR. We clarified RSD’s computational efficiency relative to T2I-based SR models in terms of parameter count, GPU memory, and training cost. We also updated the size of the captions in Figures 2 and 3, and added Table 9 explaining all variables in the initial pixel and latent spaces.

**2. Formulation of contributions** (Reviewers wG89, kZUH, mz7A)

**We refined the contribution statements**. On the theoretical side, we now explicitly analyze the proposed RSD objective and its relationship to VSD and IBMD objectives. On the practical side, we strengthened our claims with a new quantitative comparison between RSD and IBMD (Appendix A.3). We emphasize the improved performance–efficiency trade-off of RSD relative to SinSR and OSEDiff.

**3. Comparison with recent state-of-the-art diffusion SR models** (Reviewers MSib, kZUH, mz7A)

**We expanded the related work section to include the recent diffusion SR models highlighted by the reviewers**: PiSA-SR, AdcSR, CTMSR, and InvSR.  We added quantitative and visual comparisons with TSD-SR, PiSA-SR, AdcSR, and CTMSR in Tables 1-4 and Figure 3.
We introduced a new Appendix E that provides a detailed performance–efficiency analysis of RSD and six recent state-of-the-art diffusion SR models: PiSA-SR, AdcSR, CTMSR, InvSR, TSD-SR, and CCSR. We also expanded the full quantitative results in Appendix D to include these models (Tables 13-17), and added further visual comparisons in Figures 5 and 6 in addition to Figure 3.

**4. Generalizability of RSD** (Reviewer mz7A)

We added a dedicated discussion of the generalization properties of RSD in Appendix A.4.

**5. Additional results and ablation studies** (Reviewer MSib)

**We extended the experiments with additional real-world SR benchmarks proposed by Reviewer MSib**: RealLR200 and RealLQ250. Quantitative results are reported in Table 14, and visual comparisons are shown in Figures 5 and 6. We added a new Appendix F with further ablation studies on the hyperparameter $K$ and the stability of RSD training. We also complemented the quantitative results of Table 6 (ablation on supervised losses) with corresponding visual examples in Figure 8.

**6. Performance–efficiency trade-off and limitations** (Reviewer mz7A)

Following Reviewer mz7A’s suggestion, we extended Table 4 with training time for RSD and all competitors to **clearly demonstrate RSD’s computational efficiency**. We further elaborated the performance–efficiency trade-off between RSD and recent diffusion-based SR models in Section 4.2 and Appendix E.  We now explicitly state the teacher-capacity limitation in the conclusion and provide supporting evidence in Appendix G, where we train RSD for high-resolution images of size $512 \times 512$.

**Concluding remarks**

We kindly ask Area Chair and Reviewers to check the revised version of the text which addresses the concerns and take into account the edits in the revision when making the final decision. We believe the revision fully addresses all raised questions and concerns, both in scope and depth.

---

### Meta-Review · Area_Chair_L89M · 2026-01-05

**Summary:**

The submission proposes Residual Shifting Distillation (RSD), a one-step distillation framework tailored to ResShift for real-world super-resolution. Reviewers agree the problem is relevant and that the paper contains substantial technical material; in particular, one reviewer highlights the principled objective, broad experimental coverage, and strong efficiency/deployability.

However, the reviews raise consistent concerns about (i) limited/ incremental novelty relative to closely related distillation and joint-alignment ideas (notably its closeness to IBMD-like formulations), and (ii) positioning and empirical completeness, including missing or insufficient comparisons to very recent one-step SR baselines and questions about fairness/comparability of some settings (e.g., resolution/training setup).

Critically, one reviewer (high confidence) considers the paper’s core contribution and methodology too vague and the claims (e.g., outperforming the teacher by a large margin) insufficiently justified, which undermines credibility. Given the mixed scores and the remaining uncertainty around novelty/clarity and empirical positioning, my overall recommendation is reject.

**Reviewer Concerns:**

1) Comparisons: Reviewers requested inclusion of newer one-step SR baselines (e.g., CCSR/TSD-SR and other recent diffusion SR models). The discussion indicates the authors attempted to address missing comparisons and strengthen the performance–efficiency narrative.

2) Clarity: The authors indicated they would revise overly strong phrasing in the abstract and improve presentation issues (raised by multiple reviewers).

**Reviewer Scores:**

Had the reviewer been able to fully participate in the discussion, I believe their score would likely have remained largely unchanged. I appreciate the feedback provided and will carefully address these points in a revised version of the manuscript.

---

### Decision · Program_Chairs · 2026-01-26

Reject